# Faithful transfer of radiolarian silicon isotope signatures from water column to sediments in the South China Sea

Qiang Zhang[1, 2], George E. A. Swann[2], Vanessa  Pashley[3], and Matthew S. A. Horstwood[3]

[1]Key Laboratory of Ocean and Marginal Sea Geology, South China Sea Institute of Oceanology, Chinese Academy of Sciences, Guangzhou 510301, China
[2]School of Geography, University of Nottingham, University Park, Nottingham, NG7 2RD, UK
[3]National Environmental Isotope Facility, British Geological Survey, Keyworth, Nottingham, NG12 5GG, UK

**Correspondence:** Qiang Zhang (zhangqiang210@scsio.ac.cn)

**Abstract.** Radiolarian silicon isotopes ($\delta^{30}\mathrm{Si_{rad}}$) hold significant potential as a proxy for constraining past silicon cycling in seawater. However, the extent to which $\delta^{30}\mathrm{Si_{rad}}$ signatures in sediments accurately represent the isotopic signals of the overlying water column remains unclear, particularly under the influence of radiolarian shell dissolution during sinking and burial in the sediment record. This study presents the first comparative analysis of $\delta^{30}\mathrm{Si_{rad}}$ compositions and the radiolarian assemblage community using water column and surface sediment samples collected from the South China Sea (SCS). The results indicate that $\delta^{30}\mathrm{Si_{rad}}$ values range from 1.56-1.83‰ (mean = 1.74‰) in the water column and from 1.61-1.85‰ (mean = 1.73‰) in surface sediments, with the fractionation factor for $\delta^{30}\mathrm{Si_{rad}}$ varying from –0.33‰ to –0.92‰ (mean = 0.58‰). $\delta^{30}\mathrm{Si_{rad}}$ signatures in the water column are primarily contributed to by radiolarians from the 0-100 m water depth layer. No significant discrepancies in $\delta^{30}\mathrm{Si_{rad}}$ values were observed between plankton and sediment samples at each sampling station, as evidenced by the paired t-test ($p = 0.75$), implying that dissolution has a minimal impact on $\delta^{30}\mathrm{Si_{rad}}$ during the transfer of radiolarian shells to the sediment record. This finding may be enhanced by the dominance of more dissolution-resistant Spumellaria and Nassellaria taxa (>99% relative abundance) within the radiolarian community, coupled with the scarcity or absence of the readily dissolvable radiolarian taxa in the analysed samples. This study demonstrates the faithful preservation of the $\delta^{30}\mathrm{Si_{rad}}$ signature and its potential for studying past changes in the marine silicon cycle.

## 1   Introduction

Silicon (Si) is a crucial nutrient in marine ecosystems, playing a significant role in both the global carbon cycle and climate change. For example, during glacial periods the Southern Ocean exhibits reduced rates of primary productivity and silicon utilization efficiency due to the alleviation of iron (Fe) limitation by enhanced aeolian deposition, resulting in an excess of silicic acid ($\mathrm{Si(OH)_4}$) in the water column (Matsumoto et al., 2002; Brzezinski et al., 2002). Based on this, the "silica acid leakage hypothesis" (SALH) was proposed, in which excess $\mathrm{Si(OH)_4}$ within glacial Antarctic intermediate water was transported to lower latitudes, leading to enhanced diatom productivity at the expense of carbonate export (Brzezinski et al., 2002). This shift to diatom growth over carbonate export was expected to result in higher seawater alkalinity, and lower atmospheric $p\mathrm{CO_2}$ (Matsumoto et al., 2002; Matsumoto and Sarmiento, 2008; Ellwood et al., 2010) and is supported by significant increases

in glacial siliceous productivity in lower latitude regions of the equatorial Atlantic and low-latitude Pacific Oceans (e.g.,
Kienast et al., 2006; Bradtmiller et al., 2007; Arellano-Torres et al., 2011; Hendry et al., 2021). However, other findings have
challenged this hypothesis by suggesting that siliceous productivity and surface water $Si(OH)_4$ concentrations were higher
during interglacials (e.g., Bradtmiller et al., 2006; Dubois et al., 2010). To gain a deeper insight into the mechanisms and
impact of marine silicon cycling in both glacials and other temporal/spatial scales across the geological record, it is essential to
have a more comprehensive understanding of the relationships between marine $Si(OH)_4$ concentrations, primary productivity
and the climate system.

    The silicon isotope ($\delta^{30}Si$) composition of biogenic silica (BSi), primarily comprising radiolarians, diatoms, and sponge
spicules, serves as a key indicator for quantifying marine silicon cycling and its response to changing oceanographic and
climate processes. Whereas the $\delta^{30}Si$ of diatoms, inhabiting the photic zone of the water column, primarily reflects changes in
silicic acid utilization/supply in the surface ocean (De La Rocha et al., 1998; Reynolds et al., 2006; Egan et al., 2012; Abelmann
et al., 2015; Swann et al., 2016, 2017; Hendry et al., 2021; Worne et al., 2022), the $\delta^{30}Si$ of sponge spicules, inhabiting bottom
waters, are closely related to deep ocean $Si(OH)_4$ concentrations (De La Rocha, 2003; Wille et al., 2010; Hendry et al., 2010;
Hendry and Robinson, 2012; Hendry et al., 2014; Fontorbe et al., 2016). More recently, attempts have been made to use the
$\delta^{30}Si$ of radiolaria shells ($\delta^{30}Si_{rad}$), which typically occupy intermediate water depths in high latitudes (Boltovskoy, 2017),
to fill in the gaps between surface (diatom) and bottom (sponge) measurements of $\delta^{30}Si$ and so allow whole water column
reconstructions of silicon cycling (e.g., Abelmann et al., 2015).

    Numerous studies have demonstrated that c. 97% of biogenic silica, specifically for diatoms in surface waters, is dissolved
during sinking through the water column and/or early diagenesis (Tréguer et al., 1995; Ragueneau et al., 2000; Tréguer and
De La Rocha, 2013). The potential for this dissolution to fractionate and alter BSi $\delta^{30}Si$ is under debate. Cardinal et al. (2007)
reported a positive relationship between the silicon isotopic fractionation factor and net diatom production in the Southern
Ocean, implying a dissolution-driven isotope effect. Demarest et al. (2009) then conducted dissolution experiments on nat-
ural diatom frustules, observing a silicon isotope fractionation factor of -0.55 ± 0.05‰ with lighter isotopes preferentially
released. However, others have showed that diatom dissolution is not associated with significant silicon isotope fractionation
in both laboratory experiments (Wetzel et al., 2014) and in studies monitoring the sinking and burial of diatoms in the natural
environment (Varela et al., 2004; Fripiat et al., 2012; Egan et al., 2012; Closset et al., 2015; Panizzo et al., 2016; Grasse et al.,
2021). These inconsistent findings make it unclear to what extent silicon isotopic fractionation might impact the $\delta^{30}Si$ records
of other forms of biogenic silica preserved in the sediments.

    Polycystine radiolarians, a primary component of biogenic silica in tropical ocean surface sediments (Lisitzin, 1972; Zhang et
al., 2015), typically inhibit the water column from the surface to the deep ocean (e.g., Kling, 1979; Hu et al., 2015; Boltovskoy,
2017) and exhibit a greater resistance to dissolution (Suzuki and Aita, 2011; Morley et al., 2014) than the more fragile and solu-
ble diatoms. However, existing investigations of $\delta^{30}Si_{rad}$ have been limited by difficulties in successfully culturing radiolarians
and/or purifying their tests from the natural environment (Suzuki and Aita, 2011; Suzuki and Not, 2015; Zhang and Swann,
2023). Whilst a few studies have explored $\delta^{30}Si_{rad}$ compositions in surface sediments and their correlations with $Si(OH)_4$ con-
centrations in the overlying water column (Ding et al., 1996; Egan et al., 2012; Abelmann et al., 2015; Doering et al., 2021), as

well as their response to changing climatic conditions (Hendry et al., 2014; Fontorbe et al., 2016), the impact of dissolution on $\delta^{30}Si_{rad}$ during the sinking and burial of radiolarian shells into sediments remains unknown. Given the lightweight nature of shells and high diversity of radiolarians, particularly in low-latitude oceans (Moore, 1969; Takahashi, 1982; Boltovskoy et al., 2010), combined with the observation that most species typically comprise less than 10% of the total radiolarian assemblage in both the water column and the sedimentary record (Chen et al., 2008a; Boltovskoy et al., 2010), obtaining sufficient material from individual radiolarian taxa for silicon isotopic analysis remains a considerable challenge. Using a new method for extracting and purifying bulk radiolarian tests from sediments (Zhang and Swann, 2023), this study documents the $\delta^{30}Si_{rad}$ in water column and surface sediments in the South China Sea (SCS) and then determines 1) the water column depth interval from which the radiolarian community contributes to $\delta^{30}Si_{rad}$; 2) the magnitude of silicon isotope fractionation by radiolarians, and 3) whether there is evidence of a dissolution impact on values of $\delta^{30}Si_{rad}$.

## 2 Material and methods

### 2.1 Sample material

A total of 21 samples, comprising 14 plankton samples and 7 surface sediment samples, were collected at seven stations spanning the upper continental slope to the deep basin in the northwestern and southern SCS (Figure 1). All samples were collected during SCS cruises by R/V Shiyan 3 and Shiyan 1 of the South China Sea Institute of Oceanology, Chinese Academy of Sciences, in September 2016, November 2020 and August 2021 (Table S1 in the Supplementary Material 1). Plankton samples were collected from the 0-100 m and 100-300 m water layers at each station using a Hydro-Bios MultiNet with an aperture area of approximately 0.25 m$^2$ and 63 μm mesh size, and were preserved in 5% buffered formalin. Each plankton sample was then divided into two or four aliquots, washed through a 63 μm sieve with deionized water, reacted with 30% H$_2$O$_2$ in a water bath at 75°C for 24 hours, and freeze-dried. Due to insufficient radiolarian tests, only one plankton sample from either the 0-100 m or 100-300 m layer was used for isotopic analysis at each station. Surface sediments were sampled from the top 1 cm of the seabed using a box corer and stored in the cold storage at 5°C. No discernible evidence of bioturbation or disturbance from turbidity currents was observed in the sediment samples at these stations. Based on regional variability in Holocene sedimentation rates in the SCS (Sheng et al., 2024), the estimated age of the uppermost 1cm of sediment used in this study ranges from <40 to 133 years.

### 2.2 Radiolarian composition and preservation analysis

To determine the species composition of radiolarians, in both plankton samples and surface sediments, all samples were wet-sieved through a 63 μm sieve and prepared into radiolarian slides following the method described by Zhang et al. (2014). Briefly, samples were treated with a sufficient volume of 5% HCl solution for 15 minutes to eliminate calcareous organisms. Subsequently, the residual was processed using a sonic oscillator for one minute, and subjected to differential settling to remove impurities potentially adhering to the radiolarian tests. Following these procedures, all residual material was strewn nearly

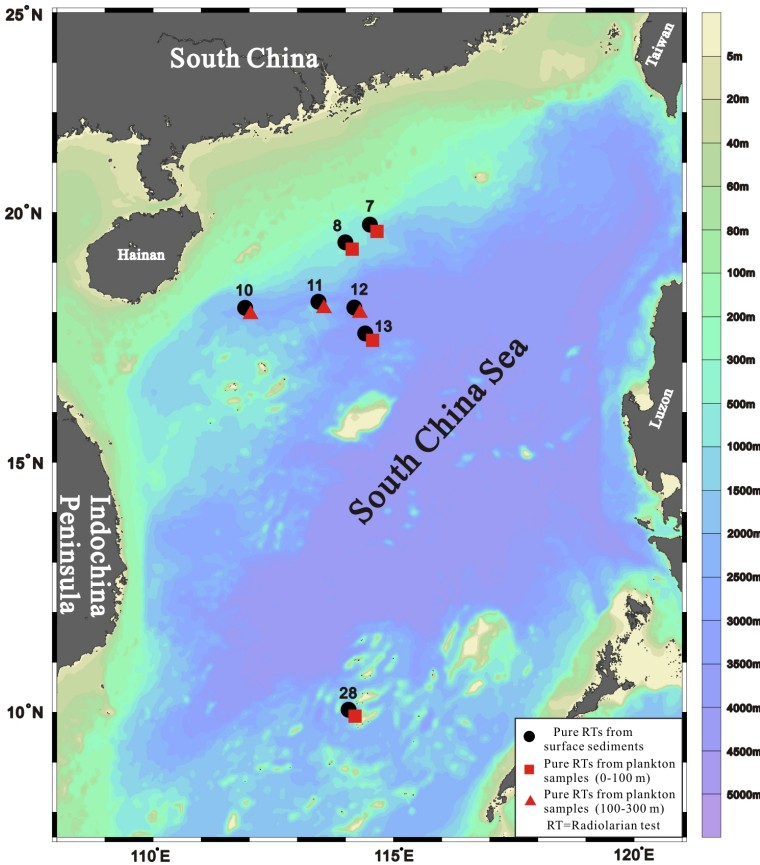

**Figure 1.** Locations of sampling stations in the South China Sea (base map created using Ocean Data View software, version 5.3.0).

homogenously onto microscope slides and permanently mounted with Canada Balsam. Radiolarian species were then identified and counted under a Nikon optical microscope at x100 or x200 magnification, with more than 500 specimens identified on slides using the publications of Chen and Tan (1996) and Tan and Chen (1999). Radiolarian diversity was determined by the species richness in each sample. The total number of radiolarian individuals in each sample was estimated from the count data as follows:

$$T = A * \frac{Vt}{V} * S * N \tag{1}$$

Where T is the total number of radiolarian individuals; A is the number of radiolarian shells counted from V fields of view on the slide; Vt is the total number of view fields on the radiolarian slide; V is the number of view fields examined under the microscope for the radiolarian individual count; S is the number of radiolarian slides; and N is the aliquot size of the sample (eight for the plankton sample, and one for the sediment sample in this study). Relative abundances of various species were 100 then calculated based on individual counts of each species and the total number of radiolarian specimens observed under the microscope.

To assess the preservation of radiolarian shells both within the water column and sediments, the proportion of fractured radiolarian shells was quantified in each studied sample under a Nikon optical microscope at x100 magnification. Further scanning electron microscopy (SEM) was employed to examine the potential for dissolution-induced alteration of the radiolarian skeleton.

## 2.3 Silicon isotope analysis

For isotope analysis, a minimum of 1 mg of radiolarian tests were extracted and purified from 7 plankton samples and 7 surface sediment samples following the method of Zhang and Swann (2023). Overall, the procedure for extracting and purifying radiolarian tests from marine sediments in Zhang and Swann (2023) comprises four stages: chemical treatment, initial sieving and differential settling, subsequent sieving, and finally density separation. In the first stage, raw samples were treated with 30% $H_2O_2$ and 15% HCl to remove the organic matters and calcareous components, and to facilitate particle dispersal. Following chemical treatment, the particles were rinsed and filtered using a 53 μm sieve to remove fine detritus, small diatoms, seaweed spines, and some sponge spicules. The filtered particles then underwent differential settling two to three times, followed by sonication, to further isolate radiolarians from large diatoms. Subsequently, particles were filtered three times using a 53 μm sieve by half immersing the sieve in a container filled with distilled deionized water (DDW) and gently tapping the base of the sieve for 10-15 minutes. This process aimed to remove monoaxonic spicules and a portion of the small non-monoaxonic spicules. Finally, the retained fraction was further refined by density separation, using specific gravities from 2.1-2.0 $g/cm^3$ at 0.05-unit interval, to remove any remaining coarse detritus and non-monoaxonic sponge spicules. The purity of extracted radiolarian tests in each sample was visually assessed under a Zeiss inverted light microscope at x100 magnification, with selected samples further examined using a SEM (JEOL JSM-IT200) equipped with an energy dispersive X-ray spectroscopy (EDS) detector at the Nanoscale and Microscale Research Centre, University of Nottingham.

Pure radiolarian samples were digested using NaOH fusion, with 1-1.5 mg radiolarian tests fused with a 200 mg NaOH pellet. Subsequent purification was achieved via ion exchange chromatography at a pH of 2-8 to ensure complete removal of cations, such as magnesium (Mg) and/or sodium (Na) (Georg et al., 2006; van den Boorn et al., 2006). Prior to analysis, each sample was filtered through a Millipore PTFE 0.2 nm filter to remove any particulate matter. The samples were then acidified using hydrochloric acid (HCl, to a concentration of 0.05 M, using twice quartz-distilled acid) and sulfuric acid ($H_2SO_4$, to a concentration of 0.001 M using Romil UPA). Swamping the natural abundance of chloride ($Cl^-$) and sulfate ($SO_4^{2-}$) ions, in both samples and reference materials alike, will induce a similar mass bias response in each (Hughes et al., 2011). Finally, all samples are spiked with approximately 300 ppb magnesium (Mg, Alfa Aesar SpectraPure, $^{24}Mg/^{25}Mg$ = 0.126633) to enable correction of the data for instrument-induced mass bias (Cardinal et al., 2003). Any deviation of the measured $^{24}Mg/^{25}Mg$ value from the known value is attributed to the effects of mass bias. The isotopes of Si are assumed to be similarly affected, and consequently, an exponential drift correction is applied (Cardinal et al., 2003).

Silicon isotope analysis was performed in dry plasma mode using the high mass-resolution capability of a Thermo Fisher Scientific Neptune Plus MC-ICP-MS (multi collector inductively coupled plasma mass spectrometer) at the Geochronology and Tracers Facility, British Geological Survey. Samples were typically prepared to yield a Si concentration of approximately 2

ppm and introduced to the MC-ICP-MS via an Aridus de-solvating unit, incorporating a PFA nebulizer with an uptake rate of 50 μL/min. In high-resolution mode, the instrument typically exhibited a mass-resolution between approximately 9,000 to 10,000, and a sensitivity of 4-5 V/ppm. Analytical replicates of the standard sample bracketing procedure were conducted where sample volume allowed, with repeated analysis of the standard (diatomite) to validate the data and sample bracketing with the international Si standard NBS28 to correct for any instrumental drift (Panizzo et al., 2016). The silicon isotope results are present in $\delta$-notation in per mille (‰) deviations relative to the standard NBS28 ($\delta^{30}Si = [(^{30}Si/^{28}Si)_{sample}/(^{30}Si/^{28}Si)_{NBS28}-1] \times 1000$), using within-run aliquots of NBS28. All uncertainties are reported at 2 standard deviations (2 SD) and are propagated to incorporate an excess variance derived from the diatomite validation material, which was quadratically added to the analytical uncertainty of each measurement. The reproducibility and instrument accuracy were assessed by analyzing the secondary reference material (Diatomite). Repeated measurements of diatomite yielded average $\delta^{30}Si$ values of 1.26 ± 0.14‰ (2 SD, n = 16), which agree well with the values obtained from the inter-laboratory comparison experiment (consensus value of 1.26‰ ± 0.2‰, 2 SD) (Reynolds et al., 2006). $\delta^{29}Si$ and $\delta^{30}Si$ values of radiolarian tests fall on the expected mass-dependent fractionation line $\delta^{29}Si = 0.51 \times \delta^{30}Si$ (Reynolds et al., 2006) (Figure S1 in the Supplementary Material 1), indicating the effective removal of all polyatomic interferences during measurement.

## 2.4   Paired t-tests and Wilcoxon signed-rank tests

To assess potential differences in radiolarian species composition and $\delta^{30}Si_{rad}$ values between water and surface sediment samples at each sampling station, paired t-tests and Wilcoxon signed-rank tests were employed. These statistical analyses were conducted using IBM SPSS Statistics Version 28, with the selection of the appropriate test determined by whether the data for relative abundances of prominent radiolarian species and $\delta^{30}Si_{rad}$ values conformed to a normal distribution.

## 3   Results

More than 400 radiolarian species have been reported in the SCS (Chen and Tan, 1996; Chen et al., 2008a), with 67-134 radiolarian species (mean = 114) identified in individual samples in this current study. The radiolarian diversity and the number of their shells in the plankton samples collected from the 0-100 m water layer, ranged from 67 to 116 species and from 28,025 to 102,443 individuals, respectively (Figure 2). These values represent 70-93% (mean = 83%) and 52-86% (mean = 72%) of the total observed in the upper 300 m of the water column (Figure 2). To characterize the radiolarian composition of each sample, species with a relative abundance exceeding 2% were defined as prominent species. Based on this, 29 prominent species belonging to Polycystine radiolarians were identified (Figure 3), most of which are typical tropical-subtropical species dwelling in warm surface to subsurface waters at low latitudes (Chen and Tan, 1996; Boltovskoy et al., 2010). The total abundance of prominent radiolarian species constitutes 62-80% (mean=68%) of the radiolarian community in the studied samples (Figure 3 and radiolarian count data in Supplementary Material 2). Wilcoxon test analysis for relative abundances indicated no statistically significant differences in the radiolarian community between plankton and sediment samples at each sampling station ($p$ = 0.19-0.98) (Table S2 in Supplementary Material 1).

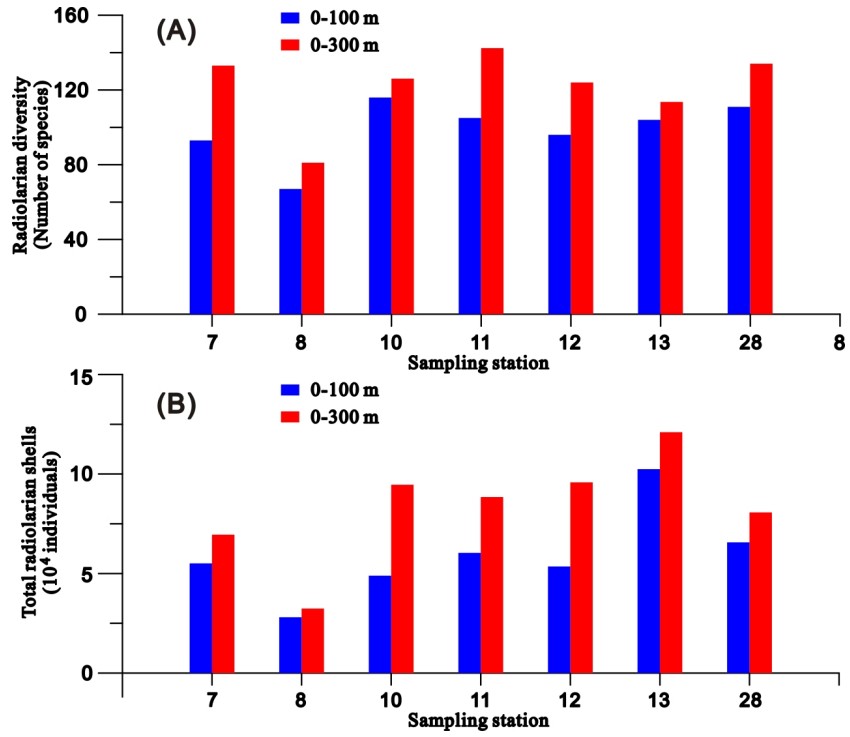

**Figure 2.** Radiolarian diversity (A) and shell numbers (B) in the upper 0-100 m and 0-300 m water columns at each sampling station.

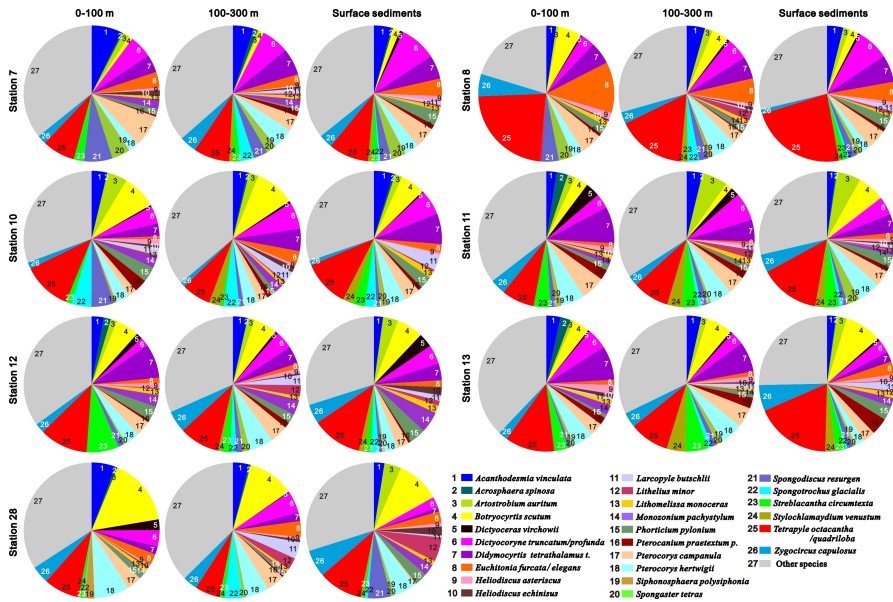

**Figure 3.** The prominent species and their relative abundances in plankton samples and surface sediments at each sampling station.

The proportion of fractured shells was low in the studied samples (Figure 4), generally ranging from 2% to 5% (mean = 3%) in plankton samples and from 3% to 9% (mean = 6%) in surface sediments. SEM images reveal a typical morphology of the pores on the radiolarian shell (Figure 4E), along with a high degree of integrity in the delicate skeletal structures (Figure 4F). These observations suggest good preservation of radiolarian shells, with no significant dissolution evident in either the water column or sediment samples.

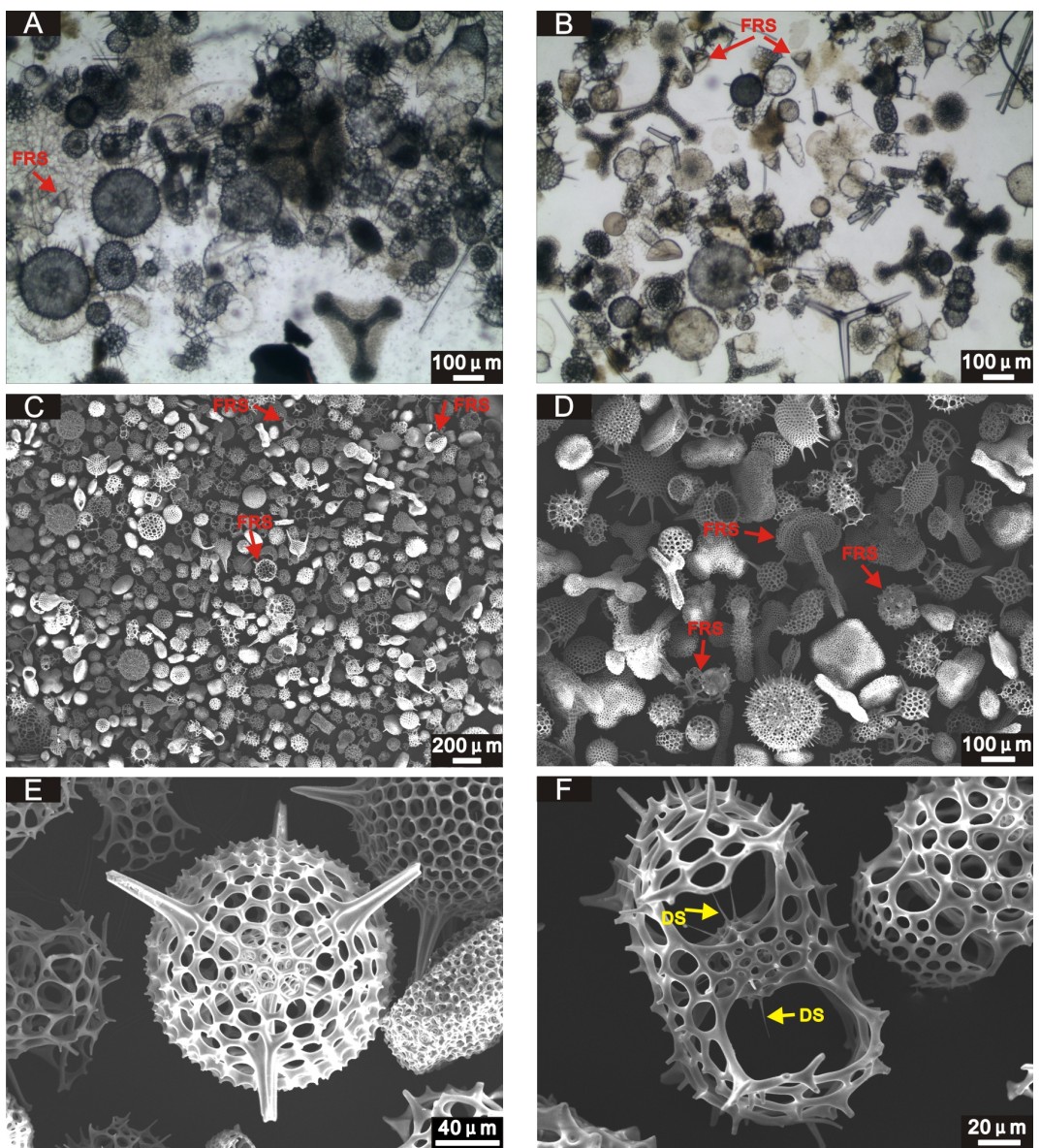

**Figure 4.** Observations of radiolarian shell preservation at station 12 (water depth: 3,497 m) in the plankton sample (100-300 m) (A and C) and surface sediments (B, D, E, and F) using optical microscope and the SEM. FRS=Fractured radiolarian shells; DS=Delicate skeletons.

Visual inspection, by light microscopy and SEM, indicated a high degree of purity (>99%) and cleanliness in the radiolarian tests used for silicon isotope analysis with negligible contamination from clay minerals or fine detrital particles on their shells (Figure 5). EDS analysis further showed that Si and O were the primary constituents of pure radiolarian test samples, with their combined concentration exceeding 99%. Other elements (Fe, Al, Mg, Sr, Na, K and Cl) flagged in red within the EDS spectrum images (Figure 5) indicate that either their concentrations were below the minimum detectable limit ( 0.1%) for the EDS detector, or the software lacked a 99% statistical confidence in the presence of these elemental signatures within the analysed sample.

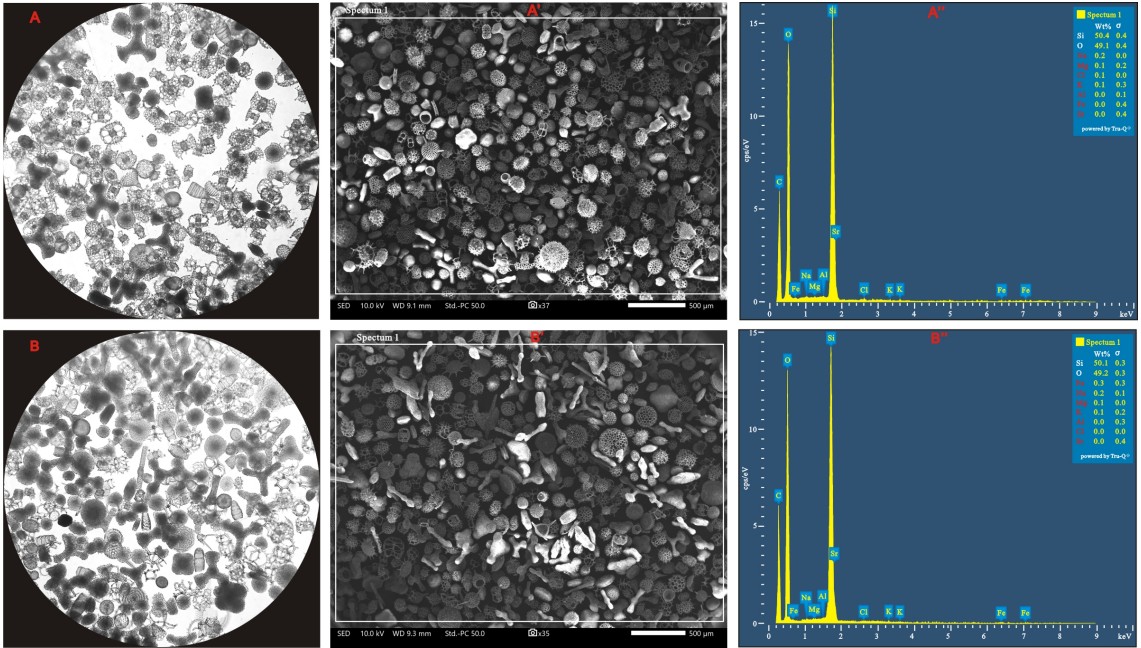

**Figure 5.** Pure radiolarian tests extracted from the plankton sample at station 8 (A) and the surface sediment at station 28 (B) under an inverted microscope at x100 magnification, as well as their SEM (A' and B') and EDS (A" and B") spectrum images.

$\delta^{30}Si_{rad}$ values ranged from 1.68‰ ± 0.20 to 1.81‰ ± 0.20 (mean = 1.76 ± 0.20‰) between 0-100 m of the water column, from 1.56‰ ± 0.20 to 1.83‰ ± 0.20 (mean = 1.72 ± 0.20‰) between 100-300 m, and from 1.61‰ ± 0.10 to 1.85‰ ± 0.21 (mean = 1.73 ± 0.17‰) in surface sediments (Figure 6A and Table S1 in Supplementary Material 1). A paired t-test indicates no significant difference between the plankton and surface sediment $\delta^{30}Si_{rad}$ values at each station ($p = 0.75$) (Figure 6B).

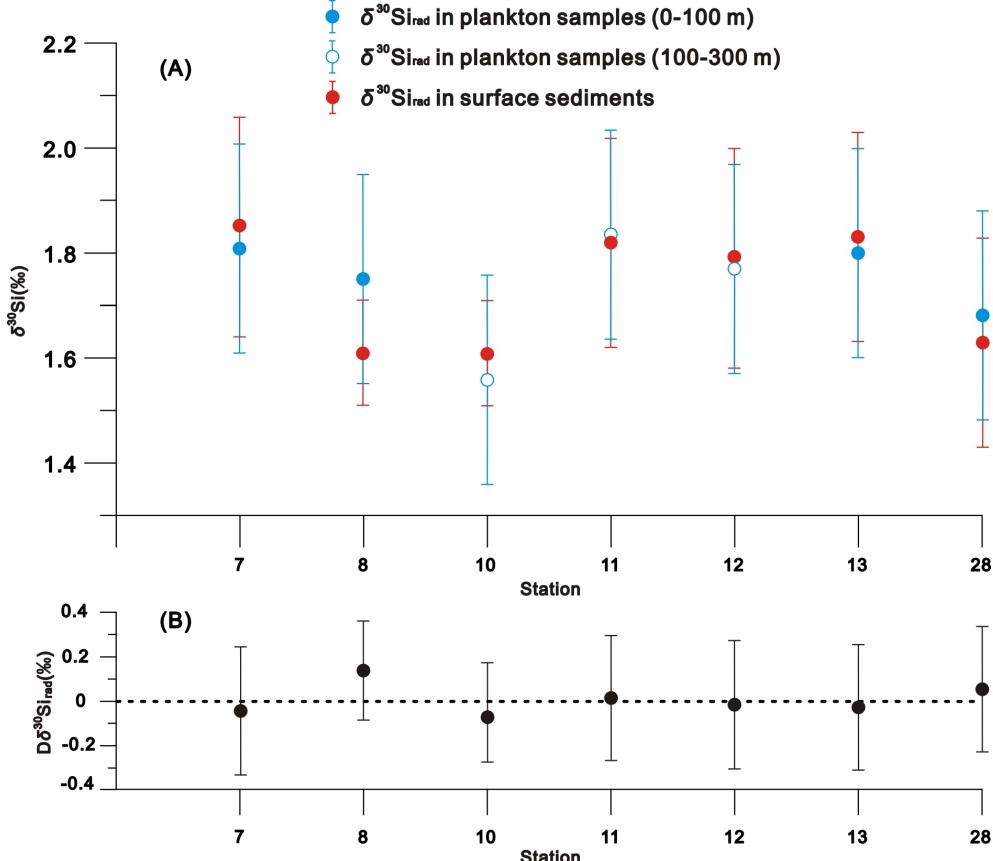

**Figure 6.** Comparison (A) and differences ($D\delta^{30}Si_{rad}$) (B) of $\delta^{30}Si_{rad}$ values between plankton samples and surface sediments at each sampling station. The vertical bar denotes the analytical uncertainty ($2\sigma$) of $\delta^{30}Si_{rad}$ compositions in (A), and the combined uncertainty ($2\sigma$) of $D\delta^{30}Si_{rad}$ in (B). The horizontal dashed line in (B) represents the zero line, indicating no differences in $\delta^{30}Si_{rad}$ compositions between plankton samples and surface sediments.

## 4  Discussions

### 4.1  Contributors to radiolarian $\delta^{30}Si$ signatures in water column plankton samples

$\delta^{30}Si_{rad}$ values (1.56 to 1.85‰) of water column and surface sediment samples in this study fall within the range of published data for sediment cores across the globe (-1.87 to 2.00 ‰; Hendry et al., 2014; Abelmann et al., 2015; Fontorbe et al., 2016, 2017, 2020), and are approximately consistent with $\delta^{30}Si_{rad}$ values of *Dictyocoryne profunda/truncatum* (1.42 to 1.74 ‰; Doering et al., 2021), a species of Spumellaria radiolarians dwelling between 50–100 m along the Peruvian coast (eastern East Pacific). However, they are higher than reported $\delta^{30}Si$ compositions of mixed radiolarians in surface sediments from the mid-Pacific (-0.2 to 0.3 ‰, Ding et al., 1996), the Southern Ocean (-0.74 to 1.33‰; Abelmann et al., 2015), and off the Peruvian coast (0.86 to 1.22‰, Doering et al., 2021).

In the oceans, radiolarians are distributed throughout the water column from surface to bottom waters (e.g., Kling, 1979; Abelmann and Gowing, 1996; Zhang et al., 2009; Hu et al., 2015; Boltovskoy, 2017). In this study from the SCS, the radio-
larians extracted from plankton samples and surface sediments for isotopic analysis comprise a mix of species from the bulk community, theoretically representing the average $\delta^{30}Si_{rad}$ signature across different water layers. Indeed, radiolarians in the SCS inhabit water depths from the surface to c. 2000 m (Hu et al., 2015). However, we suggest that the $\delta^{30}Si_{rad}$ signal in our samples primarily originates from radiolarians dwelling in surface to subsurface water (0-100 m). Firstly, the diversity and abundance of radiolarians in the SCS generally peak between 25 and 75 m water depth, declining significantly below
100 m (Zhang et al., 2009; Hu et al., 2015). This is consistent with global observations, which show that standing stock and species richness of polycystine radiolarians typically reach their highest values at 0-100 m in tropical and subtropical oceans (Boltovskoy, 2017). Secondly, radiolarians living above 100 m in the SCS account for a mean of 72% of the radiolarian shells in the plankton $\delta^{30}Si_{rad}$ samples analyzed in this current study (Figure 2), aligning with the observation that living radiolarians above 150 m contribute c. 77% of all the radiolarian shells from the surface to a depth of 2,000 m (Hu et al., 2015).

Given the differences in both the abundance and dominant species composition of living radiolarians between 0-75 m and the 75-300 m water depth in the SCS (Hu et al., 2015), it is expected that $\delta^{30}Si_{rad}$ values from plankton samples at 0-100 m water depth might potentially differ from those at 100-300 m. However, no significant difference was detected ($p = 0.52$). We suggest that the similarity of $\delta^{30}Si_{rad}$ values from samples at 0-100 m and 100-300 m water depth may be attributed to the fact that radiolarians at 100-300 m water depth primarily consist of dead individuals settling from the upper 100 m of the
water column. This is supported by the observation that: 1) the radiolarian species present in this study at 100-300 m water depth are normally found within the 0-100 m layer; and 2) no statistically significant differences were found between the prominent radiolarian species in these two water layers at each station (Wilcoxon signed-rank tests: $p = 0.32$ to 0.98) (Table S2 in Supplementary Material 1).

Wilcoxon signed-rank tests also show no statistical difference between the relative abundances of prominent radiolarian
species in plankton water column samples and surface sediment samples at each station ($p = 0.19$ to 0.98, Table S2 in Supplementary Material 1). This is in agreement with previous studies demonstrating that the radiolarian thanatocoenose in surface sediments generally reflect the major species of living radiolarian assemblage in the overlying water column in the SCS (Hu et al., 2014) and other oceans (Takahashi, 1982; Itaki, 2003; Itaki et al., 2003). Although discernible differences in the relative abundances of some prominent radiolarian species are observed, these likely result from the varying temporal coverage of each
sample. The SCS, influenced by the East Asian Monsoon, exhibits seasonal and interannual variability in its hydrological environment (Ning et al., 2009; Palacz et al., 2011) and radiolarian community (Wang et al., 2000; Zhang et al., 2020). Plankton samples were collected during a single snapshot in time, capturing radiolarians in a particular season, whereas radiolarians in the sediment samples represent an average record accumulated over multiple seasons and years. Although variations in the $\delta^{30}Si$ composition have been documented among different radiolarian taxa (Doering et al., 2021), the relative abundance
differences in certain prominent species, such as *Botryocyrtis scutum*, *Didymocyrtis tetrathalamus t.*, *Tetrapyle octacantha/quadribola*, and *Zygocircus capulosus* between plankton and surface sediment samples (Figure 3) do not result in significant $\delta^{30}Si_{rad}$ disparities in our study in the SCS. This may be attributed to two factors: 1) the minor differences in the relative

abundance of most prominent taxa between water column and sediment samples (Wilcoxon signed-rank tests: $p = 0.19$ to $0.98$, Table S2 in Supplementary Material 1); and 2) the high diversity of radiolarians in each sample averaging out the $\delta^{30}\mathrm{Si_{rad}}$ signal across different taxa, mitigating potential bias from individual taxa distorting the measured isotopic value.

Overall, we conclude that $\delta^{30}\mathrm{Si_{rad}}$ signatures in the water column, and subsequently incorporated in surface sediments, are primarily influenced by radiolarians from the 0-100 m water depth layer (surface to subsurface water), due to the substantial contribution of radiolarians in this depth range to the bulk radiolarian community in the SCS water column and sediment record.

## 4.2  Radiolaria silicon isotope fractionation

The isotope fractionation factor between radiolaria and seawater is a critical parameter for reconstructing past silicon cycle in the mid-upper ocean using $\delta^{30}\mathrm{Si_{rad}}$ in downcore records (Hendry et al., 2014; Abelmann et al., 2015; Doering et al., 2021). As discussed previously, the $\delta^{30}\mathrm{Si_{rad}}$ signatures in the water column of the SCS are predominantly contributed to by radiolarians from the 0-100 m depth layer. Therefore, seawater $\delta^{30}\mathrm{Si}$ ($\delta^{30}\mathrm{Si_{sw}}$) data pertinent to constraining silicon isotope fractionation by radiolarians in the SCS should be derived primarily from this depth interval. Currently, there is no corresponding $\delta^{30}\mathrm{Si_{sw}}$ data from the stations in the SCS where water column radiolaria samples were collected. Previous work, however, has measured seasonal changes in $\delta^{30}\mathrm{Si_{sw}}$ in the northern SCS (Cao et al., 2012). At the two sites from that study situated off the continental shelf (A10 and SEATS), $\delta^{30}\mathrm{Si_{sw}}$ above 100 m varies from 1.4‰ to 2.9‰. Assuming: 1) these values are broadly comparable to stations where $\delta^{30}\mathrm{Si_{rad}}$ was measured; and 2) that radiolarians construct their skeletons in equilibrium with ambient seawater (Abelmann et al., 2015; Fontorbe et al., 2016; Doering et al., 2021), the isotope fractionation factor ($\epsilon$) by radiolarians can be calculated as:

$$\epsilon \sim \Delta^{30}\mathrm{Si} = \delta^{30}\mathrm{Si_{rad}} - \delta^{30}\mathrm{Si_{sw}} \tag{2}$$

Using Monte Carlo simulations (10,000 replicates) to account for the 2SD uncertainty of both $\delta^{30}\mathrm{Si_{rad}}$ and $\delta^{30}\mathrm{Si_{sw}}$, $\epsilon$ is estimated to range from –0.33‰ to –0.92‰ (mean = 0.58‰) (Table S1 in Supplementary Material 1). Due to the difficulties in culturing radiolaria (e.g., Suzuki and Not, 2015), relatively little is known about the fractionation factor for $\delta^{30}\mathrm{Si_{rad}}$. Values for $\epsilon$ obtained here from the SCS are comparable to those measured by Abelmann et al. (2015) in the Southern Ocean (–0.54‰ to –0.91‰, mean = –0.75‰) and by Doering et al. (2021) along the Peruvian Shelf (–0.35‰ to –0.79‰, mean = –0.62‰, mixed radiolaria). The large range of values for $\epsilon$ has been suggested to indicate species/order-specific fractionation during the uptake of silicic acid by radiolaria (Doering et al., 2021). Such a process would necessitate the extraction of species-specific radiolarian samples for isotope analysis in future palaeoceanographic research (e.g., Zhang and Swann, 2023). However in this current study from the SCS, whilst the species compositions of the $\delta^{30}\mathrm{Si_{rad}}$ samples is known (Figure 3), an absence of matching $\delta^{30}\mathrm{Si_{sw}}$ data from exactly the same stations prevents this issue being further investigated at this time.

### 4.3 Transfer of radiolarian $\delta^{30}$Si signatures into the sediment record

At each sampling station, $\delta^{30}\text{Si}_{\text{rad}}$ compositions (mean = 1.73‰) in the surface sediment closely resembles those (mean = 1.74‰) in the overlying water column evidenced by the paired t-test (p=0.75) (Figure 6), indicating a faithful transfer of the $\delta^{30}$Si signal incorporated into radiolarian skeletons from the water column to sediments. This suggests that dissolution has a minimal impact on the $\delta^{30}\text{Si}_{\text{rad}}$ signatures as radiolarians shells sink through the water column and become incorporated into the sediment record. One of two possibilities may account for this observation: 1) the radiolarian shells may not have undergone substantial dissolution during sinking; 2) the radiolarian shells have experienced substantial dissolution, but this process may not significantly alter their isotope composition. As shell fracture of microfossils preserved within sediments is commonly attributed to partial dissolution (e.g., Murray and Alve, 1999; Ryves et al., 2001), the proportion of fractured radiolarian shells may indicate the potential for radiolarian shell dissolution during sinking. In this study, fractured radiolarian shells comprised only a mean of 3% in plankton samples and 6% in surface sediment (Figure 4). Considering the low proportion of fractured radiolarian shells, the well-preserved state of radiolarian shells (Figure 4), and the minor differences in the relative abundances of prominent radiolarian species between plankton samples and surface sediments at each sampling station (as detailed in Section 4.1), we propose that radiolarian shells have not experienced substantial dissolution or remineralisation during their transfer from the water column to the sediments.

The susceptibility of radiolarian shells to dissolution vary considerably with different taxonomic groups due to the differences in the chemical constituent of their skeletons. It is generally accepted that radiolaria comprise three classes: Polycystinea, Acantharea, and Taxopodida (Adl et al., 2019), despite debate concerning radiolarian classification schemes (Chen and Tan, 1996; Suzuki and Not, 2015; Adl et al., 2019; Biard, 2022). Acantharians are composed of celestite $\text{SrSO}_4$, and their skeletons are highly susceptible to dissolution due to the significant undersaturation of seawater with respect to this mineral (Bernstein et al., 1992; De Deckker, 2004; Shimmen et al., 2009). However, their absence of a silicon based skeletons renders their dissolution inconsequential for $\delta^{30}\text{Si}_{\text{rad}}$ measurements in the water column and sediment. Taxopodida, though composed of silica, are generally rare in the water column, and no fossil records of this Class have been discovered in surface sediments of the world ocean to date (Biard, 2022). In our studied samples from the SCS, both Acantharea and Taxopodida taxa are trace components (<0.1%) in plankton samples and absent in surface sediments.

In the oceans, particularly in the SCS, polycystine radiolarians are the predominant group of the radiolarian community in both the water column and in sediments (Chen and Tan, 1996; Chen et al., 2008a, b; Suzuki and Aita, 2011) and are comprised of three Orders: Spumellaria, Nassellaria, and Collodaria. Collodaria radiolaria, with skeletons composed of both opal and Celestine (Afanasieva et al., 2005; Afanasieva and Amon, 2014), are generally susceptible to dissolution in seawater and poorly preserved in sediment samples. However, Collodaria radiolaria are less common in the oceans (Probert et al., 2014; Zhang et al., 2018) and generally represent a low abundance in both the water column and sediments (Suzuki and Not, 2015; Biard, 2022). In the SCS, the abundance of Collodaria radiolarians in the water column ranges from 2-270 $\text{individuals}/\text{m}^3$ (mean = 30 $\text{individuals}/\text{m}^3$) (Zhang et al., 2020; Cheng et al., 2023), while the mean abundance of Spumellaria and Nassellaria in the upper water exceeds 3000 $\text{individuals}/\text{m}^3$ (Zhang et al., 2009; Hu et al., 2015). Spumellaria and Nassellaria taxa, which have

been widely used as proxies for paleoenvironmental reconstruction (e.g., Itaki et al., 2004; Abelmann and Nimmergut, 2005; Zhang et al., 2014; Cortese and Prebble, 2015), are characterized by a greater resistance to dissolution and are generally well preserved in sediments (Takahashi, 1982; Morley et al., 2014). In this study Collodaria taxa have very low relative abundance in the analysed samples (mean < 1%), which are instead dominated by the more dissolution-resistant Spumellaria and Nassellaria taxa (Figure 3). Since Spumellaria and Nassellaria radiolarians, the primary constituent of the radiolarian community in the SCS, are characterized by a great resistance to dissolution, dissolution is expected to have limited impacts on these radiolarian shells, and thus on $\delta^{30}Si_{rad}$ compositions during sinking and burial.

As discussed above, we suggest that the effect of dissolution on the $\delta^{30}Si_{rad}$ composition in the SCS is negligible during the transfer from the water column to the surface sediment, based on the comparable $\delta^{30}Si_{rad}$ compositions observed in plankton samples and surface sediment at each sampling station. This is likely attributed to the dominance of Spumellaria and Nassellaria taxa in the radiolarian community, which generally exhibit a high resistance to dissolution, as well as the scarcity or absence of the readily dissolvable Taxopodida, Acantharea, and Collodaria taxa.

# 5 Conclusions

To investigate the potential effects of dissolution on $\delta^{30}Si_{rad}$ signatures during the sinking and burial processes from the water column to sediments, we conducted a comparative analysis of radiolarian community and $\delta^{30}Si_{rad}$ compositions between plankton samples and surface sediments from seven stations in the SCS. The key findings are summarized as follows:

1. $\delta^{30}Si_{rad}$ compositions ranged from 1.56-1.83‰ (mean = 1.74‰) in the plankton samples and from 1.61-1.85‰ (mean = 1.73‰) in surface sediments, with the fractionation factor for $\delta^{30}Si_{rad}$ varying from –0.33‰ to –0.92‰ (mean = 0.58‰). No statistically significant differences in $\delta^{30}Si_{rad}$ values were observed between the water column and surface sediment samples at each sampling station.

2. Minor differences in the relative abundance of prominent taxa between the water column and sediments are not statistically significant. As most radiolarians in the water column originate from depths above 100 m in this study, we suggest that the $\delta^{30}Si_{rad}$ signature preserved in surface sediments is primarily contributed from radiolarians within this depth range in the SCS.

3. The similar $\delta^{30}Si_{rad}$ compositions in plankton samples and surface sediment at each station suggest a faithful transfer of $\delta^{30}Si_{rad}$ signatures from the water column to the surface sediment in the SCS. This is aided by the dominance of the more dissolution-resistant Spumellaria and Nassellaria taxa (>99%) within the radiolarian community, coupled with the scarcity or absence of the readily dissolvable taxa (Taxopodida, Acantharea, and Collodaria).

*Data availability.* The $\delta^{30}Si_{rad}$ data and original radiolarian count data from this paper can be found in Supplementary Material 1 and 2, respectively.

*Author contributions.* QZ and GEAS conceived and designed the study. QZ analysed the radiolarian assemblages and prepared the samples for isotope analysis, supported by GEAS. Isotope samples were prepared and analysed by VP and QZ with additional support from MSAH. All authors contributed to the writing of the paper and commented on drafts.

*Competing interests.* The authors declare that they have no conflict of interest.

*Acknowledgements.* We would like to thank Dr. Weifen Hu for providing the raw radiolarian data from her publication (Hu et al., 2014). We thank the editor and two anonymous reviewers for their meticulous reviews and insightful comments, which significantly improved the manuscript. This work was supported by the NERC Isotope Geosciences Facilities Steering Committee (Grant No. 2230.0320), Guangdong Basic and Applied Basic Research Foundation (Grant Nos. 2020A1515010499, 2022A1515010932), National Natural Science Foundation

of China (Grant Nos. 42076073), and the CAS Scholarship.

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
