# Peer review of "Faithful transfer of radiolarian silicon isotope signatures from water column to sediments in the South China Sea"

_EGUsphere, 2024_

## Author Comment (AC1)

We would like to thank the editor and the reviewer for the constructive comments and all the ideas suggested to our manuscript. These comments and suggestions are insightful and very helpful in improving the quality of our paper. We have read all the comments carefully and have made the necessary revisions to the manuscript. All changes to the text are indicated in **blue**. A detailed explanation for each issue raised is provided below:

**Reviewer 1**

**There is growing interest in how the silicon isotope composition (expressed as δ30Si) of radiolarians can be used to supplement/complement those of the more established proxy archives in diatoms and sponges. As with these two groups, there is a need to understand and account for any post-mortem alteration of the initial isotope signal. Here, Zhang et al compare bulk-assemblage radiolarian δ30Si values from water-column plankton tows and underlying coretop sediments. They demonstrate the two are statistically indistinguishable, lending more confidence to the use of radiolarian δ30Si as a window on to past Si cycling.**

**In general, this manuscript is very well written and the figures are clear (with possible exception of Fig.3). The referencing is generally appropriate – with some notable absences (see below). The data are generated by appropriate techniques (though some more details may be warranted). Overall, there is little to criticise in terms of the central conclusion – that the radiolarian δ30Si signal is not altered during water column sinking – which is supported by the data (though I have a series of minor comment/suggestions that I detail below). Nevertheless, this is a relatively small dataset and I have the impression that with a slightly expanded dataset much more could be done. I note in the supplement, Fig. S1 contains 22 'unpublished' radiolarian datapoints. By integrating these, and dissolved Si δ30Si data, a much more impactful paper would result. Some suggestions are below, but this is ultimately an editorial decision.**

**Response:** We appreciate the reviewer's insightful comments. Yes, we have included 22 'unpublished' radiolarian $\delta^{30}Si$ data in Fig. S1 of the Supplement Material. Given that the primary objective of the present study was to ascertain whether and why radiolarian silicon isotopes ($\delta^{30}Si_{rad}$) signatures are faithfully transferred from the water column to the sediments, combined $\delta^{30}Si_{rad}$ records from paired water column and surface sediment samples are required. However, the radiolarian tests used to obtain these 22 data were extracted from surface sediments, and lack corresponding $\delta^{30}Si_{rad}$ data available from the overlying water column (although plankton samples were collected from some of these 22 stations, the quantity of radiolarian tests was insufficient for isotope analysis). As these 22 data were measured within the same analytical batch as samples used in this study, we included them in the Supplement Material to demonstrate that $\delta^{29}Si$ and $\delta^{30}Si$ values of radiolarian tests fall on the expected mass-dependent fractionation line $\delta^{29}Si = 0.51 \times \delta^{30}Si$ (Reynolds et al., 2006) (Figure S1 in the Supplementary Material), thereby indicating the effective removal of all polyatomic interferences during measurement.

Regarding dissolved Si $\delta^{30}Si$ data and the constraint of the radiolaria Si isotope fractionation factor, we also collected water samples from the relevant depth range when collecting plankton samples and surface sediments at each sampling station in this study. Considering the relatively low $Si(OH)_4$ concentrations in these water samples from the upper water column, we plan to use a

Neoma MC-ICP-MS for the silicon isotopic analysis of dissolved silicon because of its enhanced sensitivity and mass resolution. However, isotopic analyses for seawater samples are pending due to ongoing instrument-related issues with the Neoma MC-ICP-MS at the Geochronology and Tracers Facility, British Geological Survey. When these data are obtained, we will prepare a separate manuscript addressing the radiolarian fractionation factor and its implication for nutrient levels in the mid-upper water column.

**Minor comments and suggestions**

**L16-27 – the prominence of the SALH in the introduction is a bit strange to me, considering it's not the focus of the manuscript (and isn't returned to)**

**Response:** In the introduction, we referenced the "silica acid leakage hypothesis" (SALH) due to its emphasis on the influence of changes in dissolved silicon (DSi) concentrations between the Southern Ocean and low-latitude regions on atmospheric $p$CO$_2$ and climate. While the SALH remains debated, partly due to inconsistent records of siliceous productivity and DSi concentrations in overlying waters across different low-latitude regions during the late Quaternary, we cite it here to highlight the importance of reconstructing past Si nutrient levels throughout the water column. This is expected to facilitate a more comprehensive understanding of the relationships between the Si cycle, biological pump efficiency, and global climate change.

**L48: Two papers that deserve citation/discussion here and elsewhere are Closset et al. 2015 (doi: 10.1002/2015GB005180) and Grasse et al. 2021 (doi: 10.3389/fmars.2021.697400) – both present a comparison of plankton tow diatoms and sediment trap (Closset) or core-top (Grasse) material, concluding that the transfer of biogenic silica from surface ocean to depth isn't associated with a resolvable change in δ30Si. See also Varela et al. 2004 (doi: 10.1029/2003GB002140; sediment trap data) and Fripiat et al. 2012 (doi: 10.5194/bg-9-2443-2012; water column biogenic silica data).**

**Response:** We appreciate the reviewer bringing these important references to our attention, and we have included them in the revised version. Indeed, the constancy of $\delta^{30}$Si of diatom biogenic silica ($\delta^{30}$Si$_{BSi}$) with depth has been documented for suspended particles in the Atlantic sector of the Southern Ocean (Fripiat et al., 2012), as well as in sediment traps in the Southern Ocean south of New Zealand (Varela et al., 2004) and Australian (Closset et al. 2015). Furthermore, a strong agreement has also been observed between $\delta^{30}$Si$_{BSi}$ from seawater samples and those from core-top sediments in the central upwelling region off Peru (Grasse et al. 2021). These findings suggest that the transfer of diatom biogenic silica from the surface ocean to sediments is not associated with a dissolution-driven alteration of $\delta^{30}$Si.

**References:**

Closset, I., Cardinal, D., Bray, S. G., Thil, F., Djouraev, I., Rigual−Hernández, A. S., Trull, T. W.: Seasonal variations, origin, and fate of settling diatoms in the Southern Ocean tracked by silicon isotope records in deep sediment traps. Global Biogeochemical Cycles, 29(9), 1495-1510. https://doi.org/10.1002/2015GB005180, 2015.

Fripiat, F., Cavagna, A. J., Dehairs, F., De Brauwere, A., André, L., Cardinal, D.: Processes controlling the Si-isotopic composition in the Southern Ocean and application for paleoceanography, Biogeosciences, 9(7), 2443-2457, https://doi.org/10.5194/bg-9-2443-2012, 2012.

Grasse, P., Haynert, K., Doering, K., Geilert, S., Jones, J. L., Brzezinski, M. A., Frank, M.: Controls on the silicon isotope composition of diatoms in the peruvian upwelling. Frontiers in Marine Science, 8, 697400, https://doi.org/10.3389/fmars.2021.697400, 2021.

Varela, D. E., Pride, C. J., Brzezinski, M. A.: Biological fractionation of silicon isotopes in Southern Ocean surface waters. Global biogeochemical cycles, 18(1). https://doi.org/10.1029/2003GB002140, 2004.

**L85 and introduction: In general, there is a growing awareness that 'bulk' assemblage $\delta^{30}$Si data have disadvantages as a paleo-archive, and that where possible single-species records are much stronger. Therefore it would be good to see some justification for why this was not attempted here.**

**Response:** We thank the reviewer for raising this important point.. We agree with the reviewer that $\delta^{30}$Si records derived from single-species shells are generally preferred for paleo-archive studies, as the potential for different vital effects on $\delta^{30}$Si fractionation may exist between various radiolarian species (e.g. Doering et al., 2021). However, obtaining sufficient material for silicon isotopic analysis from individual radiolarian taxa in our samples proved to be a significant challenge, necessitating our focus on the bulk radiolarian species record for this study.

We did, in fact, attempt to isolate single-species tests via manual picking under a microscope, but encountered several obstacles. Firstly, for silicon isotope analysis using a Thermo Fisher Scientific Neptune Plus MC-ICP-MS at the Geochronology and Tracers Facility, British Geological Survey, 1 to 1.5 mg of purified radiolarian tests is typically required. Radiolarian tests from sediments in the tropical Ocean are notably lightweight, averaging 0.063 to 0.136 mg/shell (Moore, 1969; Takahashi, 1982). Consequently, a mean of approximately 7,000 to 15,000 individual tests are required to achieve the ~1 mg of material needed for the silicon isotope analysis. Given the high diversity of radiolarians in Holocene sediments, particularly in low-latitude oceans (Moore, 1969; Takahashi, 1982; Boltovskoy et al., 2010) and in our samples, which average over 100 species per sample, coupled with the observation that most species typically constitute less than 10% of the total radiolarian community (Boltovskoy et al., 2010; Chen et al., 2008), this means that 70,000 to 150,000 bulk radiolarian tests are potentially required for each sample to obtain sufficient single-species material. Regrettably, the limited volume of the plankton samples made it difficult to obtain such a large number of radiolarian shells. Secondly, manually picking 7,000 to 15,000 individual tests out of 70,000 to 150,000 bulk radiolarian tests is highly time-consuming, rendering this approach impractical for the study of single-species radiolarian $\delta^{30}$Si. Furthermore, even with meticulous hand-picking under a microscope, the electro static adsorption of these light tests to transfer tools, such as brushes, inevitably results in test loss during transfer to storage vials. This further compounded the difficulty of accumulating the required quantity of monospecific tests for the isotope analysis.

It was these insurmountable challenges encountered during our attempts at hand-picking that prompted us to develop the method for extracting and purifying bulk radiolarian tests from the sediments for the study of the radiolarian $\delta^{30}$Si (Zhang and Swann, 2023). This method enabled us to effectively obtain sufficient pure radiolarian tests to conduct the comparative analysis of radiolaria $\delta^{30}$Si compositions using water column and surface sediment samples in this current study.

Based on the reviewer's comment, we have included a brief explanation in the revised manuscript regarding the absence of silicon isotopic analysis for individual radiolarian taxa:

"Given the lightweight nature of shells and the high diversity of radiolarians, particularly in low-latitude oceans (Moore, 1969; Takahashi, 1982; Boltovskoy et al., 2010), combined with the observation that most species typically comprise less than 10% of the total radiolarian assemblage (Boltovskoy et al., 2010; Chen et al., 2008), obtaining sufficient material from individual radiolarian taxa for silicon isotopic analysis remains a considerable challenge."

By the way, we have measured bulk radiolarian $\delta^{30}Si$ ($\delta^{30}Si_{rad}$) from a sediment core collected in the northern South China Sea (SCS), spanning a period from ~17 ka to the present. The results indicate that $\delta^{30}Si_{rad}$ values, and the calculated silicic acid utilisation efficiency, were generally lower during the last deglacial period compared to the Holocene. This suggests a potential increase in silicic acid concentrations in the mid-upper water column during the last deglacial, potentially stemming from a deepened mixed layer induced by an intensified winter monsoon in the northern SCS (e.g. Steinke et al., 2011), or from an enhanced influx of silica-rich Antarctic Intermediate Water (AAIW) into the northern SCS (e.g. Huang et al., 2014; Yang et al., 2017). Such conditions, characterized by increased silicic acid levels in the mid-upper water column, could account for the previously observed higher productivity in the northern SCS during the last glacial period (e.g. Lin et al., 1999; Higginson et al., 2003). This finding may provide compelling evidence for the effectiveness of bulk radiolarian $\delta^{30}Si$ as a proxy for reconstructing past changes in silicic acid levels or availability within the water column. A manuscript detailing these findings is currently in preparation.

References:

Boltovskoy, D., Kling, S. A., Takahashi, K., Bjørklund, K.: World atlas of distribution of recent polycystina (Radiolaria), Palaeontologia Electronica, 13, 1–229, 2010.

Chen M H, Zhang L L, Zhang L L , et al. 2008a. Preservation of radiolarian diversity and abundance in surface sediments of the South China Sea and its environmental implication. Journal of China University of Geosciences, 19: 217–229.

Doering, K., Ehlert, C., Pahnke, K., Frank, M., Schneider, R., Grasse, P.: Silicon isotope signatures of radiolaria reveal taxon-specific differences in isotope fractionation, Front. Mar. Sci., 8, 666896, https://doi.org/10.3389/fmars.2021.666896, 2021.

Higginson, M. J., Maxwell, J. R., Altabet, M. A.: Nitrogen isotope and chlorin paleoproductivity records from the Northern South China Sea: remote vs. local forcing of millennial-and orbital-scale variability, Marine Geology, 201(1-3), 223-250, https://doi.org/10.1016/S0025-3227(03)00218-4, 2003.

Huang, K. F., You, C. F., Chung, C. H., Lin, Y. H., Liu, Z.: Tracing the Nd isotope evolution of North Pacific Intermediate and Deep Waters through the last deglaciation from South China Sea sediments, Journal of Asian Earth Sciences, 79, 564-573, https://doi.org/10.1016/j.jseaes.2013.01.004, 2014.

Lin, H. L., Lai, C. T., Ting, H. C., Wang, L., Sarnthein, M., Hung, J. J.: Late Pleistocene nutrients and sea surface productivity in the South China Sea: A record of teleconnections with Northern Hemisphere events, Marine Geology, 156(1-4), 197-210, https://doi.org/10.1016/S0025-3227(98)00179-0, 1999.

Moore, JR. T. C: Radiolaria: change in skeletal weight and resistance to solution. Geological Society of America Bulletin, 80(10), 2103-2108. https://doi.org/10.1130/0016-7606(1969)80[2103:RCISWA]2.0.CO;2, 1969

Takahashi, K.: Vertical flux, ecology and dissolution of Radiolaria in tropical oceans: implications for the silica cycle. PhD thesis. Woods Hole Oceanographic Institution and Massachusetts Institute of Technology, 1982

Steinke, S., Glatz, C., Mohtadi, M., Groeneveld, J., Li, Q., Jian, Z.: Past dynamics of the East Asian monsoon: No inverse behaviour between the summer and winter monsoon during the Holocene, Global and Planetary Change, 78(3-4), 170-177, https://doi.org/10.1016/j.gloplacha.2011.06.006, 2011.

Yang, Y., Xiang, R., Liu, J., Fu, S., Zhou, L., Du, S., Lü, H.: Changes in intermediate water conditions in the northern South China Sea using Globorotalia inflata over the last 20 ka. Journal of Quaternary Science, 32(7), 1037-1048, https://doi.org/10.1002/jqs.2974, 2017.

Zhang, Q., Swann, G. E. A.: An effective method to extract and purify radiolaria from tropical marine sediments, Front. Mar. Sci., 10, 1150518, https://doi.org/10.3389/fmars.2023.1150518, 2023.

**Section 2.3: I would suggest more detail is needed here. Specific suggestions include:**

● **Define what is 'sufficient' radiolarian tests (L80; what is the typical mass of Si processed)**

**Response:** We are sorry for the lack of clarity regarding the typical mass of processed Si. For silicon isotope analysis using a Thermo Fisher Scientific Neptune Plus MC-ICP-MS at the Geochronology and Tracers Facility, British Geological Survey, 1 to 1.5 mg of purified radiolarian tests is typically required. Accordingly, it was expected that at least approximately 1 mg of purified radiolarian tests would be extracted and purified from the samples used in this study. In this study, we successful extracted over 1.5 mg of pure radiolarian tests for isotope analysis from all surface sediment samples. However, due to limited sample volumes, the quantity of pure radiolarian tests obtained from plankton samples was approximately 1 mg.

In the revised manuscript, we have substituted "sufficient radiolarian tests" with "approximate 1 to 1.5 mg of radiolarian tests".

● **Give a brief overview of Zhang and Swann (L81). Is there potential for larger diatoms or sponge spicules to 'contaminate' the sample?**

**Response:** Following the reviewer's suggestion, we have added a brief overview of Zhang and Swann (2023) in the revised manuscript as follows: Overall, the procedure for extracting and purifying radiolarian tests from marine sediments in Zhang and Swann (2023) comprises four stages: chemical treatment, initial sieving and differential settling, subsequent sieving, and finally density separation (Figure A). In the first stage, raw samples were treated with ~30% $H_2O_2$ and 15% HCl to remove the organic matters and calcareous components, and to facilitate particle dispersal. Following chemical treatment, the particles were rinsed and filtered using a 53 μm sieve to remove fine detritus, small diatoms, seaweed spines, and some sponge spicules. The filtered particles then underwent differential settling two to three times, followed by sonication, to further isolate radiolarians from large diatoms. Subsequently, particles were filtered three times using a 53 μm sieve by half immersing the sieve in a container filled with distilled deionized water (DDW) and gently tapping the base of the sieve for 10-15 minutes. This process aimed to remove all monoaxonic spicules and a portion of the small non-monoaxonic spicules. Finally, the retained fraction was further refined by density separation, using specific gravities from 2.1-2.0 $g/cm^3$ at 0.5-unit interval, to remove any remaining coarse detritus and non-monoaxonic sponge spicules.

Larger diatoms and sponge spicules can also be effectively separated from radiolarian tests following the method of Zhang and Swann (2023). The majority of larger diatoms can be removed through differential settling (Figure A). Residual diatoms can then be broken down by sonication treatment for no more than 10 minutes, to avoid the breakage of radiolarian tests. These fragmented diatoms can then be further removed through sieving or differential settling. Although most sponge spicules are greater than 53 μm in length, the diameter (cross section) of monoaxonic sponge spicules is generally several micrometers. Therefore, monoaxonic spicules may pass

through a 53 μm sieve during wet sieving (Figure A), provided they are repeatedly suspended and settle non-horizontally in the water during the sieving process. This can be achieved by half immersing samples in a container filled with DDW, and gently tapping the base of the sieve to maintain particle suspension. Non-monoaxonic sponge spicules can be further refined via density separation (Figure A), using specific gravities from 2.1-2.0 g/cm$^3$ at 0.5-unit interval. This is based on findings that the mean density of sponge spicules is higher than that of tropical radiolarians in late Quaternary sediments (Zhang and Swann, 2023).

[Figure]

Figure A the main process for separating and purifying the radiolarian tests from tropical marine sediments (Zhang and Swann, 2023). All photos taken under the inverted microscope at 100x magnification.

Microscopic examination of purified radiolarian tests, across several fields of views at x100 magnification using inverted light microscopy, revealed a negligible quantity of larger diatoms and sponge spicules (<1%) remaining with the purified radiolarian tests (as shown in Videos 1 and 2 in the supplementary material of Zhang and Swann (2023)). Consequently, the potential for larger diatoms or sponge spicules to contaminate purified radiolarian samples is considered minimal or negligible.

**References:**

Zhang, Q., Swann, G. E. A.: An effective method to extract and purify radiolaria from tropical marine sediments, Front. Mar. Sci., 10, 1150518, https://doi.org/10.3389/fmars.2023.1150518, 2023.

● **Confirm that Na (used in dissolution) was successfully removed by the ion exchange chromatography**

**Response:** Prior to isotopic analysis, all samples are purified using cation exchange chromatography. At a pH of 2 and 8, Si species are either neutral (Si(OH)$_4$) or anionic (H$_3$SiO$_4^-$) and will therefore pass freely through a cation exchange resin whilst all major cations, including Na+, remain trapped on the column.

In the revised manuscript, the statement "Subsequent purification was achieved via ion exchange chromatography at a pH of between 2-8 (Georg et al., 2006; van den Boorn et al., 2006)." has been revised to "Subsequent purification was achieved via ion exchange chromatography at a

pH of 2-8 to ensure complete removal of cations, such as magnesium (Mg) and/or sodium (Na) (Georg et al., 2006; van den Boorn et al., 2006)."

- **Give approximate mass resolution (m/Dm, L93)**

**Response:** In high-resolution (HR) mode, resolution (R) = $m/\Delta m$, where $\Delta m$ is derived from the rising edge of the peak, measured at 5% and 95% relative peak height. On the Neptune Plus at the British Geological Survey, R typically equates to between 9,000 –10,000, which is sufficient for partial (or pseudo) resolution of each of the silicon (Si) isotopes from their respective interferent/s.

In the revised manuscript, we have included the information of resolution "In high-resolution mode, the instrument typically exhibited a mass-resolution between approximately 9,000 to 10,000, and a sensitivity of 4-5 V/ppm."

- **Give details of how Mg measured/which ratio(s) (presumably in 'dynamic' mode), and a reference to Cardinal et al. 2004 (doi: 10.1039/b210109b) is probably appropriate.**

**Response:** The samples were doped with ~300ppb magnesium (Mg, Alfa Aesar SpectraPure). Spiking with an external element of known isotopic composition ($^{24}Mg/^{25}Mg$ = 0.126633) allows the data to be corrected for the effects of instrument-induced mass bias. Simply, any deviation of the measured $^{24}Mg/^{25}Mg$ value from the known value is attributed to the effects of mass bias. The isotopes of Si are assumed to be similarly affected, and consequently, an exponential drift correction is applied. The collector configuration used is illustrated below:

| Sequence \ Detector position | Low 4 | Low 3 | Axial | High 3 | Integration time/seconds | Settle time /seconds |
|---|---|---|---|---|---|---|
| 1 | | $^{28}Si$ | $^{29}Si$ | $^{30}Si$ | 16.8 | 3 |
| 2 | $^{24}Mg$ | | $^{25}Mg$ | | 8.4 | 3 |

Yes, we agree that the reference to Cardinal (Cardinal. D et al. J. Anal. At. Spectrom., 2003, 18, 213–218) (not Cardinal et al., 2004) should be cited here.

In the revised manuscript, the statement "Finally, all samples are spiked with approximately 300 ppb magnesium (Mg, Alfa Aesar SpectraPure, $^{24}Mg/^{25}Mg$ = 0.126633) to enable correction of the data for instrument-induced mass bias." has been revised to "Finally, all samples are spiked with approximately 300 ppb magnesium (Mg, Alfa Aesar SpectraPure, $^{24}Mg/^{25}Mg$ = 0.126633) to enable correction of the data for instrument-induced mass bias (Cardinal et al., 2003). Any deviation of the measured $^{24}Mg/^{25}Mg$ value from the known value is attributed to the effects of mass bias. The isotopes of Si are assumed to be similarly affected, and consequently, an exponential drift correction is applied (Cardinal et al., 2003)."

**References:**

Cardinal, D., Alleman, L. Y., De Jong, J., Ziegler, K., & André, L.: Isotopic composition of silicon measured by multicollector plasma source mass spectrometry in dry plasma mode. Journal of Analytical Atomic Spectrometry, 18(3), 213-218, https://doi.org/10.1039/B210109B, 2003.

- **Give approximate sample introduction rate, concentration, and instrument sensitivity**

**Response:** Yes, we have included the information of sample concentration, introduction rate, and instrument sensitivity in the revised manuscript. The statement "Silicon isotope analysis was performed in dry plasma mode using the high mass-resolution capabilityof a Thermo Fisher

Scientific Neptune Plus MC-ICP-MS (multi collector inductively coupled plasma mass spectrometer) at the Geochronology and Tracers Facility, British Geological Survey." has been revised to "Silicon isotope analysis was performed in dry plasma mode using the high mass-resolution capability of a Thermo Fisher Scientific Neptune Plus MC-ICP-MS (multi collector inductively coupled plasma mass spectrometer) at the Geochronology and Tracers Facility, British Geological Survey. Samples were typically prepared to yield a Si concentration of approximately 2 ppm and introduced to the MC-ICP-MS via an Aridus de-solvating unit, incorporating a PFA nebulizer with an uptake rate of 50 μL/min. In high-resolution mode, the instrument typically exhibited a mass-resolution between approximately 9,000 to 10,000, and a sensitivity of 4-5 V/ppm."

- **Confirm what one analytical replicate represents (just one standard-sample-standard bracket, or (as is usual) three or four?)**

**Response:** "Analytical replicates" in our study refer to the repeated analysis of the standard-sample-standard bracket, typically performed for three times. As we usually do not have sufficient sample, specifically for radiolarian tests from plankton samples, it is difficult to conduct a full procedural replicate that encompass both chemical processing and analysis. The samples selected for replication following the standard sample bracketing procedure are the ones that have a higher Si concentration and which can, therefore, be diluted to run multiple times.

   To clarify this point, in the revised manuscript, the statement "Analytical replicates were conducted where sample volume allowed…" has been revised to "Analytical replicates of the standard sample bracketing procedure were conducted where sample volume allowed…".

**L114: It's not clear what volume the 28025-102443 individuals refer to – in a 1m2 water column? It is also not clear how these numbers are derived – presumably because the volume of water passing through the nets (L66) is known? This could be clarified.**

**Response:** We are sorry for the confusion caused by our lack of clarity. The Hydro-Bios MultiNet used for collecting plankton samples has an aperture area of approximately 0.25 m$^2$ (a square aperture with 0.5 m sides). During sampling, we typically employ a retrieval rate of ~0.1 m/s for the MultiNet to ensure adequate filtration of the seawater. Therefore, the figures of 28,025–102,443 individuals represent the number of radiolarian shells in a 0.25 m$^2$ water column sampled from 0–100 m.

   In the revised manuscript, we have included the aperture area of the Hydro-Bios MultiNet. The statement "Plankton samples were collected from the 0-100 m and 100-300 m water layers at each station using a Hydro-Bios MultiNet with a 63 μm mesh size…" has been revised to "Plankton samples were collected from the 0-100 m and 100-300 m water layers at each station using a Hydro-Bios MultiNet with an aperture area of approximately 0.25 m$^2$ and a 63 μm mesh size…".

   In this study, 1/8 of each plankton sample was prepared as radiolarian slides for light microscope observations. More than 500 specimens were quantitatively identified and counted on each slide under the microscope at x100 magnification. The total number of radiolarian individuals in each sample was calculated from the count data as follows:

$$T = A * Vt/V * S * N$$

   Where T is the total number of radiolarian individuals; A is the number of radiolarian shells counted from V fields of view on the slide; Vt is the total number of view fields on the radiolarian

slide; V is the number of view fields examined under the microscope for the radiolarian individual count; S is the number of radiolarian slides; and N is the aliquot size of the sample (eight for the plankton sample, and one for the sediment sample in this study).

Based on comments from both reviewers, the original statement "To determine the species composition of radiolarians, in both plankton samples and surface sediments, all samples were wet-sieved through a 63 μm sieve and prepared into radiolarian slides following the method described by Zhang et al. (2014). Radiolarian species were then identified and counted under a Nikon optical microscope at x100 or x200 magnification, with more than 500 specimens identified on slides using the publications of Chen and Tan (1996) and Tan and Chen (1999). Relative abundances of various species were then calculated based on individual count of each species and the total number of radiolarian specimens observed under the microscope." has been revised to:

"To determine the species composition of radiolarians, in both plankton samples and surface sediments, all samples were wet-sieved through a 63 μm sieve and prepared into radiolarian slides following the method described by Zhang et al. (2014). Briefly, samples were treated with a sufficient volume of 5% HCl solution for 15 minutes to eliminate calcareous organisms. Subsequently, the residual was processed using a sonic oscillator for one minute, and subjected to differential settling to remove impurities potentially adhering to the radiolarian tests. Following these procedures, all residual material was strewn onto microscope slides and permanently mounted with Canada Balsam. Radiolarian species were then identified and counted under a Nikon optical microscope at x100 or x200 magnification, with more than 500 specimens identified on slides using the publications of Chen and Tan (1996) and Tan and Chen (1999). Radiolarian diversity was determined by the species richness in each sample. The total number of radiolarian individuals in each sample was estimated from the count data as follows:

$$T = A * (Vt/V)*S*N \tag{1}$$

Where T is the total number of radiolarian individuals; A is the number of radiolarian shells counted from V fields of view on the slide; Vt is the total number of view fields on the radiolarian slide; V is the number of view fields examined under the microscope for the radiolarian individual count; S is the number of radiolarian slides; and N is the aliquot size of the sample (eight for the plankton sample, and one for the sediment sample in this study). Relative abundances of various species were then calculated based on individual counts of each species and the total number of radiolarian specimens on each slide observed under the microscope."

**L126: Can an approximate detection limit be given for these elements?**

**Response:** In this study, the detection limit of routine EDS analysis using IT-200 EDS detector for major elements is approximately 1000pm or 0.1 %. However, elemental concentrations generated by the EDS software exceeding 0.1% do not necessarily confirm their presence within the analysed sample. As shown in Figure 4 of the manuscript, some elements (e.g. Na, exceeding 0.1%) within EDS (A" and B") spectrum images are flagged in red. This indicates that the software does not have a 99% statistical confidence in the presence of these elemental signatures within the analysed sample. This typically occurs for two reasons: either 1) the calculated wt% of these elements is very low, accompanied by a high standard deviation; or 2) the x-ray energy level peak for these elements does not stick out above the background noise peaks.

In the revised manuscript, we have included the approximate detection limit of the EDS analysis using IT-200 EDS detector for these major elements. The sentence "The concentrations of

other elements (Fe, Al, Mg, Sr, Na, K and Cl) were all below the minimum detectable limit for the EDS detector (Figure 4)" has been revised to "other elements (Fe, Al, Mg, Sr, Na, K and Cl) flagged in red within the EDS spectrum images (Figure 4) indicate that either their concentrations were below the minimum detectable limit (~ 0.1 %) for the EDS detector, or the software lacked a 99% statistical confidence in the presence of these elemental signatures within the analysed sample".

**Discussion section: In general, there is no discussion of any spatial pattern in radiolarian assemblage. But I feel there is probably useful insight here. For example, Station 28 is located away from the cluster of other stations, and visually in Fig. 3 looks different. What physicochemical parameters influence the community assemblages? As an aside, I note that the assemblage data is not made available. Could the species in Fig 3 be amalgamated at a higher taxonomic level in order to make the similarities and differences clearer? And/or condensed, via an appropriate multivariate statistical approach, to 2 axes?**

**Response:** Thanks for the important point raised by the reviewer. Yes, spatial patterns within the radiolarian assemblage are not discussed in this study. This is because, whilst samples were collected from various stations in the South China Sea (SCS) in the current study, the primary objective of the present study was to 1) determine the depth interval from which the radiolarian community contributes to the $\delta^{30}Si_{rad}$ signature in the water column, and 2) ascertain whether and why radiolarian silicon isotopes ($\delta^{30}Si_{rad}$) signatures are faithfully transferred from the water column to the sediments. Consequently, our focus has been on comparing prominent radiolarian compositions at various water depths at each station, and $\delta^{30}Si_{rad}$ values between the water column and sedimentary record at individual stations. Variations in radiolarian assemblages across different regions may potentially result in spatial differences in $\delta^{30}Si_{rad}$; however, such considerations were beyond the immediate scope of this investigation. A separate manuscript focusing on the radiolarian fractionation factor and its implication for nutrient levels in the mid-upper water column, will address the spatial distribution of $\delta^{30}Si_{rad}$ and examine the primary factors influencing this, including the regional differences in radiolarian assemblages.

The reviewer is correct in noting the difference between the radiolarian assemblage at Station 28 and those at some of the other 6 stations. This discrepancy is likely attributed to variations in regional environmental conditions, primarily including sea surface temperature (SST) and nutrient levels, such as dissolved silica or silicic acid in seawater. Station 28 is situated in the southern SCS, whereas the remaining 6 stations are located in the northern SCS. Overall, the radiolarian assemblage at station 28 exhibits a higher relative abundance of typical warm-water tropical species in radiolarian assemblages, such as *Botryocyrtis scutum* and *Pterocorys hertwigii* at station 28. This is likely due to the higher mean annual SST in the southern SCS, which forms part of the Western Pacific Warm Pool, compared to the northern SCS. Conversely, the radiolarian assemblage at station 28 has a lower relative abundance of *Didymocyrtis tetrathalamus t.,* a species indicative of nutrient-depleted Western Equatorial Pacific water intrusion into the northern SCS (e.g. Anderson et al., 1990; Zhang et al., 2015). This suggests a reduced influence of Western Pacific waters on the southern SCS compared to the northern SCS. These observed differences in specific species and radiolarian assemblages between stations in the southern and northern SCS are approximately consistent with previous investigations into variations in radiolarian

assemblages within surface sediments and the water column, and their correlation with primary environmental parameters in the SCS (Chen et al. , 2008; Zhang et al., 2005, 2009).

The primary objectives of radiolarian assemblage data presented in this study are: 1) to identify the primary contributors to $\delta^{30}Si_{rad}$ compositions in the water column at each station; 2) to ascertain the potential for substantial dissolution-induced alteration of the radiolarians by comparing the prominent radiolarian composition between the water column and sediments at each station; and 3) to provide a crucial foundation for explaining why $\delta^{30}Si_{rad}$ signatures is faithfully transferred from the water column to the sediments. We contend that amalgamating the species presented in Fig. 3 of the manuscript at a higher taxonomic level would diminish the clarity and effectiveness of this data in addressing these objectives. Therefore, we have retained the species-level information shown in Fig. 3, as it provides a more effective means of fulfilling the stated objectives for radiolarian assemblage data in this study. A correlation analysis provides a clear understanding of the similarities and differences in dominant radiolarian species between different water depth ranges and the sedimentary record at each station, as well as between various sampling stations. This analysis indicates a variation in correlations between the prominent radiolarian composition from station 28 and those from northern SCS, with Pearson correlation coefficients ranging from 0.27 to 0.83 (Table A shown on the next page).

In the revised manuscript, we will include the detailed radiolarian assemblage data (original radiolarian count data) for the samples used in this study as supplementary material.

**Reference:**

Anderson O R, Bryan M, Bennett P. Experimental and observational studies of radiolarian physiological ecology: 4. Factors determining the distribution and survival ofDidymocyrtis tetrathalamus tetrathalamus with implications for paleoecological interpretations. Marine Micropaleontology, 1990, 16(3-4): 155-167.

Chen M H, Zhang L L, Zhang L L, et al. Radiolarian assemblages in surface sediments of the South China Sea and their marine environmental correlations. Earth Science—Journal of China University of Geosciences, 2008, 33(6): 775-782.

Zhang L L, Chen M H, Lu J, et al. Polycystine radiolarian fauna and their distribution in the upper water column of the southern South China Sea. Journal of Tropical Oceanography, 2005, 24, 55–64 (in Chinese with English abstract).

Zhang L L, Chen M H, Xiang R, et al. Distribution of polycystine radiolarians in the northern South China Sea in September 2005. Marine Micropaleontology, 2009, 70(1-2): 20-38.

Table A Correlations between plankton samples and surface sediments for the prominent radiolarian species

| Sample | S7 (0-100m) | S7 (100-300m) | S7 SS | S8 (0-100m) | S8 (100-300m) | S8 SS | S10 (0-100m) | S10 (100-300m) | S10 SS | S11 (0-100m) | S11 (100-300m) | S11 SS | S12 (0-100m) | S12 (100-300m) | S12 SS | S13 (0-100m) | S13 (100-300m) | S13 SS | S28 (0-100m) | S28 (100-300m) | S28 SS |
|---|---|---|---|---|---|---|---|---|---|---|---|---|---|---|---|---|---|---|---|---|---|
| S7 (0-100m) | 1 | | | | | | | | | | | | | | | | | | | | |
| S7 (100-300m) | .77** | 1 | | | | | | | | | | | | | | | | | | | |
| S7 SS | .79** | .87** | 1 | | | | | | | | | | | | | | | | | | |
| S8 (0-100m) | .62** | .67** | .75** | 1 | | | | | | | | | | | | | | | | | |
| S8 (100-300m) | .61** | .80** | .80** | .91** | 1 | | | | | | | | | | | | | | | | |
| S8 SS | .65** | .77** | .84** | .90** | .93** | 1 | | | | | | | | | | | | | | | |
| S10 (0-100m) | .47* | .58** | .51** | .63** | .72** | .70** | 1 | | | | | | | | | | | | | | |
| S10 (100-300m) | 0.34 | .58** | .55** | .53** | .68** | .58** | .74** | 1 | | | | | | | | | | | | | |
| S10 SS | .54** | .78** | .79** | .69** | .84** | .84** | .72** | .82** | 1 | | | | | | | | | | | | |
| S11 (0-100m) | .54** | .72** | .68** | .55** | .62** | .64** | .47* | .48* | .71** | 1 | | | | | | | | | | | |
| S11 (100-300m) | .55** | .71** | .71** | .50** | .70** | .73** | .56** | .53** | .75** | .71** | 1 | | | | | | | | | | |
| S11 SS | .52** | .67** | .60** | .77** | .83** | .83** | .79** | .66** | .83** | .69** | .79** | 1 | | | | | | | | | |
| S12 (0-100m) | .50** | .62** | .62** | .66** | .68** | .74** | .64** | .55** | .79** | .84** | .64** | .80** | 1 | | | | | | | | |
| S12 (100-300m) | .56** | .77** | .71** | .69** | .80** | .74** | .73** | .70** | .80** | .71** | .68** | .86** | .75** | 1 | | | | | | | |
| S12 SS | .41* | .60** | .60** | .71** | .78** | .78** | .78** | .70** | .74** | .60** | .64** | .85** | .74** | .85** | 1 | | | | | | |
| S13 (0-100m) | .58** | .81** | .77** | .72** | .81** | .84** | .66** | .65** | .86** | .87** | .72** | .82** | .89** | .82** | .75** | 1 | | | | | |
| S13 (100-300m) | .49* | .70** | .66** | .62** | .71** | .69** | .62** | .69** | .86** | .81** | .75** | .83** | .84** | .79** | .66** | .84** | 1 | | | | |
| S13 SS | .40* | .55** | .60** | .84** | .81** | .84** | .78** | .64** | .74** | .54** | .57** | .88** | .72** | .81** | .86** | .74** | .73** | 1 | | | |
| S28 (0-100m) | 0.27 | 0.38 | .41* | .50** | .57** | .42* | .61** | .76** | .62** | .49* | 0.36 | .63** | .53** | .70** | .62** | .54** | .63** | .63** | 1 | | |
| S28 (100-300m) | 0.27 | .46* | .45* | .48* | .58** | .42* | .58** | .83** | .67** | 0.37 | 0.33 | .55** | .41* | .71** | .54** | .48* | .56** | .56** | .85** | 1 | |
| S28 SS | 0.30 | 0.38 | 0.36 | .61** | .65** | .51** | .72** | .58** | .48* | 0.30 | 0.380 | .70** | .48* | .74** | .72** | .47* | .47* | .73** | .71** | .64** | 1 |

** Correlation is significant at the 0.01 level (2-tailed)     * Correlation is significant at the 0.05 level (2-tailed)     S=station     SS=surface sediments

**L134-140: In general, it's the fractionation between dissolved Si and biogenic silica that is relevant, not the absolute $\delta^{30}$Si value. So it's a shame not to see any dissolved Si $\delta^{30}$Si data from the water samples (perhaps this is coming in a later publication?). An existing dataset of $\delta^{30}$Si exists for the SCS (Cao et al. 2012, doi: 10.1016/j.gca.2012.08.039). The overlap isn't perfect in terms of location or seasonality but its surprising that it's not mentioned here, given the importance of water $\delta^{30}$Si in setting radiolarian $\delta^{30}$Si, and that it would allow the authors to place constraints on the fractionation of Si isotopes by radiolarians.**

**Response:** The reviewer is correct. The fractionation between $\delta^{30}Si_{rad}$ and dissolved silicon isotopes ($\delta^{30}Si_{DSi}$), expressed as the apparent Si isotope fractionation factor ($\Delta^{30}Si_{rad} \sim {}^{30}\varepsilon = \delta^{30}Si_{rad} - \delta^{30}Si_{DSi}$), is an important parameter for understanding to what extent the radiolarians fractionate the $\delta^{30}Si_{DSi}$ during Si absorption, and a potential proxy for reconstructing the silicon cycle in mid-upper depth waters. Previous published data show that the $\delta^{30}Si(OH)_4$ values in the upper water column (above 100 m) of the northern South China Sea (SCS) range from 1.33 to 2.94 ‰, with a mean of 2.3 ‰ (Cao et al., 2012). Based on $\delta^{30}Si_{rad}$ compositions in plankton samples and surface sediments from our study, and a mean $\delta^{30}Si(OH)_4$ values in the northern SCS from Cao et al., (2012), the radiolarian fractionation factors can be calculated to range from −0.45 to −0.74‰, with a mean of -0.56‰ in the SCS. The mean $\Delta\delta^{30}Si_{rad}$ value calculated in our study is more positive than the radiolarian fractionation factor (−1.5‰) applied by Hendry et al. (2014), but is close to the factor (-0.8‰) reported by Abelmann et al. (2015) and that (-0.62‰, mixed radiolaria) reported by Doering et al. (2021).

However, the absence of such data in this study is for the following reasons: 1) the primary objective of the present study was to determine the depth interval from which the radiolarian community contributes to the $\delta^{30}Si_{rad}$ signature in the water column, and to ascertain whether and why radiolarian silicon isotopes ($\delta^{30}Si_{rad}$) signatures are faithfully transferred from the water column to the sediments. Consequently, our focus has been on comparing prominent radiolarian compositions at various water depths at each station, and $\delta^{30}Si_{rad}$ values between the water column and sedimentary record at individual stations. The calculation of $\Delta^{30}Si_{rad}$ is not essential for this specific investigation; 2) although Cao et al. (2012) have provided a $\delta^{30}Si(OH)_4$ dataset for the northern SCS, as the reviewer points out, the spatial and temporal overlap with our sampling regime is not sufficient to permit an accurate constraint of $\Delta^{30}Si_{rad}$. Actually, at each sampling station, when collecting plankton samples and surface sediments, we have also collected water samples from the relevant depth range for the $\Delta^{30}Si_{rad}$ constraint. Considering the typically low $Si(OH)_4$ concentrations in these water samples from the upper water column, we plan to use a Neoma MC-ICP-MS for the silicon isotopic analysis of dissolved silicon because of its enhanced sensitivity and mass resolution. However, isotopic analyses for these seawater samples are pending due to ongoing instrument-related issues with the Neoma MC-ICP-MS at the Geochronology and Tracers Facility, British Geological Survey. Once these data are obtained, a more robust mean fractionation factor can be determined; and 3) A separate manuscript detailing the radiolarian fractionation factor and its implication for nutrient levels in the mid-upper water column is to be prepared in the near future.

Reference:

Abelmann, A., Gersonde, R., Knorr, G., Zhang, X., Chapligin, B., Maier, E., Esper, O., Friedrichsen, H., Lohmann, G., Meyer, H., Tiedemann, R.: The seasonal sea-ice zone in the glacial Southern Ocean as a carbon sink, Nat. Commun., 6(1), 8136, https://doi.org/10.1038/ncomms9136, 2015.

Cao, Z., Frank, M., Dai, M., Grasse, P., Ehlert, C.: Silicon isotope constraints on sources and utilization of silicic acid in the northern South China Sea, Geochimica et Cosmochimica Acta, 97, 88-104, https://doi.org/10.1016/j.gca.2012.08.039, 2012.

Doering, K., Ehlert, C., Pahnke, K., Frank, M., Schneider, R., Grasse, P.: Silicon isotope signatures of radiolaria reveal taxon-specific differences in isotope fractionation, Front. Mar. Sci., 8, 666896, https://doi.org/10.3389/fmars.2021.666896, 2021.

Hendry. K. R., Robinson, L. F., McManus, J. F., Hays, J. D.: Silicon isotopes indicate enhanced carbon export efficiency in the North Atlantic during deglaciation. Nat. Commun., 5(1), 3107, https://doi.org/10.1038/ncomms4107, 2014.

**L165: Can an indication of the timespan covered by the upper 1cm be given? I presume there are some constraints on sedimentation rates and bioturbation in this well studied region.**

**Response:** Thanks for the reviewer's comment. Sheng et al., (2024) provide a comprehensive quantification of the spatial distribution of sedimentation rate (SR) and sediment budget over the entire South China Sea (SCS), using $^{210}$Pb measurements from 409 sediment cores, AMS$^{14}$C data from 112 gravity cores, and 33 sediment trap observations. The results show that Holocene sedimentation rates in the SCS exhibit regional variability. Specifically, under depositional conditions unaffected by turbidity currents and other anomalous processes, the mean sedimentation rate (SR) exceeds 25 cm/ka on upper continental slopes (<1000m), approximates 19 cm/ka in deeper water areas (≥1000m) along the lower continental slope slopes, and approximately 5-10 cm/ka near the deep-sea basin (>3000m) (Sheng et al., 2024). Based on these findings by Sheng et al., (2024), the estimated age of the uppermost 1cm of sediment used in this study ranges from less than approximately 40 to 130 years. During the collection of surface sediments used in this study, the retrieved box-core samples were carefully inspected to avoid sampling at any stations exhibiting apparent bioturbation or disturbance from turbidity currents.

In the revised manuscript, we have included information regarding the timespan of the uppermost 1cm sediment at the end of the section "2.1 Sample material", as follows: No discernible evidence of bioturbation or disturbance from turbidity currents was observed in the sediment samples at these stations. Based on regional variability in Holocene sedimentation rates in the SCS (Sheng et al., 2024), the estimated age of the uppermost 1cm of sediment used in this study ranges from <40 to 133 years.

**References:**

Sheng, J., Qiao, S., Shi, X., Liu, J., Liu, Y., Liu, S., Wang, K., Mohamed, C., Khokiattiwong, S., Kornkanitnan, N.: Modern sedimentation and sediment budget in the South China Sea and their comparisons with the eastern China seas, Marine Geology, 475, 107348, https://doi.org/10.1016/j.margeo.2024.107348, 2024.

**Section 4.2: The radiolarian assemblages are similar between water column and core-top, so the inference is that 'dissolution is expected to have limited impacts on these radiolarian shells' (L209) – but is this necessarily true? Is is possible weakly silicified parts of the tests are dissolving? This would be interesting to know, as it has different implications for \*how\* the δ$^{30}$Si is preserved: if no dissolution occurs, then there's no real potential for altering the isotopic signature (which therefore means the conclusions here are not transferable to other settings where more dissolution does occur). But if dissolution does occur and the δ$^{30}$Si**

**remains the same, then either a) different parts of the tests have the same δ³⁰Si or b) a coincidental balance of heavier and lighter parts dissolved. To begin to address this, it would be good to see an independent constraint on the (radiolarian) biogenic silica preservation effeciency, either from the literature, from a comparison of export vs. sediment-trap/burial fluxes, or even a theoretical predicted efficiency based on sinking speeds, water column depth, and dissolution kinetics. Finally, It would be good to see an attempt to engage with what may happen with progressive dissolution in the upper centimeters of the sediment – if there is preferential dissolution of some species here, might that introduce a bias (an apparent fractionation) to the bulk-assemblage δ³⁰Si data?**

**Response:** Thanks for reviewer's insightful comments. Yes, the observation that the radiolarian assemblages are similar between water column and surface sediments does not preclude the possibility that weakly silicified parts of the radiolarian tests have been dissolved. Although we have realized that some degree of dissolution is inevitable during the transport of radiolarians from the water column to the sediment, there is currently a paucity of established proxies for identifying weakly silicified parts of the radiolarian tests, or for tracing subtle dissolution effects. Moreover, no quantitative data are available in the literature regarding the preservation efficiency or burial fluxes of polycystine radiolarians from the South China Sea (SCS). Existing data of biogenic silica preservation efficiency (~1-3%) are not directly applicable to our study, as diatoms, a primary component of biogenic silica, exhibit a significantly higher degree of dissolution than polycystine radiolarians (e.g. Tréguer et al., 1995; Ragueneau et al., 2000). Consequently, obtaining effective preservation efficiency data suitable for assessing the dissolution of polycystine radiolarians is a significant challenge.

As shell fracture of microfossils preserved within sediments is commonly attributed to partial dissolution (e.g. Murray and Alve, 1999; Ryves et al., 2001), the proportion of fractured radiolarian shells may be indicative of the potential for radiolarian shell dissolution during sinking. Moreover, the preservation of delicate radiolarian skeletal structures, as assessed by SEM images, may also serve to determine whether the radiolarian shells have undergone significant dissolution. Therefore, in the revised manuscript, we have added the content regarding the assessment of radiolarian shell preservation in both the water column and sediments. This allows us to examine whether significant dissolution of radiolarian shells occurs during sinking within the water column. We have therefore made following essential revisions to the manuscript:

In the revised manuscript, the title of section 2.2 "Radiolarian composition analysis" has been revised to "Radiolarian composition and preservation analysis", and we have included a second paragraph regarding assessing the preservation of radiolarian shells: "To assess the preservation of radiolarian shells both within the water column and seabed sediments, the proportion of fractured radiolarian shells was quantified in each studied sample under a Nikon optical microscope at x100 magnification. Further scanning electron microscopy (SEM) was employed to examine the potential for dissolution-induced alteration of the radiolarian skeleton.".

Following the first paragraph of section 2.3, we have included the result of radiolarian preservation analysis and a new Figure 4 (see below): "The proportion of fractured shells was low in the studied samples (Figure 4), generally ranging from 2 to 5% (mean = 3%) in plankton samples and from 3 to 9% (mean = 6%) in surface sediments. SEM images reveal a typical morphology of the pores on the radiolarian shell (Figure 4E), along with a high degree of integrity in the delicate skeletal structures (Figure 4F). These observations suggest good preservation of

radiolarian shells, with no significant dissolution evident in either the water column or the sediments."

[Figure]

Figure 4 Observations of radiolarian shell preservation at station 12 (water depth: 3497 m) in the plankton sample (100-300 m) (A and C) and surface sediments (B, D, E, and F) using optical and the scanning electron microscope. FRS=Fractured radiolarian shells; DS=Delicate skeletons

The first paragraph of Section "4.2 Transfer of radiolarian $\delta^{30}$Si signatures into the sediment record" has been revised to "At each sampling station, $\delta^{30}$Si$_{rad}$ compositions (mean = 1.73‰) in the surface sediment closely resemble those (mean = 1.74‰) in the overlying water column evidenced by the paired t-test (p=0.75), indicating a faithful transfer of the $\delta^{30}$Si signal incorporated into radiolarian skeletons from the water column to sediments. This suggests that dissolution has a minimal impact on the $\delta^{30}$Si$_{rad}$ signatures as radiolarian shells sink through the water column and become incorporated into the sediment record. One of two possibilities may account for this observation: 1) the radiolarian shells may not have undergone substantial dissolution during sinking; 2) the radiolarian shells have experienced substantial dissolution, but this process may not significantly alter their isotope composition. Considering minor differences in the mean proportion of fractured radiolarian shells between the plankton sample (~3%) and surface sediments (6%), the well-preserved state of radiolarian shells (Figure 4), and the

approximate correspondence between prominent radiolarian species identified in plankton samples and those present in surface sediments at each sampling station (as detailed in section 4.1), we propose that radiolarian shells have not experienced substantial dissolution or remineralisation during their transfer from the water column to the sediments. ”

At the beginning of the second paragraph of section "4.2 Transfer of radiolarian $\delta^{30}$Si signatures into the sediment record", we have added "The susceptibility of radiolarian shells to dissolution varies considerably with different taxonomic groups due to the differences in the chemical constituent of their skeletons."

**References:**

Murray, J. W., Alve, E.: Natural dissolution of modern shallow water benthic foraminifera: taphonomic effects on the palaeoecological record, Palaeogeography, Palaeoclimatology, Palaeoecology, 146(1-4), 195-209, https://doi.org/10.1016/S0031-0182(98)00132-1, 1999.

Ragueneau, O., Tréguer, P., Leynaert, A., Anderson, R. F., Brzezinski, M. A., DeMaster, D. J., Dugdale, R. C., Dymond, J., Fischer, G., Francois, R., Heinze, C., Maier-Reimer, E., Martin-Jézéquel, V. Nelson D. M., Quéguiner, B.: A review of the Si cycle in the modern ocean: recent progress and missing gaps in the application of biogenic opal as a paleoproductivity proxy, Global. Planet. Change, 26(4), 317-365, https://doi.org/10.1016/S0921-8181(00)00052-7, 2000.

Ryves, D. B., Juggins, S., Fritz, S. C., Battarbee, R. W.: Experimental diatom dissolution and the quantification of microfossil preservation in sediments. Palaeogeography, Palaeoclimatology, Palaeoecology, 172(1-2), 99-113, https://doi.org/10.1016/S0031-0182(01)00273-5, 2001.

Treguer, P., Nelson, D. M., Van Bennekom, A. J., DeMaster, D. J., Leynaert, A., Quéguiner, B. The silica balance in the world ocean: a reestimate, Science, 268(5209), 375-379, https://doi.org/10.1126/science.268.5209.375, 1995.

---

## Author Comment (AC3)

The authors would like to thank the editor and the reviewer for their critical assessment of our work, as well as their invaluable and constructive comments, which significantly improve the manuscript. We have addressed all comments point by point as follows, and changes to the text in the manuscript are shown in **blue**.

**Reviewer 2**

**In their manuscript Faithful transfer of radiolarian silicon isotope signatures from water column to sediments in the South China Sea" the authors present the first combined water column and surface sediment dataset of silicon isotope compositions of radiolaria. This data is further supplemented with radiolarian assemblage countings, which allows to investigate the influence of transfer from the water column to the sediment, especially via dissolution. The data presented is of great interest as d30Sirad has a high potential as a proxy to allow surface to intermediate water column reconstructions of the silica cycle, but is limited by few investigations on specific processes.**

**While I find the dataset of great importance and the presented findings are interesting and presented in a concise format, the authors should re-structure the discussion a bit and not solely discuss the influence of cleaning method and dissolution. This dataset has much more potential, especially as there is a dataset for d30Si-DSi values from the same area that could be used to investigate fractionation factors. I have summarized my main questions and comments below, followed by detailed comments per line.**

**Main questions:**

**1. The major question in terms of silicon isotope compositions of radiolaria in the community at the moment is the question about their fractionation factor. As your study provides $\delta^{30}Si_{rad}$ from the water column and the sediment I am wondering why there is no discussion at all about the fractionation of Si and a calculation of $\delta^{30}Si$ (fractionation factor) even if you just measured the bulk radiolarian composition, as there are silicon isotope compositions of DSi from the water column available (Cao et al 2012, https://doi.org/10.1016/j.gca.2012.08.039).**

**Response:** We thank the reviewer for raising this important point. We understand that the fractionation between $\delta^{30}Si_{rad}$ and dissolved silicon isotopes ($\delta^{30}Si_{DSi}$), expressed as the apparent Si isotope fractionation factor ($\Delta^{30}Si_{rad} \sim {}^{30}\varepsilon = \delta^{30}Si_{rad} - \delta^{30}Si_{DSi}$), is an important parameter for understanding to what extent the radiolarians fractionate the $\delta^{30}Si_{DSi}$ during Si absorption, and a potential proxy for reconstructing the silicon cycle in mid-upper depth waters. Cao et al.,(2012) have documented that the $\delta^{30}Si(OH)_4$ values in the upper water column (above 100 m) of the northern South China Sea (SCS) range from 1.33 to 2.94 ‰, with a mean of 2.3 ‰. Based on $\delta^{30}Si_{rad}$ compositions in plankton samples and surface sediments from our study, and a mean $\delta^{30}Si(OH)_4$ values in the northern SCS from Cao et al., (2012), the radiolarian fractionation factors can be calculated to range from −0.45 to −0.74‰, with a mean of -0.56‰ in the SCS. The mean $\Delta\delta^{30}Si_{rad}$ value calculated in our study is more positive than the radiolarian fractionation factor (−1.5‰) applied by Hendry et al. (2014), but is close to the factor (-0.8‰) reported by Abelmann et al. (2015) and that (-0.62‰, mixed radiolaria) reported by Doering et al. (2021).

However, we do not report $\Delta\delta^{30}Si_{rad}$ values in this study for the following reasons: 1) the primary objective of the present study was to determine the depth interval from which the radiolarian community contributes to the $\delta^{30}Si_{rad}$ signature in the water column, and to ascertain whether and why radiolarian silicon isotopes ($\delta^{30}Si_{rad}$) signatures are faithfully transferred from the water column to the sediments. Consequently, our focus has been on comparing prominent radiolarian compositions at various water depths at each station, and $\delta^{30}Si_{rad}$ values between the water column and sedimentary record at individual stations. The calculation of $\Delta^{30}Si_{rad}$ is not essential for this specific investigation; 2) although Cao et al. (2012) have provided a $\delta^{30}Si(OH)_4$ dataset for the northern SCS, the spatial overlap with our sampling regime is not sufficient to permit an accurate constraint of $\Delta^{30}Si_{rad}$ and a meaningful discussion regarding the impact of $\delta^{30}Si_{DSi}$ signatures on $\delta^{30}Si_{rad}$ compositions. Actually, at each sampling station, when collecting plankton samples and surface sediments, we have also collected water samples from the relevant depth range for the $\Delta^{30}Si_{rad}$ constraint. Considering the typically low dissolved silicon concentrations in these water samples from the upper water column, we plan to use a Neoma MC-ICP-MS for the silicon isotopic analysis of dissolved silicon because of its enhanced sensitivity and mass resolution. However, isotopic analyses for seawater samples are pending due to ongoing instrument-related issues with the Neoma MC-ICP-MS at the Geochronology and Tracers Facility, British Geological Survey. Once these data are obtained, a more robust fractionation factor can be determined; and 3) A separate manuscript detailing the radiolarian fractionation factor and its implication for nutrient levels in the mid-upper water column is to be preparation in the near future.

**Reference:**

Abelmann, A., Gersonde, R., Knorr, G., Zhang, X., Chapligin, B., Maier, E., Esper, O., Friedrichsen, H., Lohmann, G., Meyer, H., Tiedemann, R.: The seasonal sea-ice zone in the glacial Southern Ocean as a carbon sink, Nat. Commun., 6(1), 8136, https://doi.org/10.1038/ncomms9136, 2015.

Cao, Z., Frank, M., Dai, M., Grasse, P., Ehlert, C.: Silicon isotope constraints on sources and utilization of silicic acid in the northern South China Sea, Geochimica et Cosmochimica Acta, 97, 88-104, https://doi.org/10.1016/j.gca.2012.08.039, 2012.

Doering, K., Ehlert, C., Pahnke, K., Frank, M., Schneider, R., Grasse, P.: Silicon isotope signatures of radiolaria reveal taxon-specific differences in isotope fractionation, Front. Mar. Sci., 8, 666896, https://doi.org/10.3389/fmars.2021.666896, 2021.

Hendry. K. R., Robinson, L. F., McManus, J. F., Hays, J. D.: Silicon isotopes indicate enhanced carbon export efficiency in the North Atlantic during deglaciation. Nat. Commun., 5(1), 3107, https://doi.org/10.1038/ncomms4107, 2014.

**2. The sample preparation and evaluation should be described in a bit more detail as you are referring to a newly published method it would be good to at least mention the main steps involved in the preparation (see detailed comments below). Furthermore, I would be interested to know if sample amounts were generally too low to try to pick single species or orders of radiolaria, as this would help better understand how different radiolarians incorporate Si and if there is a difference in Si fractionation.**

**Response:** According to the reviewer's suggestion, we have included a summary of the method used to extract and purify radiolarian tests for silicon isotope analysis (Zhang and Swann, 2023) in the revised version of the manuscript. The statement in the current manuscript "For isotope

analysis, sufficient radiolarian tests were extracted and purified from 7 plankton samples and 7 surface sediment samples following the method of Zhang and Swann (2023)." has been revised to "For isotope analysis, approximate 1 to 1.5 mg of radiolarian tests were extracted and purified from 7 plankton samples and 7 surface sediment samples following the method of Zhang and Swann (2023). Overall, the procedure for extracting and purifying radiolarian tests from marine sediments in Zhang and Swann (2023) comprises four stages: chemical treatment, initial sieving and differential settling, subsequent sieving, and finally density separation (Figure A). In the first stage, raw samples were treated with ~30% $H_2O_2$ and 15% HCl to remove the organic matters and calcareous components, and to facilitate particle dispersal. Following chemical treatment, the particles were rinsed and filtered using a 53 µm sieve to remove fine detritus, small diatoms, seaweed spines, and some sponge spicules. The filtered particles then underwent differential settling two to three times, followed by sonication, to further isolate radiolarians from large diatoms. Subsequently, particles were filtered three times using a 53 µm sieve by half immersing the sieve in a container filled with distilled deionized water (DDW) and gently tapping the base of the sieve for 10-15 minutes. This process aimed to remove all monoaxonic spicules and a portion of the small non-monoaxonic spicules. Finally, the retained fraction was further refined by density separation, using specific gravities from 2.1-2.0 g/cm$^3$ at 0.5-unit interval, to remove any remaining coarse detritus and non-monoaxonic sponge spicules. The purity of extracted radiolarian tests in each sample was visually assessed under a Nikon optical microscope at x100 magnification, with selected samples further examined using a scanning electron microscopy (SEM) (JEOL JSM-IT200) equipped with an energy dispersive X-ray spectroscopy (EDS) detector at the Nanoscale and Microscale Research Centre, University of Nottingham."

Indeed, $\delta^{30}$Si records from single-species tests may improve our understanding of how different radiolarians incorporate Si and allow us to examine potential variations in Si isotope fractionation between different radiolarian species. However, obtaining sufficient material for silicon isotopic analysis from individual radiolarian taxa in our samples proved to be a significant challenge, necessitating our focus on the bulk species record for this study. We did, in fact, attempt to isolate single-species tests via manual picking under a microscope, but encountered several obstacles. Firstly, for silicon isotope analysis using a Thermo Fisher Scientific Neptune Plus MC-ICP-MS at the Geochronology and Tracers Facility, British Geological Survey, 1 to 1.5 mg of purified radiolarian tests is typically required. Radiolarian tests from sediments in the tropical Ocean are notably lightweight, averaging 0.063 to 0.136 mg/shell (Moore, 1969; Takahashi, 1982). Consequently, a mean of approximately 7,000 to 15,000 individual tests are required to achieve the ~1 mg of material needed for the silicon isotope analysis. Given the high diversity of radiolarians in Holocene sediments, particularly in low-latitude oceans (Moore, 1969; Takahashi, 1982; Boltovskoy et al., 2010) and in our samples, which average over 100 species per sample, coupled with the observation that most species typically constitute less than 10% of the total radiolarian community (Boltovskoy et al., 2010; Chen et al., 2008), this means that 70,000 to 150,000 bulk radiolarian tests are required for each sample to obtain sufficient single-species material. Regrettably, the limited volume of the plankton samples made it difficult to obtain such a large number of radiolarian shells. Secondly, manually picking 7,000 to 15,000 individual tests out of 70,000 to 150,000 bulk radiolarian tests is highly time-consuming, rendering this approach impractical for the study of single-species radiolarian $\delta^{30}$Si. Furthermore, even with meticulous hand-picking under a microscope, the electro static adsorption of these light tests to transfer tools,

such as brushes, inevitably results in test loss during transfer to storage vials. This further compounded the difficulty of accumulating the required quantity of monospecific tests for the isotope analysis.

It was these challenges encountered during our attempts at hand-picking that prompted us to develop the method for extracting and purifying bulk radiolarian tests from the sediments for the study of the $\delta^{30}Si_{rad}$ (Zhang and Swann, 2023). This method enabled us to effectively obtain sufficient pure radiolarian tests to determine mixed-radiolarian $\delta^{30}Si$ compositions, prior to the availability of more effective techniques for obtaining sufficient monospecific tests for isotope analysis.

In the revised manuscript we have included a brief explanation regarding the absence of silicon isotopic analysis for individual radiolarian taxa: "Given the lightweight nature of shells and the high diversity of radiolarians, particularly in low-latitude oceans (Moore, 1969; Takahashi, 1982; Boltovskoy et al., 2010), combined with the observation that most species typically comprise less than 10% of the total radiolarian assemblage (Boltovskoy et al., 2010; Chen et al., 2008a), obtaining sufficient material from individual radiolarian taxa for silicon isotopic analysis remains a considerable challenge."

**References:**

Boltovskoy, D., Kling, S. A., Takahashi, K., Bjørklund, K.: World atlas of distribution of recent polycystina (Radiolaria), Palaeontologia Electronica, 13, 1–229, 2010.

Chen M H, Zhang L L, Zhang L L , et al. 2008a. Preservation of radiolarian diversity and abundance in surface sediments of the South China Sea and its environmental implication. Journal of China University of Geosciences, 19: 217–229.

Doering, K., Ehlert, C., Pahnke, K., Frank, M., Schneider, R., Grasse, P.: Silicon isotope signatures of radiolaria reveal taxon-specific differences in isotope fractionation, Front. Mar. Sci., 8, 666896, https://doi.org/10.3389/fmars.2021.666896, 2021.

Moore, JR. T. C: Radiolaria: change in skeletal weight and resistance to solution. Geological Society of America Bulletin, 80(10), 2103-2108. https://doi.org/10.1130/0016-7606(1969)80[2103:RCISWA]2.0.CO;2, 1969

Takahashi, K.: Vertical flux, ecology and dissolution of Radiolaria in tropical oceans: implications for the silica cycle. PhD thesis. Woods Hole Oceanographic Institution and Massachusetts Institute of Technology, 1982

Zhang, Q., Swann, G. E. A.: An effective method to extract and purify radiolaria from tropical marine sediments, Front. Mar. Sci., 10, 1150518, https://doi.org/10.3389/fmars.2023.1150518, 2023.

**3. The discussion feels a bit shallow. The highlight of your manuscript is the investigation of any dissolution effect on the sedimentary radiolarian signal. However, as you point out in the discussion, the prominent radiolarian species in your samples turn out to be species that are considered to be quite resistant to resolution. I suggest including some of the references you refer to in the last part of the discussion in the introduction instead and cutting the actual discussion about dissolution shorter, as you basically show that there is no evidence on dissolution based on the abundance data and the silicon isotope data. While you show this with your statistical analyses, you can also highlight that your d30Sirad values of water column and surface sediments are the same within analytical precision as indicated in Figure 5.**

**Response:** Thanks for the reviewer's comment. In the manuscript, the discussion section "4 Discussions" is structured in two parts. Given that radiolarians are distributed throughout the

water column from surface to bottom waters in the oceans (e.g. Kling, 1979; Boltovskoy et al., 2010; Hu et al., 2015), the first part of the discussion, "4.1 Radiolarian $\delta^{30}$Si signatures in water column plankton and surface sediment samples", primarily aims to determine the depth interval from which the radiolarian community contributes to the $\delta^{30}$Si$_{rad}$ signature in the water column. This determination is achieved through the analysis and comparison of radiolarian diversity and abundance at various depths. The primary objective is to improve our understanding of the water depth interval represented by $\delta^{30}$Si$_{rad}$ records in both the water column and sediments, thereby providing the essential basis for reconstructing the silicon cycle within specific depth intervals, utilising radiolarian silicon isotope records from sedimentary sequences. To improve clarity concerning the focus of this section, the original title of this part "Radiolarian $\delta^{30}$Si signatures in water column plankton and surface sediment samples" has been revised as "4.1 Contributors to radiolarian $\delta^{30}$Si signatures in the water column plankton" in the revised version of the manuscript. Furthermore, the finding that "$\delta^{30}$Si $_{rad}$ signatures in the water column are primarily contributed to by radiolarians from the 0-100 m water depth layer in the SCS" has been added in the abstract.

The second part of the discussion "4.2 Transfer of radiolarian $\delta^{30}$Si signatures into the sediment record" focuses on explaining why the $\delta^{30}$Si$_{rad}$ signal from the water column is faithfully transferred to the sediments. Our data indicate a good consistency between $\delta^{30}$Si$_{rad}$ signatures in the water column and the sedimentary record, indicating a faithful transfer of the $\delta^{30}$Si signal incorporated into radiolarian skeletons from the water column to sediments. This suggests that dissolution has a minimal impact on the $\delta^{30}$Si$_{rad}$ signatures as radiolarian shells sink through the water column and become incorporated into the sediment record. This observation may be attributable to one of two possible reasons: 1) the radiolarian shells may not have undergone substantial dissolution during sinking; 2) the radiolarian shells have experienced substantial dissolution, but this process may not significantly alter their isotope composition. As shell fracture of microfossils preserved within sediments is commonly attributed to partial dissolution (e.g. Murray and Alve, 1999; Ryves et al., 2001), the proportion of fractured radiolarian shells may be indicative of the potential for radiolarian shell dissolution during sinking. Moreover, the preservation of delicate radiolarian skeletal structures, as assessed by SEM images, may also serve to determine whether the radiolarian shells have undergone significant dissolution. Therefore, in the revised manuscript, we have added the content regarding the assessment of the preservation of radiolarian shells in both the water column and sediments. This allows us to examine whether significant dissolution of radiolarian shells occurs during sinking within the water column. We have therefore made following essential revisions to the manuscript:

In the revised manuscript, the title of section 2.2 "Radiolarian composition analysis" has been revised to "Radiolarian composition and preservation analysis", and we have included a second paragraph regarding assessing the preservation of radiolarian shells: "To assess the preservation of radiolarian shells both within the water column and seabed sediments, the proportion of fractured radiolarian shells was quantified in each studied sample under a Nikon optical microscope at x100 magnification. Further scanning electron microscopy (SEM) was employed to examine the potential for dissolution-induced alteration of the radiolarian skeleton.".

Following the first paragraph of section 2.3, we have included the result of radiolarian preservation analysis and a new Figure 4 (see below): "The proportion of fractured shells was low in the studied samples (Figure 4), generally ranging from 2 to 5% (mean = 3%) in plankton samples and from 3 to 9% (mean = 6%) in surface sediments. SEM images revealed a typical

morphology of the pores on the radiolarian shell (Figure 4E), along with a high degree of integrity in the delicate skeletal structures (Figure 4F). These observations suggest good preservation of radiolarian shells, with no significant dissolution evident in either the water column or the sediments."

[Figure]

Figure 4 Observations of radiolarian shells preservation at station 12 (water depth: 3497 m) in the plankton sample (100-300 m) (A and C) and surface sediments (B, D, E, and F) using optical and the scanning electron microscope. FRS=Fractured radiolarian shells; DS=Delicate skeletons

The first paragraph of Section "4.2 Transfer of radiolarian $\delta^{30}$Si signatures into the sediment record" has been revised to "At each sampling station, $\delta^{30}$Si$_{rad}$ compositions (mean = 1.73‰) in the surface sediment closely resembles those (mean = 1.74‰) in the overlying water column evidenced by the paired t-test (p=0.75), indicating a faithful transfer of the $\delta^{30}$Si signal incorporated into radiolarian skeletons from the water column to sediments. This suggests that dissolution has a minimal impact on the $\delta^{30}$Si$_{rad}$ signatures as radiolarian shells sink through the water column and become incorporated into the sediment record. One of two possibilities may account for this observation: 1) the radiolarian shells may not have undergone substantial dissolution during sinking; 2) the radiolarian shells have experienced substantial dissolution, but this process may not significantly alter their isotope composition. Considering minor differences

in the mean proportion of fractured radiolarian shells between the plankton sample (~3%) and surface sediments (6%), the well-preserved state of radiolarian shells (Figure 4), and the approximate correspondence between prominent radiolarian species identified in plankton samples and those present in surface sediments at each sampling station (as detailed in section 4.1), we propose that radiolarian shells have not experienced substantial dissolution or remineralisation during their transfer from the water column to the sediments. "

The dissolution characteristics of radiolarian shells vary considerably with different taxonomic groups due to notable differences in the chemical constituent of their skeletons. For example, Collodaria radiolarians, a group belonging to polycystine radiolarians, are highly susceptible to dissolution during sinking due to their skeletons composed of both opal and Celestine (Afanasieva et al., 2005; Afanasieva and Amon, 2014). However, as indicated above, our observations provide no evidence that the radiolarian shells have experienced substantial dissolution during sinking within the water column. Therefore, the second part of the discussion need to addresses why radiolarians preserved in surface sediments have not undergone significant dissolution. Only by clarifying the shell chemical composition and dissolution characteristics of different radiolarian groups, as well as the approximate proportion of each radiolarian group in our studied samples, can we better understand why the radiolarian shells are minimally impacted by dissolution during sinking through the water column. This thus allows for a compelling explanation of why the radiolarian silicon isotope signal from the water column is faithfully transferred to the sediments. Therefore, information regarding the radiolarian taxonomy, chemical composition and dissolution characteristics of each taxonomic group, and proportion of each taxonomic group in radiolarian community in our analyzed samples, is essential in the discussion of the manuscript.

In the revised manuscript, we have added "The susceptibility of radiolarian shells to dissolution varies considerably with different taxonomic groups due to the differences in the chemical constituent of their skeletons." to the beginning of the second paragraph of section "4.2 Transfer of radiolarian $\delta^{30}$Si signatures into the sediment record".

While $\delta^{30}$Si$_{rad}$ values of water column and surface sediments are considered to be the same within analytical precision, we conducted a statistical analysis because it could more objectively describe this similarity in $\delta^{30}$Si$_{rad}$ values in water column and surface sediments.

**References:**

Afanasieva, M. S., Amon, E. O.: Biomineralization of radiolarian skeletons, Paleontological Journal, 48, 1473-1486, https://doi.org/10.1134/S0031030114140020, 2014.

Afanasieva, M. S., Amon, E. O., Agarkov, Y. V., Boltovskoy, D. S.: Radiolarians in the geological Record, Paleontological Journal, 39, 135-392, https://doi.org/10.1666/0094-8373(2005)031, 2005.

Boltovskoy, D.: Vertical distribution patterns of Radiolaria Polycystina (Protista) in the World Ocean: living ranges, isothermal submersion and settling shells, Journal of Plankton Research, 39(2), 330-349, https://doi.org/10.1093/plankt/fbx003, 2017.

Hu, W. F., Zhang, L. L., Chen, M. H., Zeng, L. L., Zhou, W. H., Xiang, R., Zhang, Q., Liu, S. H.: Distribution of living radiolarians in spring in the South China Sea and its responses to environmental factors, Sci. China Earth Sci., 58, 270-85, https://doi.org/10.1007/s11430-014-4950-0, 2015

Kling, S. A.: Vertical distribution of polycystine radiolarians in the central North Pacific, Mar. Micropaleontol., 4, 295-318, https://doi.org/10.1016/0377-8398(79)90022-7, 1979.

**Comments per line:**

**L6: You should remove the comma after "water column".**

Response: Yes, the comma after "water column" has been deleted in the revised manuscript.

**L8: Add a comma before "as evidenced".**

Response: Yes, the comma before "as evidenced" has been added.

**L18: Add a comma behind "Based on this".**

Response: Yes, the comma before "as evidenced" has been added.

**L25: Remove "actually".**

Response: Yes, the term "actually" has been deleted.

**L26-27: Remove the commas behind "mechanism" and "impact" and "in both glacials".**

Response: Yes, the commas behind "mechanism", "impact", and "in both glacials" have been deleted.

**L29: I suggest removing "wider" or replacing it with "broader".**

Response: According to the reviewer's suggestion, the term "wider" has been deleted.

**L42: Remove "so".**

Response: Yes, the term "so" has been deleted.

**L46: Change to "others".**

Response: We appreciate the reviewer's suggestion. However, we have confirmed that the term "others" is correct in line 46 of the current manuscript.

**L49: Remove "also".**

Response: Yes, the word "also" has been deleted.

**L 54: The authors claim that purifying radiolarian test from the natural environment have difficulties. Can the authors add a reference here and specify what difficulties they mean?**

Response: We agree with the reviewer that the reference is needed regarding the difficulties in purifying radiolarian test from the natural environment. Prior to the method for extracting and purifying radiolarians from sediments proposed by Zhang and Swann (2023), the approach to obtain purified radiolarians for isotope analysis was manual picking under a light microscope (Hendry et al., 2014; Fontorbe et al., 2016; Fontorbe et al., 2017; Doering et al., 2021). In general, there are three problems with manual picking. Firstly, radiolarian tests from Quaternary sediments in the tropical Pacific Ocean are light with an average weight from 0.063 to 0.136 mg/shell (Moore, 1969; Takahashi, 1982). As such, 7,000 to 15,000 radiolarian tests need to be handpicked per 1 mg of material needed for isotope analysis. Secondly, even if tests were handpicked under a

microscope, it is unavoidable to lose some radiolarian tests when transferring samples to a storage vial due to electrostatic adsorption of the light shells to the transfer tool, such as a brush. Finally, the composition of the radiolarian assemblages may be changed substantially because of the randomness of manual picking or priority selection of larger individuals, which may bias the measured isotope compositions. Because these difficulties were stated in Zhang and Swann (2023), we do not reiterate them in this manuscript.

In the revised manuscript, we have included the relevant references regarding the difficulties in successfully culturing radiolarians and purifying radiolarian test from the natural environment (Suzuki and Aita, 2011; Suzuki and Not, 2015; Zhang and Swann, 2023).

References:

Suzuki, N., Aita, Y.: Radiolaria: achievements and unresolved issues: taxonomy and cytology, Plankton and Benthos Research, 6(2), 69-91, https://doi.org/10.3800/pbr.6.69, 2011.

Suzuki, N., Not, F: Biology and Ecology of Radiolaria, In: Marine Protists, edited by: Ohtsuka, S., Suzaki, T., Horiguchi, T., Suzuki, N., Not, F., Springer, Tokyo, Japan, https://doi.org/10.1007/978-4-431-55130-0_8, 2015.

Zhang, Q., Swann, G. E. A.: An effective method to extract and purify radiolaria from tropical marine sediments, Front. Mar. Sci., 10, 1150518, https://doi.org/10.3389/fmars.2023.1150518, 2023.

**L70: Remove the comma behind "box corer" and add an article (either "the" or "a") before "cold storage"**

**Response:** Yes, we have deleted the comma behind "box corer" and added "the" before the term "cold storage".

**L73-78: Under section 2.2 the authors refer to the preparation of radiolarian slides but only refer to a previous paper concerning the actual method. Please add some specifics of which steps this methodology includes concisely.**

**Response:** Based on the reviewer's comment, we have added a brief description regarding the preparation of radiolarian slides to the revised manuscript. Furthermore, we have clarified the methodology used to estimate the total number of radiolarian individuals in each sample in accordance with the comment from the other reviewer.

The statement in lines 73-78 of the current manuscript "To determine the species composition of radiolarians, in both plankton samples and surface sediments, all samples were wet-sieved through a 63 μm sieve and prepared into radiolarian slides following the method described by Zhang et al. (2014). Radiolarian species were then identified and counted under a Nikon optical microscope at x100 or x200 magnification, with more than 500 specimens identified on slides using the publications of Chen and Tan (1996) and Tan and Chen (1999). Relative abundances of various species were then calculated based on individual count of each species and the total number of radiolarian specimens observed under the microscope." has been revised to:

"To determine the species composition of radiolarians, in both plankton samples and surface sediments, all samples were wet-sieved through a 63 μm sieve and prepared into radiolarian slides following the method described by Zhang et al. (2014). Briefly, samples were treated with a sufficient volume of 5% HCl solution for 15 minutes to eliminate calcareous organisms. Subsequently, the residual was processed using a sonic oscillator for one minute, and subjected to differential settling to remove impurities potentially adhering to the radiolarian tests. Following these procedures, all residual material was strewn nearly homogenously onto microscope slides

and permanently mounted with Canada Balsam. Radiolarian species were then identified and counted under a Nikon optical microscope at x100 or x200 magnification, with more than 500 specimens identified on slides using the publications of Chen and Tan (1996) and Tan and Chen (1999). Radiolarian diversity was determined by the species richness in each sample. The total number of radiolarian individuals in each sample was estimated from the count data as follows:

$$T = A * (Vt/V)*S*N \qquad\qquad (1)$$

Where T is the total number of radiolarian individuals; A is the number of radiolarian shells counted from V fields of view on the slide; Vt is the total number of view fields on the radiolarian slide; V is the number of view fields examined under the microscope for the radiolarian individual count; S is the number of radiolarian slides; and N is the aliquot size of the sample (eight for the plankton sample, and one for the sediment sample in this study). Relative abundances of various species were then calculated based on individual counts of each species and the total number of radiolarian specimens on each slide observed under the microscope."

**L78: Changes to "individual counts".**
**Response:** Thanks for the meticulous review by the reviewer. The phrase that the reviewer mentioned should be "individual count", which appears on line 77 of the current manuscript. We have revised the manuscript accordingly, replacing "individual count" with "individual counts".

**L 80-81: Similarly to my comment above the others only refer to a previous paper concerning the preparation of surface sediment, please add a short description what this includes, freezing, freeze-drying, wet-sieving as stated in the section above? Further, can you specify what "sufficient" radiolarian tests are?**
**Response:** Yes, following the reviewer's comment, we have included a brief overview of Zhang and Swann (2023) in the revised manuscript as follows: Overall, the procedure for extracting and purifying radiolarian tests from marine sediments in Zhang and Swann (2023) comprises four stages: chemical treatment, initial sieving and differential settling, subsequent sieving, and finally density separation (Figure A). In the first stage, raw samples were treated with ~30% $H_2O_2$ and 15% HCl to remove the organic matters and calcareous components, and to facilitate particle dispersal. Following chemical treatment, the particles were rinsed and filtered using a 53 μm sieve to remove fine detritus, small diatoms, seaweed spines, and some sponge spicules. The filtered particles then underwent differential settling two to three times, followed by sonication, to further isolate radiolarians from large diatoms and other detrital fine-grains. Subsequently, particles were filtered three times using a 53 μm sieve by half immersing the sieve in a container filled with distilled deionized water (DDW) and gently tapping the base of the sieve for 10-15 minutes. This process aimed to remove all monoaxonic spicules and a portion of the small non-monoaxonic spicules. Finally, the retained fraction was further refined by density separation, using specific gravities from 2.1-2.0 $g/cm^3$ at 0.5-unit interval, to remove any remaining coarse detritus and non-monoaxonic sponge spicules. The purity of extracted radiolarian tests in each sample was visually assessed under a Nikon optical microscope at x100 magnification, with selected samples further examined using a scanning electron microscopy (SEM) (JEOL JSM-IT200) equipped with an energy dispersive X-ray spectroscopy (EDS) detector at the Nanoscale and Microscale Research Centre, University of Nottingham.

We are sorry for the lack of clarity regarding the typical mass of radiolarian tests for isotope analysis. For silicon isotope analysis, approximately 1 to 1.5 mg of purified radiolarian tests is typically required using a Thermo Fisher Scientific Neptune Plus MC-ICP-MS at the Geochronology and Tracers Facility, British Geological Survey. In this study, we typically extracted over 1.5 mg of pure radiolarian tests for isotope analysis from all surface sediment samples. However, due to limited sample volumes, the quantity of pure radiolarian tests obtained from plankton samples was approximately 1 mg.

In the revised manuscript, the statement "…sufficient radiolarian tests…" has been revised to "…approximate 1 to 1.5 mg of radiolarian tests…".

**L85: rephrase to "using NaOH fusion".**
**Response:** Yes, the phrase "using the NaOH fusion method" has been revised to "using NaOH fusion" in the revised manuscript.

**L86: Remove "between".**
**Response:** Yes, the word "between" has been deleted, and the phrase "a pH of between 2-8" has therefore been revised to "a pH of 2-8".

**L95: You are referring to analytical replicates here. Was the same sample prepared by NaOH fusion measured on a different day, or are you referring to replicates within the standard sample bracketing procedure?**
**Response:** "Analytical replicates" in our study refer to the repeated analysis of the standard-sample-standard bracket, typically performed for three times. As we usually do not have sufficient sample specifically for radiolarian tests from plankton samples, it difficult to conduct a full procedural replicate that encompass both chemical processing and analysis. The samples selected for replication following the standard sample bracketing procedure are the ones that have a higher Si concentration and which can, therefore, be diluted to run multiple times.

To clarify this point, in the revised manuscript, the statement "Analytical replicates were conducted where sample volume allowed…" has been revised to "Analytical replicates of the standard sample bracketing procedure were conducted where sample volume allowed…".

**L96: "sampling of the standard (diatomite)" What does sampling mean here? Was the diatomite measured on several days? Were the columns prepared again? Was it prepared again?**
**Response:** We are sorry for the confusion caused by the incorrect word. In the revised manuscript, the phrase "…repeated sampling of the standard (diatomite)…" has been revised to "…repeated analysis of the standard (diatomite)…".

**L98: What do you mean by 2σ absolute?**
**Response:** We are sorry for the confusion. The Greek letter σ (sigma) is the symbol for standard deviation. To ensure consistency in the abbreviation of standard deviation throughout the manuscript, the statement "All uncertainties are reported at 2σ absolute" has been revised to "All uncertainties are reported at 2 standard deviations (2 SD)" in the revised manuscript.

**L98-100: I don't see the value of adding the error of the diatomite to your analytical uncertainty. The diatomite and other standards are mainly measured to see if you get the correct values. If at all you can use the diatomite value to normalize your values. But why would you add the uncertainty?**

**Response:** Running Diatomite throughout the analytical session allows us to calculate excess variance i.e. measurements of Diatomite represent a single population, which should yield an MSWD (chi-square) of 1. Values much above 1 indicate the analytical error alone does not adequately capture the session variance and, as such, the addition of an excess is necessary.

**L100: Reproducibility and instrument accuracy are generally indicated based on the NBS28, I don't see any values for the NBS28 at all.**

**Response:** Thanks for the reviewer's comment. NBS28 is used as the primary reference against which all other samples are normalised. Therefore, the normalised value derived for the secondary reference material (Diatomite) is viewed as a reflection of the accuracy of the reproducibility and instrument.

**L107: Remove "for" after "To assess".**

**Response:** We appreciate reviewer's careful review. The word "for" after "To assess" has been removed in the revised manuscript.

**Figure 3: The numbers in the figure and species names in the legend are hard to read. I suggest increasing the size of the figure for publication.**

**Response:** Thanks for the reviewer's suggestion. In the revised manuscript, we have increased the size of Figure 3. Furthermore, we have also increased the front size of the numbers within the figure from 4 pt to 7 pt, and the front size of species names in the legend from 5 pt to 7 pt.

L125: Change to "constituents".

**Response:** Yes, the term "constitutes" has been replaced with "constituents".

L126: Remove "all".

**Response:** Yes, the word "all" has been removed. Furthermore, according to the comment from another reviewer, the statement in the manuscript "The concentrations of other elements (Fe, Al, Mg, Sr, Na, K and Cl) were all below the minimum detectable limit for the EDS detector (Figure 4)" has been revised to "other elements (Fe, Al, Mg, Sr, Na, K and Cl) flagged in red within the EDS (A" and B") spectrum images (Figure 5) indicates that either their concentrations were below the minimum detectable limit (~ 0.1 %) for the EDS detector, or the software lacked a 99% statistical confidence in the presence of these elemental signatures within the analysed sample".

**Figure 4: I am wondering if samples have been checked at higher magnification as well. If radiolaria are not completely clean, contaminants are found within the radiolarian structure and are not visible at this resolution with EDS. These would rather be seen by looking closer at individual specimens. Also, what contaminations would the authors have expected to see with the help of the EDS and SEM here that would affect the silicon isotope composition? If the main goal was to check for the effects of dissolution, I would have expected a much**

**higher magnification of the SEM to be able to see if there were traces of dissolution or remineralization. which is hard with these pictures. Contamination by other Si phases, such as diatoms and sponges, is already visible with the light microscope. The SEM and EDS with this kind of use only show that there seems to be no clay contamination, which would also be better seen at higher magnification.**

**Response:** Thanks for the reviewer's constructive comment. Yes, the extracted radiolarian tests have also been examined at higher magnification. We did not include high-magnification images of the radiolarian shells in Figure 4 in the current manuscript, because images derived from the SEM and EDS analyses were primarily employed to further ascertain the potential presence of any extraneous siliceous phases, such as silicate detritus that potentially affect the silicon isotope composition of the radiolarians. Even if contaminants specifically from siliceous detrital material are present within the internal structures of the radiolarian shells, they can be identified using EDS analysis. The numerous pores on the surface of radiolarian shells allowed the electron beam to penetrate through these pores during EDS mapping analysis, thereby enabling the identification of siliceous detrital material within the shell based on elemental composition.

The reviewer is correct to note that potential contamination from diatoms and sponge spicules could be assessed using light microscopy. However, some other contaminations from silicate detritus, such as clay mineral particles (typically less than 2 μm in size), which could adhere to or fill the radiolarian shells, are challenging to discern clearly even with high-magnification (400x) light microscopy. We, therefore, conducted EDS analysis on extracted radiolarian tests, as silicate detritus typically contains aluminium (Al) and iron (Fe), and can be identified by elemental composition via EDS mapping. Consequently, we used both light microscopy and SEM coupled with an EDS detector to evaluate the purity of radiolarian tests, as stated in lines 81-84 of the manuscript.

Based on the reviewer's valuable comment, we have included a new Figure (Figure 4, please refer to the figure in the response to main question 3 above) in the revised manuscript to indicate the preservation state of radiolarian shells and the potential for dissolution-induced alteration of the radiolarian skeleton (the original Figure 4 and 5 has been renumbered as Figures 5 and 6, respectively, in the revised manuscript). Consequently, we have made following revisions to the manuscript as detailed in the response to main question 3 above:

In the revised manuscript, the title of section 2.2 "Radiolarian composition analysis" has been revised to "Radiolarian composition and preservation analysis", and we have included a second paragraph regarding assessing the preservation of radiolarian shells: "To assess the preservation of radiolarian shells both within the water column and seabed sediments, the proportion of fractured radiolarian shells was quantified in each studied sample under a Nikon optical microscope at x100 magnification. Further scanning electron microscopy (SEM) was employed to examine the potential for dissolution-induced alteration of the radiolarian skeleton.".

Following the first paragraph of section 2.3, we have included the result of radiolarian preservation analysis and a new Figure 4 (see below): "The proportion of fractured shells was low in the studied samples (Figure 4), generally ranging from 2 to 5% (mean = 3%) in plankton samples and from 3 to 9% (mean = 6%) in surface sediments. SEM images revealed a typical morphology of the pores on the radiolarian shell (Figure 4E), along with a high degree of integrity in the delicate skeletal structures (Figure 4F). These observations suggest good preservation of radiolarian shells, with no significant dissolution evident in either the water column or the

sediments."

The first paragraph of Section "4.2 Transfer of radiolarian $\delta^{30}$Si signatures into the sediment record" has been revised to "At each sampling station, $\delta^{30}$Si$_{rad}$ compositions (mean = 1.73‰) in the surface sediment closely resembles those (mean = 1.74‰) in the overlying water column evidenced by the paired t-test (p=0.75), indicating a faithful transfer of the $\delta^{30}$Si signal incorporated into radiolarian skeletons from the water column to sediments. This suggests that dissolution has a minimal impact on the $\delta^{30}$Si$_{rad}$ signatures as radiolarian shells sink through the water column and become incorporated into the sediment record. One of two possibilities may account for this observation: 1) the radiolarian shells may not have undergone substantial dissolution during sinking; 2) the radiolarian shells have experienced substantial dissolution, but this process may not significantly alter their isotope composition. Considering minor differences in the mean proportion of fractured radiolarian shells between the plankton sample (~3%) and surface sediments (6%), the well-preserved state of radiolarian shells (Figure 4), and the approximate correspondence between prominent radiolarian species identified in plankton samples and those present in surface sediments at each sampling station (as detailed in section 4.1), we propose that radiolarian shells have not experienced substantial dissolution or remineralisation during their transfer from the water column to the sediments. "

At the beginning of the second paragraph of section "4.2 Transfer of radiolarian $\delta^{30}$Si signatures into the sediment record", we have added "The susceptibility of radiolarian shells to dissolution varies considerably with different taxonomic groups due to the differences in the chemical constituent of their skeletons."

**References:**

Murray, J. W., Alve, E.: Natural dissolution of modern shallow water benthic foraminifera: taphonomic effects on the palaeoecological record, Palaeogeography, Palaeoclimatology, Palaeoecology, 146(1-4), 195-209, https://doi.org/10.1016/S0031-0182(98)00132-1, 1999.

Ragueneau, O., Tréguer, P., Leynaert, A., Anderson, R. F., Brzezinski, M. A., DeMaster, D. J., Dugdale, R. C., Dymond, J., Fischer, G., Francois, R., Heinze, C., Maier-Reimer, E., Martin-Jézéquel, V. Nelson D. M., Quéguiner, B.: A review of the Si cycle in the modern ocean: recent progress and missing gaps in the application of biogenic opal as a paleoproductivity proxy, Global. Planet. Change, 26(4), 317-365, https://doi.org/10.1016/S0921-8181(00)00052-7, 2000.

Ryves, D. B., Juggins, S., Fritz, S. C., Battarbee, R. W.: Experimental diatom dissolution and the quantification of microfossil preservation in sediments. Palaeogeography, Palaeoclimatology, Palaeoecology, 172(1-2), 99-113, https://doi.org/10.1016/S0031-0182(01)00273-5, 2001.

Treguer, P., Nelson, D. M., Van Bennekom, A. J., DeMaster, D. J., Leynaert, A., Quéguiner, B. The silica balance in the world ocean: a reestimate, Science, 268(5209), 375-379, https://doi.org/10.1126/science.268.5209.375, 1995.

**Figure 5: $\Delta^{30}$Si is usually referred to as the fractionation factor, so its use here is a bit confusing. Also, (A) already indicates that the data are the same within error, so I see little use in plotting the difference here as they are within your analytical uncertainty.**

**Response:** We are sorry for the confusion from the use of the triangle symbol "$\Delta$". Yes, the reviewer is correct to point out that $\Delta^{30}$Si is usually used as the fractionation factor. In this context, the symbol "$\Delta$" represents the uppercase of Greek letter Delta when denoting the fractionation factor. To distinguish this from the "$\Delta$" used for the fractionation factor, we, in fact, employed a

triangle symbol "Δ" in Figure 5B to represent the differences in $\delta^{30}Si_{rad}$ values between plankton samples and surface sediments at each sampling station. While similar in appearance, these two symbols are actually distinct.

To avoid this confusion, we have replaced the triangle symbol "Δ" with the capital letter "D" of "Different" in Figure 5B of the manuscript.

Yes, the data presented in Figure 5A are consistent within the reported errors. However, these data points, representing the mean $\delta^{30}Si_{rad}$ values measured by the Neptune ICP-MS, also exhibit discrepancies between paired $\delta^{30}Si_{rad}$ data. Figure 5B provides the differences, or shifts, in the mean values of $\delta^{30}Si_{rad}$ as radiolarian shells are transferred from the water column to the sediment, thereby offering a supplementary perspective to the information presented in Figure 5A. Consequently, we respectfully suggest retaining Figure 5B in the revised manuscript.

**L134-140: The discussion starts with a comparison of the absolute $\delta^{30}Si_{rad}$ values measured in this study with previous studies. While it is important to note that the range in silicon isotopes measured here agrees with previous studies, I don't see a value in solely comparing absolute values between regions here. While this can serve as an introduction to the discussion, a comparison of the impact of source signatures of $\delta^{30}Si$-DSi and potential fractionation factors of radiolaria is missing from the discussion so far.**

**Response:** Thanks for reviewer's comment. We have presented a comparison of the absolute radiolarian silicon isotope ($\delta^{30}Si_{rad}$) values obtained in this study with those from previous studies. This is because absolute $\delta^{30}Si_{rad}$ values documented herein, and a comparison with those from other regions, would provide a fundamental dataset for investigating the range of $\delta^{30}Si_{rad}$ composition and its spatial distribution in the oceans. This may advance the understanding of potential relationships between $\delta^{30}Si_{rad}$ composition and both dissolved silicon concentrations and $\delta^{30}Si$ of the dissolved silicon ($\delta^{30}Si_{DSi}$) in the surrounding seawater.

The fractionation between $\delta^{30}Si_{rad}$ and dissolved silicon isotopes ($\delta^{30}Si_{DSi}$), expressed as the apparent Si isotope fractionation factor ($\Delta^{30}Si_{rad} \sim {}^{30}\varepsilon = \delta^{30}Si_{rad} - \delta^{30}Si_{DSi}$), is an important parameter for understanding to what extent the radiolarians fractionate the $\delta^{30}Si_{DSi}$ during Si absorption, and a potential proxy for reconstructing the silicon cycle in mid-upper depth waters. Previous published data show that the $\delta^{30}Si(OH)_4$ values in the upper water column (above 100 m) of the northern South China Sea (SCS) range from 1.33 to 2.94 ‰, with a mean of 2.3 ‰ (Cao et al., 2012). Based on $\delta^{30}Si_{rad}$ compositions in plankton samples and surface sediments from our study, and a mean $\delta^{30}Si(OH)_4$ values in the northern SCS from Cao et al., (2012), the radiolarian fractionation factors can be calculated to range from −0.45 to −0.74‰, with a mean of -0.56‰. The mean $\Delta\delta^{30}Si_{rad}$ value calculated in our study is more positive than the radiolarian fractionation factor (−1.5‰) applied by Hendry et al. (2014), but is close to the factor (-0.8‰) reported by Abelmann et al. (2015) and that (-0.62‰, mixed radiolaria) reported by Doering et al. (2021).

As stated in the response to the main question 1, $\Delta\delta^{30}Si_{rad}$ values are not presented in this study for the following reasons: 1) the primary objective of the present study was to determine the depth interval from which the radiolarian community contributes to the $\delta^{30}Si_{rad}$ signature in the water column, and to ascertain whether and why radiolarian silicon isotopes ($\delta^{30}Si_{rad}$) signatures are faithfully transferred from the water column to the sediments. Consequently, our focus has been on comparing prominent radiolarian compositions at various water depths at each station, and $\delta^{30}Si_{rad}$ values between the water column and sedimentary record at individual stations. The

calculation of $\Delta^{30}Si_{rad}$ is not essential for this specific investigation; 2) although Cao et al. (2012) have provided a $\delta^{30}Si(OH)_4$ dataset for the northern SCS, the spatial overlap with our sampling regime is not sufficient to permit an accurate constraint of $\Delta^{30}Si_{rad}$ and a meaningful discussion regarding the impact of $\delta^{30}Si_{DSi}$ signatures on $\delta^{30}Si_{rad}$ compositions. Actually, at each sampling station, when collecting plankton samples and surface sediments, we have also collected water samples from the relevant depth range for the $\Delta^{30}Si_{rad}$ constraint. Considering the relatively low $Si(OH)_4$ concentrations in these water samples from the upper water column, we plan to use a Neoma MC-ICP-MS for the silicon isotopic analysis of dissolved silicon because of its enhanced sensitivity and mass resolution. However, isotopic analyses for seawater samples are pending due to ongoing instrument-related issues with the Neoma MC-ICP-MS at the Geochronology and Tracers Facility, British Geological Survey. Once these data are obtained, a more robust fractionation factor can be determined; and 3) A separate manuscript detailing the radiolarian fractionation factor and its implication for nutrient levels in the mid-upper water column is to be prepared in the near future.

**L152: Correct to "all the radiolarian shells".**

**Response:** The phrase "the all radiolarian shells" in the manuscript has been revised to "all the radiolarian shells".

**L 153: Why did you expect the $\delta^{30}Si_{rad}$ values to differ between depths? (100-300m versus 0-100m).**

**Response:** A previous study documented variations in the $\delta^{30}Si$ composition among different radiolarian taxa (Doering et al., 2021). Based on findings regarding the vertical distribution of living radiolarians in the water column of the South China Sea using protoplasm staining, Hu et al., (2015) demonstrated the variations in both radiolarian abundance and dominant species composition between the 0-75 m and 75-300 m water layers. Consequently, it is expected that $\delta^{30}Si_{rad}$ values from plankton samples at 100-300 m water depth would potentially differ from those at 0-100 m.

In the revised manuscript, we have briefly explained why $\delta^{30}Si_{rad}$ values from plankton samples at 100-300 m water depth were expected to potentially differ from those at 0-100 m. The statement in the manuscript "Whilst δ30Sirad values from plankton samples at 100-300 m water depth were expected to potentially differ from those at 0-100 m, no significant difference was detected (p = 0.52)." has been revised to "Given the differences in both the abundance and dominant species composition of living radiolarians between 0-75 m and the 75-300m water depth in the SCS (Hu et al., 2015), it is expected that $\delta^{30}Si_{rad}$ values from plankton samples at 100-300 m water depth might potentially differ from those at 0-100 m. However, no significant difference was detected (p = 0.52)."

**References:**

Doering, K., Ehlert, C., Pahnke, K., Frank, M., Schneider, R., Grasse, P.: Silicon isotope signatures of radiolaria reveal taxon-specific differences in isotope fractionation, Front. Mar. Sci., 8, 666896, https://doi.org/10.3389/fmars.2021.666896, 2021.

Hu, W. F., Zhang, L. L., Chen, M. H., Zeng, L. L., Zhou, W. H., Xiang, R., Zhang, Q., Liu, S. H.: Distribution of living radiolarians in spring in the South China Sea and its responses to environmental factors, Sci. China Earth Sci., 58, 270-85, https://doi.org/10.1007/s11430-014-4950-0, 2015

L170: Remove "(vital effects)" here.

**Response:** Yes, the phrase "(vital effects)" has been removed.

**L169-172: I don't think this discussion is correct here. As far as I can see, there is no significant difference in species compositions or $\delta^{30}Si_{rad}$ between your water column and surface sediment data (line 153-159). So even if there is a difference in d30Sirad and/or fractionation between radiolarian species/order, there is no way to see any effect of this in your data.**

**Response:** While statistical analysis suggests no significant difference in either the composition of prominent species or their relative abundances between plankton samples and surface sediments, variations in the relative abundances of some prominent radiolarian species are indeed observed. Due to the high diversity of radiolarians in low-latitude oceans, few species comprise more than 10% of the radiolarian community, and those exceeding 2-3% are generally defined as prominent species (e.g. Zhang et al., 2009; Hu et al., 2015). Therefore, a discernible difference in the relative abundance of a prominent species is considered to exist when the variation in its relative abundance between samples reaches approximately or exceeds a factor of two.

Radiolarian census data in this study showed that the relative abundances of certain prominent radiolarian species, including *Botryocyrtis scutum, Didymocyrtis tetrathalamus t.*, *Tetrapyle octacantha/quadribola*, *Zygocircus capulosus*, varied by near a factor of two or more between the water column (0-100) and surface sediments (please see table A below and Figure 4 in the manuscript; original census data will also be provided as supplementary material in the revised manuscript) at some stations (e.g. station 13). However, despite variations in the relative abundance of certain prominent radiolarian species, no substantial differences in $\delta^{30}Si_{rad}$ values were observed between plankton samples and the sediment at each sampling station. We therefore propose two potential reasons for the consistency of $\delta^{30}Si_{rad}$ values between plankton samples and the sediments, as detailed in lines 169-175 of the current manuscript: 1) the minor differences in the relative abundance of most prominent taxa between water column and sediment samples; 2) the high diversity of radiolarians in each sample averaging out the $\delta^{30}Si_{rad}$ signal across different taxa, mitigating potential bias from individual taxa.

Table A Variations in the relative abundance of typical prominent species between the water column (0-100) and surface sediments at station 13

| Sample
Prominent species | Station 13 (0-100 m) | Station 13 (surface sediment) |
|---|---|---|
| *Tetrapyle octacantha/quadribola* | 4.0 % | 8.8 % |
| *Didymocyrtis tetrathalamus t.* | 8.1 % | 2.4 % |
| *Tetrapyle octacantha/quadribola* | 11.7 % | 18.1 % |
| *Zygocircus capulosus* | 2.2 % | 6.2 % |

We hope this explanation adequately addresses the reviewer's concerns. In the revised manuscript, the statement in lines 164-165 of the manuscript "Although minor differences in the relative abundances of radiolarian species are observed…" has been revised to "Although discernible differences in the relative abundances of some prominent radiolarian species are observed…". Moreover, the statement in lines 169-172 "Although variations in the $\delta^{30}Si$ composition have been documented among different radiolarian taxa (Doering et al., 2021), the

relative abundance differences between plankton and surface sediment samples do not result in significant $\delta^{30}Si_{rad}$ disparities in our study in the SCS" has been revised to "Although variations in the $\delta^{30}Si$ composition have been documented among different radiolarian taxa (Doering et al., 2021), the relative abundance differences in certain prominent species, such as *Botryocyrtis scutum, Didymocyrtis tetrathalamus t., Tetrapyle octacantha/quadribola*, and *Zygocircus capulosus*, between plankton and surface sediment samples do not result in significant $\delta^{30}Si_{rad}$ disparities in our study in the SCS".

**L180-215: Section 4.2 Transfer of radiolarian d30Si signatures into the sediment record need some re-structuring and rewriting. The first paragraph is fine, picking up on the fact that there is little difference between the water column and the sediment d30Sirad. The following paragraphs, however, are mainly a summary of the literature and should be shortened, and the information from the radiolarian abundances investigated here should be more highlighted. Additionally, references to the SEM and EDS analysis conducted are not referred to at all here.**

**Response:** As responded to the question 3, Section "4.2 Transfer of radiolarian $\delta^{30}Si$ signatures into the sediment record" focuses on explaining why the $\delta^{30}Si_{rad}$ signal from the water column is faithfully transferred to the sediments. Our data indicate a good consistency between $\delta^{30}Si_{rad}$ signatures in the water column and the sedimentary record, suggesting that dissolution may have a minimal impact on the $\delta^{30}Si_{rad}$ composition as shells sink through the water column and become incorporated into the sediment record. This may be attributable to one of two possible reasons: 1) the radiolarian shells may not have undergone substantial dissolution during sinking; 2) the radiolarian shells have experienced substantial dissolution, but this process might not have significantly altered their isotope composition. As shell fracture of microfossils preserved within sediments is commonly attributed to partial dissolution (e.g. Murray and Alve, 1999; Ryves et al., 2001), the proportion of fractured radiolarian shells may be indicative of the potential for radiolarian shell dissolution during sinking. In this study, the proportion of fractured radiolarian shells was low, and showed only slight difference between plankton samples (mean = 3%) and surface sediments (mean = 6%) (Figure 4). Considering these minor differences in the mean proportion of fractured radiolarian shells between the plankton sample and surface sediments, the well-preserved state of delicate radiolarian skeletons (Figure 4), and the correspondence between radiolarian species identified in plankton samples and those present in surface sediments at each sampling station (as detailed in section 4.1), we propose that radiolarian shells have not experienced substantial dissolution or remineralisation during their transfer from the water column to the sediments.

The susceptibility of radiolarian shells varies considerably with different taxonomic groups due to notable differences in the chemical constituent of their skeletons. For example, Collodaria radiolarians, a group belonging to polycystine radiolarians, are highly susceptible to dissolution during sinking due to their skeletons composed of both opal and Celestine (Afanasieva et al., 2005; Afanasieva and Amon, 2014). However, as stated above, our observations do not provide evidence that the radiolarian shells have experienced substantial dissolution during sinking within the water column. Only by clarifying the dissolution characteristics of different radiolarian groups which are closely related to the chemical compositions of their shells, alongside the approximate proportion of each group within the radiolarian community of our studied samples, can we better understand

why the radiolarian shells are minimally impacted by dissolution during sinking through the water column. This allows for a compelling explanation of why the radiolarian silicon isotope signal from the water column is faithfully transferred to the sediments. Therefore, the information presented in Section 4.2 regarding the radiolarian taxonomy, chemical composition and dissolution characteristics of each taxonomic group, and the proportion of each taxonomic group within the radiolarian community in our analyzed samples, is essential for this study, despite containing a summary of the literature.

The information of the radiolarian abundances in various water depths was primarily used to address the depth range from which the radiolarian community contributes to the $\delta^{30}Si_{rad}$ signature in the water column and sediments, and has been detailed in Section 4.1 as stated in the response to the main question 3.

The SEM and EDS images in the original Figure 4 were primarily employed to examine the purity and cleanliness of the radiolarian tests extracted from the plankton samples and surface sediments, thereby ensuring the reliability of the $\delta^{30}Si_{rad}$ data in this study. As such, the results of the SEM and EDS analysis were not referred in Section 4.2 of the original manuscript. However, we agree with the reviewer that the SEM images with a higher magnification may trace the potential dissolution of radiolarian shells as reviewer suggested above. Therefore, we have included a new Figure (Figure 4, please refer to the figure in the response to main question 3 above) in the revised manuscript to indicate the preservation state of radiolarian shells and the potential for dissolution-induced alteration of the radiolarian skeleton (the original Figures 4 and 5 have been renumbered as Figures 5 and 6, respectively, in the revised manuscript). Accordingly, we have made following revisions to the manuscript as detailed in the response to main question 3 above:

In section 2.2 of the revised manuscript, we have included a paragraph to assess the preservation of radiolarian shells following the line 78 "To assess the preservation of radiolarian shells both within the water column and seabed sediments, the proportion of fractured radiolarian shells was quantified in each studied sample under a Nikon optical microscope at x100 magnification. Further scanning electron microscopy (SEM) was employed to examine the potential for dissolution-induced alteration of the radiolarian skeleton.".

Following the first paragraph of section 2.3, we have included the result of radiolarian preservation analysis "The proportion of fractured shells was low in the studied samples (Figure 4), generally ranging from 2 to 5% (mean = 3%) in plankton samples and from 3 to 9% (mean = 6%) in surface sediments. SEM images revealed a typical morphology of the pores on the radiolarian shell (Figure 4E), along with a high degree of integrity in the delicate skeletal structures (Figure 4F). These observations suggest good preservation of radiolarian shells, with no significant dissolution evident in either the water column or the sediments."

The first paragraph of Section "4.2 Transfer of radiolarian $\delta^{30}Si$ signatures into the sediment record" has been revised to "At each sampling station, $\delta^{30}Si_{rad}$ compositions (mean = 1.73‰) in the surface sediment closely resembles those (mean = 1.74‰) in the overlying water column evidenced by the paired t-test (p=0.75), indicating a faithful transfer of the $\delta^{30}Si$ signal incorporated into radiolarian skeletons from the water column to sediments. This suggests that dissolution has a minimal impact on the $\delta^{30}Si_{rad}$ signatures of radiolarians as shells sink through the water column and become incorporated into the sediment record. One of two possibilities may account for this: 1) the radiolarian shells may not have undergone substantial dissolution during

sinking; 2) the radiolarian shells have experienced substantial dissolution, but this process might not have significantly altered their isotope composition. As shell fracture of microfossils preserved within sediments is commonly attributed to partial dissolution (e.g. Murray and Alve, 1999; Ryves et al., 2001), the proportion of fractured radiolarian shells may be indicative of the potential for radiolarian shell dissolution during sinking. In the present study, the proportion of fractured radiolarian shells was low, and showed only slight difference between plankton samples (mean = 3%) and surface sediments (mean = 6%) (Figure 4). Considering these minor differences in the mean proportion of fractured radiolarian shells between the plankton sample and surface sediments, the well-preserved state of radiolarian shells (Figure 4), and the approximate correspondence between prominent radiolarian species identified in plankton samples and those present in surface sediments at each sampling station (as detailed in section 4.1), we propose that radiolarian shells have not experienced substantial dissolution or remineralisation during their transfer from the water column to the sediments. "

At the beginning of the second paragraph of section "4.2 Transfer of radiolarian $\delta^{30}$Si signatures into the sediment record", we have add "The susceptibility of radiolarian shells to dissolution varies considerably with different taxonomic groups due to the differences in the chemical constituent of their skeletons."

**L181: Change to "resemble".**
Response: Yes, the term "resembles" has been revised to "resemble".

**L201: Add a comma after "in the SCS".**
Response: Yes, we have included a comma after "in the SCS" in Line 201 of the current manuscript.

**L206: Change to "in this study,…".**
Response: Yes, the phrase "in the current study…" in line 206 of the manuscript has been revised to "in this study,…".

**L209: Correct to "expected to have..".**
Response: Yes, the phrase "dissolution is expected have…" in Line 209 has been revised to "dissolution is expected to have…".

---

## Author Response (AR1)

Dear Editor and Reviewers:

Thank you for your careful review and constructive comments to our manuscript "Faithful transfer of radiolarian silicon isotope signatures from water column to sediments in the South China Sea". These comments are valuable and highly beneficial in improving the quality of our paper. We have carefully read each comment and revised the manuscript accordingly, providing a detailed explanation for each issue raised. We sincerely appreciate the diligent efforts of the Editor/Reviewers. The revised texts of the manuscript have been highlighted in red using track changes in the Author's track-changes file. Our responses to the editor's and reviewers' comments, along with revision notes and all relevant changes (**in blue**) are presented as follows:

| | |
|---|---|
| Responses to the Editor's comments with revision notes | p. 1-2 |
| Responses to the Reviewer 1's comments with revision notes | p. 3-18 |
| Responses to the Reviewer 2's comments with revision notes | p. 19-40 |

**Responses to the Editor's comments with revision notes**

**Dear authors, both reviewers indicate that the manuscript represent a solid contribution to the measurement of radiolarian dSi isotopes, and determining the water depth at which radiolaria reside. However, both reviewers also comment that the manuscript should be substantiated with a discussion on the fractionation factor of the radiolaria. The authors propose that this can be included in a follow-up manuscript, but I agree with the reviewers that the absence of this fractionation factor makes the currently submitted study less relevant for the readership of Biogeosciences. I suggest that the authors add a discussion section on constraining the fractionation factor, potentially using the anticipated measurements on the Neoma MC-ICP-MS, or resubmit the current manuscript to a journal that focuses on methodology.**

**Response:** We are appreciative of the editor's valuable comment. In the revised manuscript, we have added a new Section "4.2 Radiolaria silicon isotope fractionation" regarding the discussion on the fractionation factor of the radiolarian, as detailed below (Please see lines 244-266 in the Author's track-changes file):

"4.2 Radiolaria silicon isotope fractionation
The isotope fractionation factor between radiolaria and seawater is a critical parameter for reconstructing past silicon cycle in the mid-upper ocean using $\delta^{30}Si_{rad}$ in downcore records (Hendry et al., 2014; Abelmann et al., 2015; Doering et al., 2021). As discussed previously, the $\delta^{30}Si_{rad}$ signatures in the water column of the SCS are predominantly contributed to by radiolarians from the 0-100 m depth layer. Therefore, $\delta^{30}Si_{sw}$ data pertinent to constraining silicon isotope fractionation by radiolarians in the SCS should derived primarily from this depth interval. Currently, there is no corresponding seawater $\delta^{30}Si$ ($\delta^{30}Si_{sw}$) data from the stations in the SCS where water column radiolaria samples were collected. Previous work, however, has measured seasonal changes in $\delta^{30}Si_{sw}$ in the northern SCS (Cao et al 2012). At the two sites from that study that are off the continental shelf (A10 and SEATS), $\delta^{30}Si_{sw}$ above 100 m varies from 1.4‰ to 2.9‰. Assuming: 1) these values and their depth profiles are broadly comparable to stations where $\delta^{30}Si_{rad}$ was measured; and 2) that radiolarians construct their skeletons in equilibrium with

ambient seawater (Abelmann et al., 2015, Fontorbe et al., 2016, Doering et al 2021), the isotope fractionation factor (ε) can be calculated as:

$$\varepsilon \sim \Delta^{30}Si = \delta^{30}Si_{rad} - \delta^{30}Si_{sw} \tag{2}$$

Using Monte Carlo simulations (10,000 replicates) to account for the 2SD uncertainty of both $\delta^{30}Si_{rad}$ and $\delta^{30}Si_{sw}$, ε is estimated to range from –0.33‰ to –0.92‰ (mean = 0.58‰) (Table S1 in supplementary material 1). Due to the difficulties in culturing radiolarian (e.g. Suzuki and Not, 2015), relatively little is known about the fractionation factor for $\delta^{30}Si_{rad}$. Values for ε obtained here from the SCS are comparable to those measured by Abelmann et al. (2015) in the Southern Ocean (–0.54‰ to –0.91‰, mean = -0.75‰) and by Doering et al. (2021) along the Peruvian Shelf (–0.35‰ to –0.79‰, mean = -0.62‰, mixed radiolaria). The large range of values for ε has been suggested to indicate species/order-specific fractionation during the uptake of silicic acid by radiolaria (Doering et al., 2021). Such a process would necessitate the extraction of species-specific radiolarian samples for isotope analysis in future palaeoceanographic research (e.g., Zhang and Swann, 2023). However in this current study from the SCS, whilst the species composition of the $\delta^{30}Si_{rad}$ samples is known (Figure 3), an absence of matching $\delta^{30}Si_{sw}$ data from exactly the same sampling stations prevents this issue being further investigated at this time."

In addition, the phrase "with the fractionation factor for $\delta^{30}Si_{rad}$ varying from –0.33‰ to –0.92‰ (mean = 0.58‰)" has been added into both the "Abstract" (Please see lines 7-8 in the Author's track-changes file) and Section "5 Conclusions". (Please see lines 319-320 in the Author's track-changes file)

The $\delta^{30}Si_{sw}$ data used to constrain the fractionation factor for $\delta^{30}Si_{rad}$ were sourced from an existing dataset for the northern SCS published by Cao et al., (2012). This is because isotopic analyses of the seawater samples collected from our studied stations are still pending, due to ongoing instrument-related issues with the Neoma MC-ICP-MS at the Geochronology and Tracers Facility, British Geological Survey, and consequently, the availability of these data remains uncertain.

**Reference:**

Abelmann, A., Gersonde, R., Knorr, G., Zhang, X., Chapligin, B., Maier, E., Esper, O., Friedrichsen, H., Lohmann, G., Meyer, H., Tiedemann, R.: The seasonal sea-ice zone in the glacial Southern Ocean as a carbon sink, Nat. Commun., 6(1), 8136, https://doi.org/10.1038/ncomms9136, 2015.

Cao, Z., Frank, M., Dai, M., Grasse, P., Ehlert, C.: Silicon isotope constraints on sources and utilization of silicic acid in the northern South China Sea, Geochimica et Cosmochimica Acta, 97, 88-104, https://doi.org/10.1016/j.gca.2012.08.039, 2012.

Doering, K., Ehlert, C., Pahnke, K., Frank, M., Schneider, R., Grasse, P.: Silicon isotope signatures of radiolaria reveal taxon-specific differences in isotope fractionation, Front. Mar. Sci., 8, 666896, https://doi.org/10.3389/fmars.2021.666896, 2021.

Fontorbe G, Frings P J, De La Rocha, C., Hendry, K. R., Conley, D. J.: A silicon depleted North Atlantic since the Palaeogene: Evidence from sponge and radiolarian silicon isotopes, Earth Planet. Sci. Lett., 453, 67-77, https://doi.org/10.1016/j.epsl.2016.08.006, 2016.

Suzuki, N., Not, F: Biology and Ecology of Radiolaria, In: Marine Protists, edited by: Ohtsuka, S., Suzaki, T., Horiguchi, T., Suzuki, N., Not, F., Springer, Tokyo, Japan, https://doi.org/10.1007/978-4-431-55130-0_8, 2015.

Zhang, Q., Swann, G. E. A.: An effective method to extract and purify radiolaria from tropical marine sediments, Front. Mar. Sci., 10, 1150518, https://doi.org/10.3389/fmars.2023.1150518, 2023.

**Responses to the Reviewer 1's comments with revision notes**

There is growing interest in how the silicon isotope composition (expressed as δ30Si) of radiolarians can be used to supplement/complement those of the more established proxy archives in diatoms and sponges. As with these two groups, there is a need to understand and account for any post-mortem alteration of the initial isotope signal. Here, Zhang et al compare bulk-assemblage radiolarian δ30Si values from water-column plankton tows and underlying coretop sediments. They demonstrate the two are statistically indistinguishable, lending more confidence to the use of radiolarian δ30Si as a window on to past Si cycling.

In general, this manuscript is very well written and the figures are clear (with possible exception of Fig.3). The referencing is generally appropriate – with some notable absences (see below). The data are generated by appropriate techniques (though some more details may be warranted). Overall, there is little to criticise in terms of the central conclusion – that the radiolarian δ30Si signal is not altered during water column sinking – which is supported by the data (though I have a series of minor comment/suggestions that I detail below). Nevertheless, this is a relatively small dataset and I have the impression that with a slightly expanded dataset much more could be done. I note in the supplement, Fig. S1 contains 22 'unpublished' radiolarian datapoints. By integrating these, and dissolved Si δ30Si data, a much more impactful paper would result. Some suggestions are below, but this is ultimately an editorial decision.

**Response:** We appreciate the reviewer's insightful comments. Yes, we have included 22 'unpublished' radiolarian $\delta^{30}Si$ data in Fig. S1 of the Supplement Material 1. Given that the primary objective of the present study was to ascertain whether and why radiolarian silicon isotopes ($\delta^{30}Si_{rad}$) signatures are faithfully transferred from the water column to the sediments, combined $\delta^{30}Si_{rad}$ records from paired water column and surface sediment samples are required. However, the radiolarian tests used to obtain these 22 data were extracted from surface sediments, and lack matching $\delta^{30}Si_{rad}$ data from the overlying water column (although plankton samples were collected from some of these 22 stations, the quantity of radiolarian tests was insufficient for isotope analysis). As these 22 data were measured within the same analytical batch as samples used in this study, we included them in the Supplement Material to demonstrate that $\delta^{29}Si$ and $\delta^{30}Si$ values of radiolarian tests fall on the expected mass-dependent fractionation line $\delta^{29}Si = 0.51 \times \delta^{30}Si$ (Reynolds et al., 2006) (Figure S1 in the Supplementary Material), thereby indicating the effective removal of all polyatomic interferences during measurement.

**Reference:**

Reynolds, B. C., Frank, M., Halliday, A. N.: Silicon isotope fractionation during nutrient utilization in the North Pacific, Earth Planet. Sci. Lett., 244(1-2), 431–443, https://doi.org/10.1016/j.epsl.2006.02.002, 2006.

**Minor comments and suggestions**

1. **L16-27 – the prominence of the SALH in the introduction is a bit strange to me, considering it's not the focus of the manuscript (and isn't returned to)**

**Response:** In the introduction, we referenced the "silica acid leakage hypothesis" (SALH) due to its emphasis on the influence of changes in dissolved silicon (DSi) concentrations between the Southern Ocean and low-latitude regions on atmospheric $p$CO$_2$ and climate. While the SALH remains debated, partly due to inconsistent records of siliceous productivity and DSi concentrations in overlying waters across different low-latitude regions during the late Quaternary, we cite it here to highlight the importance of reconstructing past Si nutrient levels throughout the water column. This is expected to facilitate a more comprehensive understanding of the relationships between the Si cycle, biological pump efficiency, and global climate change.

2. **L48: Two papers that deserve citation/discussion here and elsewhere are Closset et al. 2015 (doi: 10.1002/2015GB005180) and Grasse et al. 2021 (doi: 10.3389/fmars.2021.697400) – both present a comparison of plankton tow diatoms and sediment trap (Closset) or core-top (Grasse) material, concluding that the transfer of biogenic silica from surface ocean to depth isn't associated with a resolvable change in δ30Si. See also Varela et al. 2004 (doi: 10.1029/2003GB002140; sediment trap data) and Fripiat et al. 2012 (doi: 10.5194/bg-9-2443-2012; water column biogenic silica data).**

**Response:** We appreciate the reviewer bringing these important references to our attention, and we have included them in the revised version (Please see lines 50-51, 391-393, 421-422, 425-426, and 509-510 in the Author's track-changes file). Indeed, the constancy of $\delta^{30}$Si of diatom biogenic silica ($\delta^{30}$Si$_{BSi}$) with depth has been documented for suspended particles in the Atlantic sector of the Southern Ocean (Fripiat et al., 2012), as well as in sediment traps in the Southern Ocean south of New Zealand (Varela et al., 2004) and Australian (Closset et al. 2015). Furthermore, a strong agreement has also been observed between $\delta^{30}$Si$_{BSi}$ from seawater samples and those from core-top sediments in the central upwelling region off Peru (Grasse et al. 2021). These findings suggest that the transfer of diatom biogenic silica from the surface ocean to sediments is not associated with a dissolution-driven alteration of $\delta^{30}$Si.

**References:**

Closset, I., Cardinal, D., Bray, S. G., Thil, F., Djouraev, I., Rigual−Hernández, A. S., Trull, T. W.: Seasonal variations, origin, and fate of settling diatoms in the Southern Ocean tracked by silicon isotope records in deep sediment traps. Global Biogeochemical Cycles, 29(9), 1495-1510. https://doi.org/10.1002/2015GB005180, 2015.

Fripiat, F., Cavagna, A. J., Dehairs, F., De Brauwere, A., André, L., Cardinal, D.: Processes controlling the Si-isotopic composition in the Southern Ocean and application for paleoceanography, Biogeosciences, 9(7), 2443-2457, https://doi.org/10.5194/bg-9-2443-2012, 2012.

Grasse, P., Haynert, K., Doering, K., Geilert, S., Jones, J. L., Brzezinski, M. A., Frank, M.: Controls on the silicon isotope composition of diatoms in the peruvian upwelling. Frontiers in Marine Science, 8, 697400, https://doi.org/10.3389/fmars.2021.697400, 2021.

Varela, D. E., Pride, C. J., Brzezinski, M. A.: Biological fractionation of silicon isotopes in Southern Ocean surface waters. Global biogeochemical cycles, 18(1). https://doi.org/10.1029/2003GB002140, 2004.

3. **L85 and introduction: In general, there is a growing awareness that 'bulk' assemblage $\delta^{30}$Si data have disadvantages as a paleo-archive, and that where possible single-species records are much stronger. Therefore it would be good to see some justification for why this was not attempted here.**

**Response:** We thank the reviewer for raising this important point. We agree with the reviewer that $\delta^{30}$Si records derived from single-species shells are generally preferred for paleo-archive studies, as the potential for differences in $\delta^{30}$Si fractionation may exist between various radiolarian species (e.g. Doering et al., 2021). However, obtaining sufficient material for silicon isotopic analysis from individual radiolarian taxa in our samples proved to be a significant challenge, necessitating our focus on the bulk radiolarian species record for this study.

We did, in fact, attempt to isolate single-species tests via manual picking under a microscope, but encountered several obstacles. Firstly, for silicon isotope analysis using a Thermo Fisher Scientific Neptune Plus MC-ICP-MS at the Geochronology and Tracers Facility, British Geological Survey, 1 to 1.5 mg of purified radiolarian tests is typically required. Radiolarian tests from sediments in the tropical Ocean are notably lightweight, averaging 0.063 to 0.136 mg/shell (Moore, 1969; Takahashi, 1982). Consequently, a mean of approximately 7,000 to 15,000 individual tests are required to achieve the ~1 mg of material needed for the silicon isotope analysis. Given the high diversity of radiolarians in Holocene sediments, particularly in low-latitude oceans (Moore, 1969; Takahashi, 1982; Boltovskoy et al., 2010) and in our samples, which average over 100 species per sample, coupled with the observation that most species typically constitute less than 10% of the total radiolarian community (Boltovskoy et al., 2010; Chen et al., 2008), this means that 70,000 to 150,000 bulk radiolarian tests are potentially required for each sample to obtain sufficient single-species material. Regrettably, the limited volume of the plankton samples made it difficult to obtain such a large number of radiolarian shells. Secondly, manually picking 7,000 to 15,000 individual tests out of 70,000 to 150,000 bulk radiolarian tests is highly time-consuming, rendering this approach impractical for the study of single-species radiolarian $\delta^{30}$Si. Furthermore, even with meticulous hand-picking under a microscope, the electro static adsorption of these light tests to transfer tools, such as brushes, inevitably results in test loss during transfer to storage vials. This further compounded the difficulty of accumulating the required quantity of monospecific tests for the isotope analysis.

It was these insurmountable challenges encountered during our attempts at hand-picking that prompted us to develop the method for extracting and purifying bulk radiolarian tests from the sediments for the study of the radiolarian $\delta^{30}$Si (Zhang and Swann, 2023). This method enabled us to effectively obtain sufficient pure radiolarian tests to conduct the comparative analysis of radiolaria $\delta^{30}$Si compositions using water column and surface sediment samples in this current study.

Based on the reviewer's comment, we have included a brief explanation in the revised manuscript regarding the absence of silicon isotopic analysis for individual radiolarian taxa: "Given the lightweight nature of shells and the high diversity of radiolarians, particularly in low-latitude oceans (Moore, 1969; Takahashi, 1982; Boltovskoy et al., 2010), combined with the observation that most species typically comprise less than 10% of the total radiolarian assemblage in both the water column and the sedimentary record (Boltovskoy et al., 2010; Chen et al., 2008), obtaining sufficient material from individual radiolarian taxa for silicon isotopic analysis remains a considerable challenge." (Please see lines 62-66 in the Author's track-changes file)

**References:**

Boltovskoy, D., Kling, S. A., Takahashi, K., Bjørklund, K.: World atlas of distribution of recent polycystina (Radiolaria), Palaeontologia Electronica, 13, 1–229, 2010.

Chen M H, Zhang L L, Zhang L L , et al. 2008a. Preservation of radiolarian diversity and abundance in surface

sediments of the South China Sea and its environmental implication. Journal of China University of Geosciences, 19: 217–229.

Doering, K., Ehlert, C., Pahnke, K., Frank, M., Schneider, R., Grasse, P.: Silicon isotope signatures of radiolaria reveal taxon-specific differences in isotope fractionation, Front. Mar. Sci., 8, 666896, https://doi.org/10.3389/fmars.2021.666896, 2021.

Moore, JR. T. C: Radiolaria: change in skeletal weight and resistance to solution. Geological Society of America Bulletin, 80(10), 2103-2108. https://doi.org/10.1130/0016-7606(1969)80[2103:RCISWA]2.0.CO;2, 1969

Takahashi, K.: Vertical flux, ecology and dissolution of Radiolaria in tropical oceans: implications for the silica cycle. PhD thesis. Woods Hole Oceanographic Institution and Massachusetts Institute of Technology, 1982

Zhang, Q., Swann, G. E. A.: An effective method to extract and purify radiolaria from tropical marine sediments, Front. Mar. Sci., 10, 1150518, https://doi.org/10.3389/fmars.2023.1150518, 2023.

**4. Section 2.3: I would suggest more detail is needed here. Specific suggestions include:**

**Define what is 'sufficient' radiolarian tests (L80; what is the typical mass of Si processed)**

**Response:** We are sorry for the lack of clarity regarding the typical mass of processed Si. For silicon isotope analysis using a Thermo Fisher Scientific Neptune Plus MC-ICP-MS at the Geochronology and Tracers Facility, British Geological Survey, 1 to 1.5 mg of purified radiolarian tests is typically required. Accordingly, it was expected that at least 1 mg of purified radiolarian tests would be extracted and purified from the samples used in this study. In this study, we successfully extracted over 1.5 mg of pure radiolarian tests for isotope analysis from all surface sediment samples. However, due to limited sample volumes, the quantity of pure radiolarian tests obtained from plankton samples was approximately 1 mg.

In the revised manuscript, we have substituted "sufficient radiolarian tests" with "a minimum of 1 mg of radiolarian tests". (Please see line 108 in the Author's track-changes file)

**5. Give a brief overview of Zhang and Swann (L81). Is there potential for larger diatoms or sponge spicules to 'contaminate' the sample?**

**Response:** Following the reviewer's suggestion, we have added a brief overview of Zhang and Swann (2023) in the revised manuscript as follows: Overall, the procedure for extracting and purifying radiolarian tests from marine sediments in Zhang and Swann (2023) comprises four stages: chemical treatment, initial sieving and differential settling, subsequent sieving, and finally density separation (Figure A). In the first stage, raw samples were treated with ~30% $H_2O_2$ and 15% HCl to remove the organic matters and calcareous components, and to facilitate particle dispersal. Following chemical treatment, the particles were rinsed and filtered using a 53 μm sieve to remove fine detritus, small diatoms, seaweed spines, and some sponge spicules. The filtered particles then underwent differential settling two to three times, followed by sonication, to further isolate radiolarians from large diatoms. Subsequently, particles were filtered three times using a 53 μm sieve by half immersing the sieve in a container filled with distilled deionized water (DDW) and gently tapping the base of the sieve for 10-15 minutes. This process aimed to remove all monoaxonic spicules and a portion of the small non-monoaxonic spicules. Finally, the retained fraction was further refined by density separation, using specific gravities from 2.1-2.0 g/cm$^3$ at 0.5-unit interval, to remove any remaining coarse detritus and non-monoaxonic sponge spicules (Please see lines 109-120 in the Author's track-changes file).

Larger diatoms and sponge spicules can also be effectively separated from radiolarian tests following the method of Zhang and Swann (2023). The majority of larger diatoms can be removed through differential settling (Figure A). Residual diatoms can then be broken down by sonication treatment for no more than 10 minutes, to avoid the breakage of radiolarian tests. These fragmented diatoms can then be further removed through sieving or differential settling. Although

[Figure]

Figure A the main process for separating and purifying the radiolarian tests from tropical marine sediments (Zhang and Swann, 2023). All photos taken under the inverted microscope at 100x magnification.

most sponge spicules are greater than 53 μm in length, the diameter (cross section) of monoaxonic sponge spicules is generally several micrometers. Therefore, monoaxonic spicules may pass through a 53 μm sieve during wet sieving (Figure A), provided they are repeatedly suspended and settle non-horizontally in the water during the sieving process. This can be achieved by half immersing samples in a container filled with DDW, and gently tapping the base of the sieve to maintain particle suspension. Non-monoaxonic sponge spicules can be further refined via density separation (Figure A), using specific gravities from 2.1-2.0 g/cm$^3$ at 0.5-unit interval. This is based on findings that the mean density of sponge spicules is higher than that of tropical radiolarians in late Quaternary sediments (Zhang and Swann, 2023).

Microscopic examination of purified radiolarian tests, across several fields of views at x100 magnification using inverted light microscopy, revealed a negligible quantity of larger diatoms and sponge spicules (<1%) remaining with the purified radiolarian tests (as shown in Videos 1 and 2 in the supplementary material of Zhang and Swann (2023)). Consequently, the potential for larger diatoms or sponge spicules to contaminate purified radiolarian samples is considered minimal or negligible.

**References:**

Zhang, Q., Swann, G. E. A.: An effective method to extract and purify radiolaria from tropical marine sediments, Front. Mar. Sci., 10, 1150518, https://doi.org/10.3389/fmars.2023.1150518, 2023.

**6. Confirm that Na (used in dissolution) was successfully removed by the ion exchange chromatography**

**Response:** Prior to isotopic analysis, all samples are purified using cation exchange chromatography. At a pH of 2 and 8, Si species are either neutral ($Si(OH)_4$) or anionic ($H_3SiO_4^-$) and will therefore pass freely through a cation exchange resin whilst all major cations, including $Na^+$, remain trapped on the column.

In the revised manuscript, the statement "Subsequent purification was achieved via ion exchange chromatography at a pH of between 2-8 (Georg et al., 2006; van den Boorn et al., 2006)." has been revised to "Subsequent purification was achieved via ion exchange chromatography at a pH of 2-8 to ensure complete removal of cations, such as magnesium (Mg) and/or sodium (Na) (Georg et al., 2006; van den Boorn et al., 2006)." (Please see lines 125-126 in the Author's track-changes file)

**7.  Give approximate mass resolution (m/Dm, L93)**

**Response:** In high-resolution (HR) mode, resolution (R) = $m/\Delta m$, where $\Delta m$ is derived from the rising edge of the peak, measured at 5% and 95% relative peak height. On the Neptune Plus at the British Geological Survey, R typically equates to between 9,000 –10,000, which is sufficient for partial (or pseudo) resolution of each of the silicon (Si) isotopes from their respective interferent/s.

In the revised manuscript, we have included the information of resolution "In high-resolution mode, the instrument typically exhibited a mass-resolution between approximately 9,000 to 10,000, and a sensitivity of 4-5 V/ppm." (Please see lines 139-140 in the Author's track-changes file)

**8.  Give details of how Mg measured/which ratio(s) (presumably in 'dynamic' mode), and a reference to Cardinal et al. 2004 (doi: 10.1039/b210109b) is probably appropriate.**

**Response:** The samples were doped with ~300ppb magnesium (Mg, Alfa Aesar SpectraPure). Spiking with an external element of known isotopic composition ($^{24}Mg/^{25}Mg$ = 0.126633) allows the data to be corrected for the effects of instrument-induced mass bias. Simply, any deviation of the measured $^{24}Mg/^{25}Mg$ value from the known value is attributed to the effects of mass bias. The isotopes of Si are assumed to be similarly affected, and consequently, an exponential drift correction is applied. The collector configuration used is illustrated below:

| Sequence \ Detector position | Low 4 | Low 3 | Axial | High 3 | Integration time/seconds | Settle time /seconds |
|---|---|---|---|---|---|---|
| 1 | | $^{28}Si$ | $^{29}Si$ | $^{30}Si$ | 16.8 | 3 |
| 2 | $^{24}Mg$ | | $^{25}Mg$ | | 8.4 | 3 |

Yes, we agree that the reference to Cardinal (Cardinal. D et al. J. Anal. At. Spectrom., 2003, 18, 213–218) (not Cardinal et al., 2004) should be cited here.

In the revised manuscript, the statement "Finally, all samples are spiked with approximately 300 ppb magnesium (Mg, Alfa Aesar SpectraPure, $^{24}Mg/^{25}Mg$ = 0.126633) to enable correction of the data for instrument-induced mass bias." has been revised to "Finally, all samples are spiked with approximately 300 ppb magnesium (Mg, Alfa Aesar SpectraPure, $^{24}Mg/^{25}Mg$ = 0.126633) to enable correction of the data for instrument-induced mass bias (Cardinal et al., 2003). Any deviation of the measured $^{24}Mg/^{25}Mg$ value from the known value is attributed to the effects of mass bias. The isotopes of Si are assumed to be similarly affected, and consequently, an

exponential drift correction is applied (Cardinal et al., 2003)." (Please see lines 132-134 in the Author's track-changes file)

**References:**

Cardinal, D., Alleman, L. Y., De Jong, J., Ziegler, K., & André, L.: Isotopic composition of silicon measured by multicollector plasma source mass spectrometry in dry plasma mode. Journal of Analytical Atomic Spectrometry, 18(3), 213-218, https://doi.org/10.1039/B210109B, 2003.

**9. Give approximate sample introduction rate, concentration, and instrument sensitivity**

**Response:** Yes, we have included the information of sample concentration, introduction rate, and instrument sensitivity in the revised manuscript. The statement "Silicon isotope analysis was performed in dry plasma mode using the high mass-resolution capabilityof a Thermo Fisher Scientific Neptune Plus MC-ICP-MS (multi collector inductively coupled plasma mass spectrometer) at the Geochronology and Tracers Facility, British Geological Survey." has been revised to "Silicon isotope analysis was performed in dry plasma mode using the high mass-resolution capability of a Thermo Fisher Scientific Neptune Plus MC-ICP-MS (multi collector inductively coupled plasma mass spectrometer) at the Geochronology and Tracers Facility, British Geological Survey. Samples were typically prepared to yield a Si concentration of approximately 2 ppm and introduced to the MC-ICP-MS via an Aridus de-solvating unit, incorporating a PFA nebulizer with an uptake rate of 50 μL/min. In high-resolution mode, the instrument typically exhibited a mass-resolution between approximately 9,000 to 10,000, and a sensitivity of 4-5 V/ppm." (Please see lines 135-140 in the Author's track-changes file)

**10. Confirm what one analytical replicate represents (just one standard-sample-standard bracket, or (as is usual) three or four?)**

**Response:** "Analytical replicates" in our study refer to the repeated analysis of the standard-sample-standard bracket, typically performed for three times. As we usually do not have sufficient sample, specifically for radiolarian tests from plankton samples, it is difficult to conduct a full procedural replicate that encompass both chemical processing and analysis. The samples selected for replication following the standard sample bracketing procedure are the ones that have a higher Si concentration and which can, therefore, be diluted to run multiple times.

To clarify this point, in the revised manuscript, the statement "Analytical replicates were conducted where sample volume allowed…" has been revised to "Analytical replicates of the standard sample bracketing procedure were conducted where sample volume allowed…". (Please see line 140 in the Author's track-changes file)

**11. L114: It's not clear what volume the 28025-102443 individuals refer to – in a 1m2 water column? It is also not clear how these numbers are derived – presumably because the volume of water passing through the nets (L66) is known? This could be clarified.**

**Response:** We are sorry for the confusion caused by our lack of clarity. The Hydro-Bios MultiNet used for collecting plankton samples has an aperture area of approximately 0.25 m$^2$ (a square aperture with 0.5 m sides). During sampling, we typically employ a retrieval rate of ~0.1 m/s for the MultiNet to ensure adequate filtration of the seawater. Therefore, the figures of 28,025–102,443 individuals represent the number of radiolarian shells in a 0.25 m$^2$ water column sampled from 0–100 m.

In the revised manuscript, we have included the aperture area of the Hydro-Bios MultiNet. The statement "Plankton samples were collected from the 0-100 m and 100-300 m water layers at each station using a Hydro-Bios MultiNet with a 63 μm mesh size…" has been revised to "Plankton samples were collected from the 0-100 m and 100-300 m water layers at each station using a Hydro-Bios MultiNet with an aperture area of approximately 0.25 m$^2$ and a 63 μm mesh size…". (Please see lines 76-77 in the Author's track-changes file)

In this study, 1/8 of each plankton sample was prepared as radiolarian slides for light microscope observations. More than 500 specimens were quantitatively identified and counted on each slide under the microscope at x100 magnification. The total number of radiolarian individuals in each sample was calculated from the count data as follows:

$$T = A * Vt/V * S * N$$

Where T is the total number of radiolarian individuals; A is the number of radiolarian shells counted from V fields of view on the slide; Vt is the total number of view fields on the radiolarian slide; V is the number of view fields examined under the microscope for the radiolarian individual count; S is the number of radiolarian slides; and N is the aliquot size of the sample (eight for the plankton sample, and one for the sediment sample in this study).

Based on comments from both reviewers, the original statement "To determine the species composition of radiolarians, in both plankton samples and surface sediments, all samples were wet-sieved through a 63 μm sieve and prepared into radiolarian slides following the method described by Zhang et al. (2014). Radiolarian species were then identified and counted under a Nikon optical microscope at x100 or x200 magnification, with more than 500 specimens identified on slides using the publications of Chen and Tan (1996) and Tan and Chen (1999)." has been revised to "To determine the species composition of radiolarians, in both plankton samples and surface sediments, all samples were wet-sieved through a 63 μm sieve and prepared into radiolarian slides following the method described by Zhang et al. (2014). Briefly, samples were treated with a sufficient volume of 5% HCl solution for 15 minutes to eliminate calcareous organisms. Subsequently, the residual was processed using a sonic oscillator for one minute, and subjected to differential settling to remove impurities potentially adhering to the radiolarian tests. Following these procedures, all residual material was strewn onto microscope slides and permanently mounted with Canada Balsam. Radiolarian species were then identified and counted under a Nikon optical microscope at x100 or x200 magnification, with more than 500 specimens identified on slides using the publications of Chen and Tan (1996) and Tan and Chen (1999). Radiolarian diversity was determined by the species richness in each sample. The total number of radiolarian individuals in each sample was estimated from the count data as follows:

$$T = A * (Vt/V)*S*N \tag{1}$$

Where T is the total number of radiolarian individuals; A is the number of radiolarian shells counted from V fields of view on the slide; Vt is the total number of view fields on the radiolarian slide; V is the number of view fields examined under the microscope for the radiolarian individual count; S is the number of radiolarian slides; and N is the aliquot size of the sample (eight for the plankton sample, and one for the sediment sample in this study)." (Please see lines 86-100 in the Author's track-changes file)

**12. L126: Can an approximate detection limit be given for these elements?**
**Response:** In this study, the detection limit of routine EDS analysis using IT-200 EDS detector for

major elements is approximately 1000pm or 0.1 %. However, elemental concentrations generated by the EDS software exceeding 0.1% do not necessarily confirm their presence within the analysed sample. As shown in Figure 4 of the manuscript, some elements (e.g. Na, exceeding 0.1%) within EDS (A" and B") spectrum images are flagged in red. This indicates that the software does not have a 99% statistical confidence in the presence of these elemental signatures within the analysed sample. This typically occurs for two reasons: either 1) the calculated wt% of these elements is very low, accompanied by a high standard deviation; or 2) the x-ray energy level peak for these elements does not stick out above the background noise peaks.

In the revised manuscript, we have included the approximate detection limit of the EDS analysis using IT-200 EDS detector for these major elements. The sentence "The concentrations of other elements (Fe, Al, Mg, Sr, Na, K and Cl) were all below the minimum detectable limit for the EDS detector (Figure 4)" has been revised to "other elements (Fe, Al, Mg, Sr, Na, K and Cl) flagged in red within the EDS spectrum images (Figure 4) indicate that either their concentrations were below the minimum detectable limit (~ 0.1 %) for the EDS detector, or the software lacked a 99% statistical confidence in the presence of these elemental signatures within the analysed sample". (Please see lines 179-183 in the Author's track-changes file)

13. **Discussion section: In general, there is no discussion of any spatial pattern in radiolarian assemblage. But I feel there is probably useful insight here. For example, Station 28 is located away from the cluster of other stations, and visually in Fig. 3 looks different. What physicochemical parameters influence the community assemblages? As an aside, I note that the assemblage data is not made available. Could the species in Fig 3 be amalgamated at a higher taxonomic level in order to make the similarities and differences clearer? And/or condensed, via an appropriate multivariate statistical approach, to 2 axes?**

**Response:** Thanks for the important point raised by the reviewer. The reviewer is correct in noting the difference between the radiolarian assemblage at Station 28 and those at some of the other 6 stations. This discrepancy is likely attributed to variations in regional environmental conditions, primarily including sea surface temperature (SST) and nutrient levels, such as dissolved silica or silicic acid in seawater. Station 28 is situated in the southern SCS, whereas the remaining 6 stations are located in the northern SCS. Overall, the radiolarian assemblage at station 28 exhibits a higher relative abundance of typical warm-water tropical species in radiolarian assemblages, such as *Botryocyrtis scutum* and *Pterocorys hertwigii* at station 28. This is likely due to the higher mean annual SST in the southern SCS, which forms part of the Western Pacific Warm Pool, compared to the northern SCS. Conversely, the radiolarian assemblage at station 28 has a lower relative abundance of *Didymocyrtis tetrathalamus t.,* a species indicative of nutrient-depleted Western Equatorial Pacific water intrusion into the northern SCS (e.g. Anderson et al., 1990; Zhang et al., 2015). This suggests a reduced influence of Western Pacific waters on the southern SCS compared to the northern SCS. These observed differences in specific species and radiolarian assemblages between stations in the southern and northern SCS are approximately consistent with previous investigations into variations in radiolarian assemblages within surface sediments and the water column, and their correlation with primary environmental parameters in the SCS (Chen et al. , 2008; Zhang et al., 2005, 2009).

The spatial pattern of the radiolarian assemblage is not discussed in this study. This is because, the primary objectives of radiolarian assemblage data presented in this study are to provide a crucial basis for: 1) identifying the water column depth interval from which the radiolarian community contributes to $\delta^{30}Si_{rad}$; 2) assessing the potential for substantial dissolution-induced alteration of the radiolarians by comparing the prominent radiolarian composition between the water column and sediments at each station; and 3) explaining why $\delta^{30}Si_{rad}$ signatures is faithfully transferred from the water column to the sediments. Consequently, the discussion has been primarily focused on prominent radiolarian compositions and $\delta^{30}Si_{rad}$ values between the water column and sedimentary record at individual stations. While variations in radiolarian assemblages across different regions may potentially result in spatial differences in $\delta^{30}Si_{rad}$ due to species-specific fractionation during the uptake of silicic acid by radiolaria (Doering et al., 2021) and possible variations in seawater $\delta^{30}Si$ ($\delta^{30}Si_{sw}$), a proper discussion of this issue requires corresponding data on both radiolarian assemblages and $\delta^{30}Si_{sw}$. In this study from the South China Sea, although the radiolarian assemblage of the $\delta^{30}Si_{rad}$ samples is known (Fig. 3 in the revised manuscript), the absence of matching $\delta^{30}Si_{sw}$ data from exactly the same sampling stations prevents further investigation of this issue at present.

We contend that amalgamating the species presented in Fig. 3 of the manuscript at a higher taxonomic level would diminish the clarity and effectiveness of this data in addressing the objectives outlined above. Therefore, we have retained the species-level information shown in Fig. 3, as it provides a more effective means of fulfilling the stated objectives for radiolarian assemblage data in this study. A correlation analysis provides a clear understanding of the similarities and differences in dominant radiolarian species between different water depth ranges and the sedimentary record at each station, as well as between various sampling stations. This analysis indicates a variation in correlations between the prominent radiolarian composition from station 28 and those from northern SCS, with Pearson correlation coefficients ranging from 0.27 to 0.83 (Table A shown on the next page).

In the revised manuscript, we have included the detailed radiolarian assemblage data for the samples used in this study as the supplementary material (please see original radiolarian count data in Tables 1 and 2 of supplementary material 2).

**Reference:**

Anderson O R, Bryan M, Bennett P. Experimental and observational studies of radiolarian physiological ecology: 4. Factors determining the distribution and survival ofDidymocyrtis tetrathalamus tetrathalamus with implications for paleoecological interpretations. Marine Micropaleontology, 1990, 16(3-4): 155-167.

Chen M H, Zhang L L, Zhang L L, et al. Radiolarian assemblages in surface sediments of the South China Sea and their marine environmental correlations. Earth Science—Journal of China University of Geosciences, 2008, 33(6): 775-782.

Zhang L L, Chen M H, Lu J, et al. Polycystine radiolarian fauna and their distribution in the upper water column of the southern South China Sea. Journal of Tropical Oceanography, 2005, 24, 55–64 (in Chinese with English abstract).

Zhang L L, Chen M H, Xiang R, et al. Distribution of polycystine radiolarians in the northern South China Sea in September 2005. Marine Micropaleontology, 2009, 70(1-2): 20-38.

Zhang, Q., Swann, G. E. A.: An effective method to extract and purify radiolaria from tropical marine sediments, Front. Mar. Sci., 10, 1150518, https://doi.org/10.3389/fmars.2023.1150518, 2023.

Table A Correlations between plankton samples and surface sediments for the prominent radiolarian species

| Sample | S7 (0-100m) | S7 (100-300m) | S7 SS | S8 (0-100m) | S8 (100-300m) | S8 SS | S10 (0-100m) | S10 (100-300m) | S10 SS | S11 (0-100m) | S11 (100-300m) | S11 SS | S12 (0-100m) | S12 (100-300m) | S12 SS | S13 (0-100m) | S13 (100-300m) | S13 SS | S28 (0-100m) | S28 (100-300m) | S28 SS |
|---|---|---|---|---|---|---|---|---|---|---|---|---|---|---|---|---|---|---|---|---|---|
| S7(0-100m) | 1 | | | | | | | | | | | | | | | | | | | | |
| S7 (100-300m) | .77** | 1 | | | | | | | | | | | | | | | | | | | |
| S7 SS | .79** | .87** | 1 | | | | | | | | | | | | | | | | | | |
| S8 (0-100m) | .62** | .67** | .75** | 1 | | | | | | | | | | | | | | | | | |
| S8 (100-300m) | .61** | .80** | .80** | .91** | 1 | | | | | | | | | | | | | | | | |
| S8 SS | .65** | .77** | .84** | .90** | .93** | 1 | | | | | | | | | | | | | | | |
| S10 (0-100m) | .47* | .58** | .51** | .63** | .72** | .70** | 1 | | | | | | | | | | | | | | |
| S10 (100-300m) | .34 | .58** | .55** | .53** | .68** | .58** | .74** | 1 | | | | | | | | | | | | | |
| S10 SS | .54** | .78** | .79** | .69** | .84** | .84** | .72** | .82** | 1 | | | | | | | | | | | | |
| S11 (0-100m) | .54** | .72** | .68** | .55** | .62** | .64** | .47* | .48* | .71** | 1 | | | | | | | | | | | |
| S11 (100-300m) | .55** | .71** | .71** | .50** | .70** | .73** | .56** | .53** | .75** | .71** | 1 | | | | | | | | | | |
| S11 SS | .52** | .67** | .60** | .77** | .83** | .83** | .79** | .66** | .83** | .69** | .79** | 1 | | | | | | | | | |
| S12 (0-100m) | .50** | .62** | .62** | .66** | .68** | .74** | .64** | .55** | .79** | .84** | .64** | .80** | 1 | | | | | | | | |
| S12 (100-300m) | .56** | .77** | .71** | .69** | .80** | .74** | .73** | .70** | .80** | .71** | .68** | .86** | .75** | 1 | | | | | | | |
| S12 SS | .41* | .60** | .60** | .71** | .78** | .78** | .78** | .70** | .74** | .60** | .64** | .85** | .74** | .85** | 1 | | | | | | |
| S13 (0-100m) | .58** | .81** | .77** | .72** | .81** | .84** | .66** | .65** | .86** | .87** | .72** | .82** | .89** | .82** | .75** | 1 | | | | | |
| S13 (100-300m) | .49* | .70** | .66** | .62** | .71** | .69** | .62** | .69** | .86** | .81** | .75** | .83** | .84** | .79** | .66** | .84** | 1 | | | | |
| S13 SS | .40* | .55** | .60** | .84** | .81** | .84** | .78** | .64** | .74** | .54** | .57** | .88** | .72** | .81** | .86** | .74** | .73** | 1 | | | |
| S28 (0-100m) | .27 | .38 | .41* | .50** | .57** | .42* | .61** | .76** | .62** | .49* | .36 | .63** | .53** | .70** | .62** | .54** | .63** | .63** | 1 | | |
| S28 (100-300m) | .27 | .46* | .45* | .48* | .58** | .42* | .58** | .83** | .67** | .37 | .33 | .55** | .41* | .71** | .54** | .48* | .56** | .56** | .85** | 1 | |
| S28 SS | .30 | .38 | .36 | .61** | .65** | .51** | .72** | .58** | .48* | .30 | .38 | .70** | .48* | .74** | .72** | .47* | .47* | .73** | .71** | .64** | 1 |

**14. L134-140: In general, it's the fractionation between dissolved Si and biogenic silica that is relevant, not the absolute $\delta^{30}$Si value. So it's a shame not to see any dissolved Si $\delta^{30}$Si data from the water samples (perhaps this is coming in a later publication?). An existing dataset of $\delta^{30}$Si exists for the SCS (Cao et al. 2012, doi: 10.1016/j.gca.2012.08.039). The overlap isn't perfect in terms of location or seasonality but its surprising that it's not mentioned here, given the importance of water $\delta^{30}$Si in setting radiolarian $\delta^{30}$Si, and that it would allow the authors to place constraints on the fractionation of Si isotopes by radiolarians.**

**Response:** Thanks for the reviewer's constructive comments. In the revised manuscript, we have added a new Section "4.2 Radiolaria Si isotope fractionation" regarding the discussion on the fractionation factor of the radiolarian, as detailed below (Please see lines 244-266 in the Author's track-changes file):

"4.2 Radiolaria silicon isotope fractionation

The isotope fractionation factor between radiolaria and seawater is a critical parameter for reconstructing past silicon cycle in the mid-upper ocean using $\delta^{30}$Si$_{rad}$ in downcore records (Hendry et al., 2014; Abelmann et al., 2015; Doering et al., 2021). As discussed previously, the $\delta^{30}$Si$_{rad}$ signatures in the water column of the SCS are predominantly contributed to by radiolarians from the 0-100 m depth layer. Therefore, $\delta^{30}$Si$_{sw}$ data pertinent to constraining silicon isotope fractionation by radiolarians in the SCS should derived primarily from this depth interval. Currently, there is no corresponding seawater $\delta^{30}$Si ($\delta^{30}$Si$_{sw}$) data from the stations in the SCS where water column radiolaria samples were collected. Previous work, however, has measured seasonal changes in $\delta^{30}$Si$_{sw}$ in the northern SCS (Cao et al 2012). At the two sites from that study that are off the continental shelf (A10 and SEATS), $\delta^{30}$Si$_{sw}$ above 100 m varies from 1.4‰ to 2.9‰. Assuming: 1) these values and their depth profiles are broadly comparable to stations where $\delta^{30}$Si$_{rad}$ was measured; and 2) that radiolarians construct their skeletons in equilibrium with ambient seawater (Abelmann et al., 2015, Fontorbe et al., 2016, Doering et al 2021), the isotope fractionation factor (ε) can be calculated as:

$$\varepsilon \sim \Delta^{30}Si = \delta^{30}Si_{rad} - \delta^{30}Si_{sw} \tag{2}$$

Using Monte Carlo simulations (10,000 replicates) to account for the 2SD uncertainty of both $\delta^{30}$Si$_{rad}$ and $\delta^{30}$Si$_{sw}$, ε is estimated to range from –0.33‰ to –0.92‰ (mean = 0.58‰) (Table S1 in supplementary material 1). Due to the difficulties in culturing radiolarian (e.g. Suzuki and Not, 2015), relatively little is known about the fractionation factor for $\delta^{30}$Si$_{rad}$. Values for ε obtained here from the SCS are comparable to those measured by Abelmann et al. (2015) in the Southern Ocean (–0.54‰ to –0.91‰, mean = -0.75‰) and by Doering et al. (2021) along the Peruvian Shelf (–0.35‰ to –0.79‰, mean = -0.62‰, mixed radiolaria). The large range of values for ε has been suggested to indicate species/order-specific fractionation during the uptake of silicic acid by radiolaria (Doering et al., 2021). Such a process would necessitate the extraction of species-specific radiolarian samples for isotope analysis in future palaeoceanographic research (e.g., Zhang and Swann, 2023). However in this current study from the SCS, whilst the species composition of the $\delta^{30}$Si$_{rad}$ samples is known (Figure 3), an absence of matching $\delta^{30}$Si$_{sw}$ data from exactly the same sampling stations prevents this issue being further investigated at this time."

In addition, the phrase "with the fractionation factor for $\delta^{30}$Si$_{rad}$ varying from –0.33‰ to –0.92‰ (mean = 0.58‰)" has been added into both the "Abstract" (Please see lines 7-8 in the Author's track-changes file) and Section "5 Conclusions". (Please see lines 319-320 in the Author's

**Reference:**

Abelmann, A., Gersonde, R., Knorr, G., Zhang, X., Chapligin, B., Maier, E., Esper, O., Friedrichsen, H., Lohmann, G., Meyer, H., Tiedemann, R.: The seasonal sea-ice zone in the glacial Southern Ocean as a carbon sink, Nat. Commun., 6(1), 8136, https://doi.org/10.1038/ncomms9136, 2015.

Cao, Z., Frank, M., Dai, M., Grasse, P., Ehlert, C.: Silicon isotope constraints on sources and utilization of silicic acid in the northern South China Sea, Geochimica et Cosmochimica Acta, 97, 88-104, https://doi.org/10.1016/j.gca.2012.08.039, 2012.

Doering, K., Ehlert, C., Pahnke, K., Frank, M., Schneider, R., Grasse, P.: Silicon isotope signatures of radiolaria reveal taxon-specific differences in isotope fractionation, Front. Mar. Sci., 8, 666896, https://doi.org/10.3389/fmars.2021.666896, 2021.

Fontorbe G, Frings P J, De La Rocha, C., Hendry, K. R., Conley, D. J.: A silicon depleted North Atlantic since the Palaeogene: Evidence from sponge and radiolarian silicon isotopes, Earth Planet. Sci. Lett., 453, 67-77, https://doi.org/10.1016/j.epsl.2016.08.006, 2016.

Suzuki, N., Not, F: Biology and Ecology of Radiolaria, In: Marine Protists, edited by: Ohtsuka, S., Suzaki, T., Horiguchi, T., Suzuki, N., Not, F., Springer, Tokyo, Japan, https://doi.org/10.1007/978-4-431-55130-0_8, 2015.

Zhang, Q., Swann, G. E. A.: An effective method to extract and purify radiolaria from tropical marine sediments, Front. Mar. Sci., 10, 1150518, https://doi.org/10.3389/fmars.2023.1150518, 2023.

**15. L165: Can an indication of the timespan covered by the upper 1cm be given? I presume there are some constraints on sedimentation rates and bioturbation in this well studied region.**

**Response:** Thanks for the reviewer's comment. Sheng et al., (2024) provide a comprehensive quantification of the spatial distribution of sedimentation rate (SR) and sediment budget over the entire South China Sea (SCS), using $^{210}$Pb measurements from 409 sediment cores, AMS$^{14}$C data from 112 gravity cores, and 33 sediment trap observations. The results show that Holocene sedimentation rates in the SCS exhibit regional variability. Specifically, under depositional conditions unaffected by turbidity currents and other anomalous processes, the mean sedimentation rate (SR) exceeds 25 cm/ka on upper continental slopes (water depth < 1000m), approximates 19 cm/ka in deeper water areas (water depth ≥ 1000m) along the lower continental slope slopes, and approximately 5-10 cm/ka near the deep-sea basin (water depth > 3000m) (Sheng et al., 2024). Based on these findings by Sheng et al., (2024), the estimated age of the uppermost 1cm of sediment used in this study ranges from less than 40 to 130 years. During the collection of surface sediments used in this study, the retrieved box-core samples were carefully inspected to avoid sampling at any stations exhibiting apparent bioturbation or disturbance from turbidity currents.

In the revised manuscript, we have included information regarding the timespan of the uppermost 1cm sediment at the end of the section "2.1 Sample material", as follows: No discernible evidence of bioturbation or disturbance from turbidity currents was observed in the sediment samples at these stations. Based on regional variability in Holocene sedimentation rates in the SCS (Sheng et al., 2024), the estimated age of the uppermost 1cm of sediment used in this study ranges from <40 to 133 years. (Please see lines 81-84 in the Author's track-changes file)

**References:**

Sheng, J., Qiao, S., Shi, X., Liu, J., Liu, Y., Liu, S., Wang, K., Mohamed, C., Khokiattiwong, S., Kornkanitnan, N.: Modern sedimentation and sediment budget in the South China Sea and their comparisons with the eastern China seas, Marine Geology, 475, 107348, https://doi.org/10.1016/j.margeo.2024.107348, 2024.

16. **Section 4.2: The radiolarian assemblages are similar between water column and core-top, so the inference is that 'dissolution is expected to have limited impacts on these radiolarian shells' (L209) – but is this necessarily true? Is is possible weakly silicified parts of the tests are dissolving? This would be interesting to know, as it has different implications for \*how\* the $\delta^{30}$Si is preserved: if no dissolution occurs, then there's no real potential for altering the isotopic signature (which therefore means the conclusions here are not transferable to other settings where more dissolution does occur). But if dissolution does occur and the $\delta^{30}$Si remains the same, then either a) different parts of the tests have the same $\delta^{30}$Si or b) a coincidental balance of heavier and lighter parts dissolved. To begin to address this, it would be good to see an independent constraint on the (radiolarian) biogenic silica preservation effeciency, either from the literature, from a comparison of export vs. sediment-trap/burial fluxes, or even a theoretical predicted efficiency based on sinking speeds, water column depth, and dissolution kinetics. Finally, It would be good to see an attempt to engage with what may happen with progressive dissolution in the upper centimeters of the sediment – if there is preferential dissolution of some species here, might that introduce a bias (an apparent fractionation) to the bulk-assemblage $\delta^{30}$Si data?**

**Response:** Thanks for reviewer's insightful comments. Yes, the observation that the radiolarian assemblages are similar between water column and surface sediments does not preclude the possibility that weakly silicified parts of the radiolarian tests have been dissolved. Although we have realized that some degree of dissolution is inevitable during the transport of radiolarians from the water column to the sediment, there is currently a paucity of established proxies for identifying weakly silicified parts of the radiolarian tests, or for tracing subtle dissolution effects. Moreover, no quantitative data are available in the literature regarding the preservation efficiency or burial fluxes of polycystine radiolarians from the South China Sea (SCS). Existing data of biogenic silica preservation efficiency (~1-3%) are not directly applicable to our study, as diatoms, a primary component of biogenic silica, exhibit a significantly higher degree of dissolution than polycystine radiolarians (e.g. Tréguer et al., 1995; Ragueneau et al., 2000). Consequently, obtaining effective preservation efficiency data suitable for assessing the dissolution of polycystine radiolarians is a significant challenge.

As shell fracture of microfossils preserved within sediments is commonly attributed to partial dissolution (e.g. Murray and Alve, 1999; Ryves et al., 2001), the proportion of fractured radiolarian shells may be indicative of the potential for radiolarian shell dissolution during sinking. Moreover, the preservation of delicate radiolarian skeletal structures, as assessed by SEM images, may also serve to determine whether the radiolarian shells have undergone significant dissolution. Therefore, in the revised manuscript, we have added the content regarding the assessment of radiolarian shell preservation in both the water column and sediments. This allows us to examine whether significant dissolution of radiolarian shells occurs during sinking within the water column. We have therefore made following essential revisions to the manuscript:

In the revised manuscript, the title of section 2.2 "Radiolarian composition analysis" has been revised to "Radiolarian composition and preservation analysis" (Please see line 85 in the Author's track-changes file), and we have included a second paragraph regarding assessing the preservation of radiolarian shells: "To assess the preservation of radiolarian shells both within the water column and seabed sediments, the proportion of fractured radiolarian shells was quantified in each studied sample under a Nikon optical microscope at x100 magnification. Further scanning electron microscopy (SEM) was employed to examine the potential for dissolution-induced alteration of the radiolarian skeleton." (Please see lines 103-106 in the Author's track-changes file).

Following the first paragraph of section "3 Results", we have included the result of radiolarian preservation analysis and a new Figure 4 (see below): "The proportion of fractured shells was low in the studied samples (Figure 4), generally ranging from 2 to 5% (mean = 3%) in plankton samples and from 3 to 9% (mean = 6%) in surface sediments. SEM images reveal a typical morphology of the pores on the radiolarian shell (Figure 4E), along with a high degree of integrity

[Figure]

Figure 4 Observations of radiolarian shell preservation at station 12 (water depth: 3497 m) in the plankton sample (100-300 m) (A and C) and surface sediments (B, D, E, and F) using optical and the scanning electron microscope. FRS=Fractured radiolarian shells; DS=Delicate skeletons

in the delicate skeletal structures (Figure 4F). These observations suggest good preservation of radiolarian shells, with no significant dissolution evident in either the water column or sediment samples." (Please see lines 171-175 in the Author's track-changes file)

The first paragraph of Section "4.3 Transfer of radiolarian $\delta^{30}$Si signatures into the sediment record" has been revised to "At each sampling station, $\delta^{30}$Si$_{rad}$ compositions (mean = 1.73‰) in the surface sediment closely resemble those (mean = 1.74‰) in the overlying water column evidenced by the paired t-test (p=0.75) (Figure 6), indicating a faithful transfer of the $\delta^{30}$Si signal incorporated into radiolarian skeletons from the water column to sediments. This suggests that dissolution has a minimal impact on the $\delta^{30}$Si$_{rad}$ signatures as radiolarian shells sink through the water column and become incorporated into the sediment record. One of two possibilities may account for this observation: 1) the radiolarian shells may not have undergone substantial dissolution during sinking; 2) the radiolarian shells have experienced substantial dissolution, but this process may not significantly alter their isotope composition. As shell fracture of microfossils preserved within sediments is commonly attributed to partial dissolution (e.g. Murray and Alve, 1999; Ryves et al., 2001), the proportion of fractured radiolarian shells may indicate the potential for radiolarian shell dissolution during sinking. In this study, fractured radiolarian shells comprised only a mean of 3% in plankton samples and 6% in surface sediment (Figure 4). Considering the low proportion of fractured radiolarian shells, the well-preserved state of radiolarian shells (Figure 4), and the minor differences in the relative abundances of prominent radiolarian species between plankton samples and surface sediments at each sampling station (as detailed in section 4.1), we propose that radiolarian shells have not experienced substantial dissolution or remineralisation during their transfer from the water column to the sediments." (Please see lines 268-281 in the Author's track-changes file)

At the beginning of the second paragraph of section "4.3 Transfer of radiolarian $\delta^{30}$Si signatures into the sediment record", we have added "The susceptibility of radiolarian shells to dissolution varies considerably with different taxonomic groups due to the differences in the chemical constituent of their skeletons." (Please see lines 282-283 in the Author's track-changes file)

**References:**

Murray, J. W., Alve, E.: Natural dissolution of modern shallow water benthic foraminifera: taphonomic effects on the palaeoecological record, Palaeogeography, Palaeoclimatology, Palaeoecology, 146(1-4), 195-209, https://doi.org/10.1016/S0031-0182(98)00132-1, 1999.

Ragueneau, O., Tréguer, P., Leynaert, A., Anderson, R. F., Brzezinski, M. A., DeMaster, D. J., Dugdale, R. C., Dymond, J., Fischer, G., Francois, R., Heinze, C., Maier-Reimer, E., Martin-Jézéquel, V. Nelson D. M., Quéguiner, B.: A review of the Si cycle in the modern ocean: recent progress and missing gaps in the application of biogenic opal as a paleoproductivity proxy, Global. Planet. Change, 26(4), 317-365, https://doi.org/10.1016/S0921-8181(00)00052-7, 2000.

Ryves, D. B., Juggins, S., Fritz, S. C., Battarbee, R. W.: Experimental diatom dissolution and the quantification of microfossil preservation in sediments. Palaeogeography, Palaeoclimatology, Palaeoecology, 172(1-2), 99-113, https://doi.org/10.1016/S0031-0182(01)00273-5, 2001.

Treguer, P., Nelson, D. M., Van Bennekom, A. J., DeMaster, D. J., Leynaert, A., Quéguiner, B. The silica balance in the world ocean: a reestimate, Science, 268(5209), 375-379, https://doi.org/10.1126/science.268.5209.375, 1995.

**Responses to the Reviewer 2's comments with revision notes**

In their manuscript Faithful transfer of radiolarian silicon isotope signatures from water column to sediments in the South China Sea" the authors present the first combined water column and surface sediment dataset of silicon isotope compositions of radiolaria. This data is further supplemented with radiolarian assemblage countings, which allows to investigate the influence of transfer from the water column to the sediment, especially via dissolution. The data presented is of great interest as d30Sirad has a high potential as a proxy to allow surface to intermediate water column reconstructions of the silica cycle, but is limited by few investigations on specific processes.

While I find the dataset of great importance and the presented findings are interesting and presented in a concise format, the authors should re-structure the discussion a bit and not solely discuss the influence of cleaning method and dissolution. This dataset has much more potential, especially as there is a dataset for d30Si-DSi values from the same area that could be used to investigate fractionation factors. I have summarized my main questions and comments below, followed by detailed comments per line.

**Main questions:**

**1. The major question in terms of silicon isotope compositions of radiolaria in the community at the moment is the question about their fractionation factor. As your study provides $\delta^{30}Si_{rad}$ from the water column and the sediment I am wondering why there is no discussion at all about the fractionation of Si and a calculation of $\delta^{30}Si$ (fractionation factor) even if you just measured the bulk radiolarian composition, as there are silicon isotope compositions of DSi from the water column available (Cao et al 2012, https://doi.org/10.1016/j.gca.2012.08.039).**

Response: We thank the reviewer for raising this important point. In the revised manuscript, we have added a new Section "4.2 Radiolaria silicon isotope fractionation" regarding the discussion on the fractionation factor of the radiolarian, as detailed below (Please see lines 244-266 in the Author's track-changes file):

"4.2 Radiolaria silicon isotope fractionation

The isotope fractionation factor between radiolaria and seawater is a critical parameter for reconstructing past silicon cycle in the mid-upper ocean using $\delta^{30}Si_{rad}$ in downcore records (Hendry et al., 2014; Abelmann et al., 2015; Doering et al., 2021). As discussed previously, the $\delta^{30}Si_{rad}$ signatures in the water column of the SCS are predominantly contributed to by radiolarians from the 0-100 m depth layer. Therefore, $\delta^{30}Si_{sw}$ data pertinent to constraining silicon isotope fractionation by radiolarians in the SCS should derived primarily from this depth interval. Currently, there is no corresponding seawater $\delta^{30}Si$ ($\delta^{30}Si_{sw}$) data from the stations in the SCS where water column radiolaria samples were collected. Previous work, however, has measured seasonal changes in $\delta^{30}Si_{sw}$ in the northern SCS (Cao et al 2012). At the two sites from that study that are off the continental shelf (A10 and SEATS), $\delta^{30}Si_{sw}$ above 100 m varies from 1.4‰ to 2.9‰. Assuming: 1) these values and their depth profiles are broadly comparable to stations where $\delta^{30}Si_{rad}$ was measured; and 2) that radiolarians construct their skeletons in equilibrium with ambient seawater (Abelmann et al., 2015, Fontorbe et al., 2016, Doering et al 2021), the isotope

fractionation factor (ε) can be calculated as:

$$\varepsilon \sim \Delta^{30}Si = \delta^{30}Si_{rad} - \delta^{30}Si_{sw} \tag{2}$$

Using Monte Carlo simulations (10,000 replicates) to account for the 2SD uncertainty of both $\delta^{30}Si_{rad}$ and $\delta^{30}Si_{sw}$, ε is estimated to range from –0.33‰ to –0.92‰ (mean = 0.58‰) (Table S1 in supplementary material 1). Due to the difficulties in culturing radiolarian (e.g. Suzuki and Not, 2015), relatively little is known about the fractionation factor for $\delta^{30}Si_{rad}$. Values for ε obtained here from the SCS are comparable to those measured by Abelmann et al. (2015) in the Southern Ocean (–0.54‰ to –0.91‰, mean = -0.75‰) and by Doering et al. (2021) along the Peruvian Shelf (–0.35‰ to –0.79‰, mean = -0.62‰, mixed radiolaria). The large range of values for ε has been suggested to indicate species/order-specific fractionation during the uptake of silicic acid by radiolaria (Doering et al., 2021). Such a process would necessitate the extraction of species-specific radiolarian samples for isotope analysis in future palaeoceanographic research (e.g., Zhang and Swann, 2023). However in this current study from the SCS, whilst the species composition of the $\delta^{30}Si_{rad}$ samples is known (Figure 3), an absence of matching $\delta^{30}Si_{sw}$ data from exactly the same sampling stations prevents this issue being further investigated at this time."

In addition, the phrase "with the fractionation factor for $\delta^{30}Si_{rad}$ varying from –0.33‰ to –0.92‰ (mean = 0.58‰)" has been added into both the "Abstract" (Please see lines 7-8 in the Author's track-changes file) and Section "5 Conclusions". (Please see lines 319-320 in the Author's track-changes file)

**Reference:**

Abelmann, A., Gersonde, R., Knorr, G., Zhang, X., Chapligin, B., Maier, E., Esper, O., Friedrichsen, H., Lohmann, G., Meyer, H., Tiedemann, R.: The seasonal sea-ice zone in the glacial Southern Ocean as a carbon sink, Nat. Commun., 6(1), 8136, https://doi.org/10.1038/ncomms9136, 2015.

Cao, Z., Frank, M., Dai, M., Grasse, P., Ehlert, C.: Silicon isotope constraints on sources and utilization of silicic acid in the northern South China Sea, Geochimica et Cosmochimica Acta, 97, 88-104, https://doi.org/10.1016/j.gca.2012.08.039, 2012.

Doering, K., Ehlert, C., Pahnke, K., Frank, M., Schneider, R., Grasse, P.: Silicon isotope signatures of radiolaria reveal taxon-specific differences in isotope fractionation, Front. Mar. Sci., 8, 666896, https://doi.org/10.3389/fmars.2021.666896, 2021.

Fontorbe G, Frings P J, De La Rocha, C., Hendry, K. R., Conley, D. J.: A silicon depleted North Atlantic since the Palaeogene: Evidence from sponge and radiolarian silicon isotopes, Earth Planet. Sci. Lett., 453, 67-77, https://doi.org/10.1016/j.epsl.2016.08.006, 2016.

Suzuki, N., Not, F: Biology and Ecology of Radiolaria, In: Marine Protists, edited by: Ohtsuka, S., Suzaki, T., Horiguchi, T., Suzuki, N., Not, F., Springer, Tokyo, Japan, https://doi.org/10.1007/978-4-431-55130-0_8, 2015.

Zhang, Q., Swann, G. E. A.: An effective method to extract and purify radiolaria from tropical marine sediments, Front. Mar. Sci., 10, 1150518, https://doi.org/10.3389/fmars.2023.1150518, 2023.

**2. The sample preparation and evaluation should be described in a bit more detail as you are referring to a newly published method it would be good to at least mention the main steps involved in the preparation (see detailed comments below). Furthermore, I would be interested to know if sample amounts were generally too low to try to pick single species or orders of radiolaria, as this would help better understand how different radiolarians incorporate Si and if there is a difference in Si fractionation.**

**Response:** According to the reviewer's suggestion, we have included a summary of the method used to extract and purify radiolarian tests for silicon isotope analysis (Zhang and Swann, 2023) in the revised manuscript as follows: "For isotope analysis, approximate 1 to 1.5 mg of radiolarian tests were extracted and purified from 7 plankton samples and 7 surface sediment samples following the method of Zhang and Swann (2023). Overall, the procedure for extracting and purifying radiolarian tests from marine sediments in Zhang and Swann (2023) comprises four stages: chemical treatment, initial sieving and differential settling, subsequent sieving, and finally density separation (Figure A). In the first stage, raw samples were treated with ~30% $H_2O_2$ and 15% HCl to remove the organic matters and calcareous components, and to facilitate particle dispersal. Following chemical treatment, the particles were rinsed and filtered using a 53 μm sieve to remove fine detritus, small diatoms, seaweed spines, and some sponge spicules. The filtered particles then underwent differential settling two to three times, followed by sonication, to further isolate radiolarians from large diatoms. Subsequently, particles were filtered three times using a 53 μm sieve by half immersing the sieve in a container filled with distilled deionized water (DDW) and gently tapping the base of the sieve for 10-15 minutes. This process aimed to remove all monoaxonic spicules and a portion of the small non-monoaxonic spicules. Finally, the retained fraction was further refined by density separation, using specific gravities from 2.1-2.0 $g/cm^3$ at 0.5-unit interval, to remove any remaining coarse detritus and non-monoaxonic sponge spicules." (Please see lines 109-120 in the Author's track-changes file)

[Figure]

Figure A the main process for separating and purifying the radiolarian tests from tropical marine sediments (Zhang and Swann, 2023). All photos taken under the inverted microscope at 100x magnification.

Indeed, $\delta^{30}Si$ records from single-species tests may improve our understanding of how different radiolarians incorporate Si and allow us to examine potential variations in Si isotope fractionation between different radiolarian species. However, obtaining sufficient material for silicon isotopic analysis from individual radiolarian taxa in our samples proved to be a significant challenge, necessitating our focus on the bulk species record for this study. We did, in fact, attempt to isolate single-species tests via manual picking under a microscope, but encountered several obstacles. Firstly, for silicon isotope analysis using a Thermo Fisher Scientific Neptune Plus MC-ICP-MS at the Geochronology and Tracers Facility, British Geological Survey, 1 to 1.5 mg of purified radiolarian tests is typically required. Radiolarian tests from sediments in the tropical Ocean are

notably lightweight, averaging 0.063 to 0.136 mg/shell (Moore, 1969; Takahashi, 1982). Consequently, a mean of approximately 7,000 to 15,000 individual tests are required to achieve the ~1 mg of material needed for the silicon isotope analysis. Given the high diversity of radiolarians in Holocene sediments, particularly in low-latitude oceans (Moore, 1969; Takahashi, 1982; Boltovskoy et al., 2010) and in our samples, which average over 100 species per sample, coupled with the observation that most species typically constitute less than 10% of the total radiolarian community (Boltovskoy et al., 2010; Chen et al., 2008), this means that 70,000 to 150,000 bulk radiolarian tests are required for each sample to obtain sufficient single-species material. Regrettably, the limited volume of the plankton samples made it difficult to obtain such a large number of radiolarian shells. Secondly, manually picking 7,000 to 15,000 individual tests out of 70,000 to 150,000 bulk radiolarian tests is highly time-consuming, rendering this approach impractical for the study of single-species radiolarian $\delta^{30}$Si. Furthermore, even with meticulous hand-picking under a microscope, the electro static adsorption of these light tests to transfer tools, such as brushes, inevitably results in test loss during transfer to storage vials. This further compounded the difficulty of accumulating the required quantity of monospecific tests for the isotope analysis.

It was these challenges encountered during our attempts at hand-picking that prompted us to develop the method for extracting and purifying bulk radiolarian tests from the sediments for the study of the $\delta^{30}$Si$_{rad}$ (Zhang and Swann, 2023). This method enabled us to effectively obtain sufficient pure radiolarian tests to determine mixed-radiolarian $\delta^{30}$Si compositions, prior to the availability of more effective techniques for obtaining sufficient monospecific tests for isotope analysis.

In the revised manuscript we have included a brief explanation regarding the absence of silicon isotopic analysis for individual radiolarian taxa: "Given the lightweight nature of shells and the high diversity of radiolarians, particularly in low-latitude oceans (Moore, 1969; Takahashi, 1982; Boltovskoy et al., 2010), combined with the observation that most species typically comprise less than 10% of the total radiolarian assemblage in both the water column and the sedimentary record (Boltovskoy et al., 2010; Chen et al., 2008a), obtaining sufficient material from individual radiolarian taxa for silicon isotopic analysis remains a considerable challenge." (Please see lines 62-66 in the Author's track-changes file)

**References:**

Boltovskoy, D., Kling, S. A., Takahashi, K., Bjørklund, K.: World atlas of distribution of recent polycystina (Radiolaria), Palaeontologia Electronica, 13, 1–229, 2010.

Chen M H, Zhang L L, Zhang L L , et al. 2008a. Preservation of radiolarian diversity and abundance in surface sediments of the South China Sea and its environmental implication. Journal of China University of Geosciences, 19: 217–229.

Doering, K., Ehlert, C., Pahnke, K., Frank, M., Schneider, R., Grasse, P.: Silicon isotope signatures of radiolaria reveal taxon-specific differences in isotope fractionation, Front. Mar. Sci., 8, 666896, https://doi.org/10.3389/fmars.2021.666896, 2021.

Moore, JR. T. C: Radiolaria: change in skeletal weight and resistance to solution. Geological Society of America Bulletin, 80(10), 2103-2108. https://doi.org/10.1130/0016-7606(1969)80[2103:RCISWA]2.0.CO;2, 1969

Takahashi, K.: Vertical flux, ecology and dissolution of Radiolaria in tropical oceans: implications for the silica cycle. PhD thesis. Woods Hole Oceanographic Institution and Massachusetts Institute of Technology, 1982

Zhang, Q., Swann, G. E. A.: An effective method to extract and purify radiolaria from tropical marine sediments,

Front. Mar. Sci., 10, 1150518, https://doi.org/10.3389/fmars.2023.1150518, 2023.

**3. The discussion feels a bit shallow. The highlight of your manuscript is the investigation of any dissolution effect on the sedimentary radiolarian signal. However, as you point out in the discussion, the prominent radiolarian species in your samples turn out to be species that are considered to be quite resistant to resolution. I suggest including some of the references you refer to in the last part of the discussion in the introduction instead and cutting the actual discussion about dissolution shorter, as you basically show that there is no evidence on dissolution based on the abundance data and the silicon isotope data. While you show this with your statistical analyses, you can also highlight that your d30Sirad values of water column and surface sediments are the same within analytical precision as indicated in Figure 5.**

**Response:** Thanks for the reviewer's comment. The discussion section, "4 Discussions," has been restructured from two parts in the original manuscript to three parts in the revised manuscript. Given that radiolarians are distributed throughout the water column from surface to bottom waters in the oceans (e.g. Kling, 1979; Hu et al., 2015; Boltovskoy, 2017), the first part of the discussion section, "4.1 Radiolarian $\delta^{30}$Si signatures in water column plankton and surface sediment samples", primarily aims to determine the depth interval from which the radiolarian community contributes to the $\delta^{30}$Si$_{rad}$ signature in the water column. This determination is achieved through the analysis and comparison of radiolarian diversity and abundance at various depths. The primary objective is to improve our understanding of the water depth interval represented by $\delta^{30}$Si$_{rad}$ records in both the water column and sediments, thereby providing the essential basis for reconstructing the silicon cycle within specific depth intervals, utilising radiolarian silicon isotope records from sedimentary sequences. To improve clarity concerning the focus of this section, the original title of this part "Radiolarian $\delta^{30}$Si signatures in water column plankton and surface sediment samples" has been revised as "4.1 Contributors to radiolarian $\delta^{30}$Si signatures in water column plankton samples" in the revised manuscript (Please see lines 190 in the Author's track-changes file). Furthermore, the finding that "$\delta^{30}$Si $_{rad}$ signatures in the water column are primarily contributed to by radiolarians from the 0-100 m water depth layer in the SCS" has been added in the abstract (Please see lines 8-9 in the Author's track-changes file).

"4.2 Radiolaria silicon isotope fractionation," has been added as the second part of the discussion section, to address silicon isotope fractionation in radiolarians, and to provide details relevant to main question 1. (Please see lines 244-266 in the Author's track-changes file)

The third part (the second part in the original manuscript) "4.3 Transfer of radiolarian $\delta^{30}$Si signatures into the sediment record" focuses on explaining why the $\delta^{30}$Si$_{rad}$ signal from the water column is faithfully transferred to the sediments. Our data indicate a good consistency between $\delta^{30}$Si$_{rad}$ signatures in the water column and the sedimentary record, indicating a faithful transfer of the $\delta^{30}$Si signal incorporated into radiolarian skeletons from the water column to sediments. This suggests that dissolution has a minimal impact on the $\delta^{30}$Si$_{rad}$ signatures as radiolarian shells sink through the water column and become incorporated into the sediment record. This observation may be attributable to one of two possible reasons: 1) the radiolarian shells may not have undergone substantial dissolution during sinking; 2) the radiolarian shells have experienced substantial dissolution, but this process may not significantly alter their isotope composition. As shell fracture of microfossils preserved within sediments is commonly attributed to partial

dissolution (e.g. Murray and Alve, 1999; Ryves et al., 2001), the proportion of fractured radiolarian shells may indicate the potential for radiolarian shell dissolution during sinking. Moreover, the preservation of delicate radiolarian skeletal structures, as assessed by SEM images, may also serve to determine whether the radiolarian shells have undergone significant dissolution. Therefore, in the revised manuscript, we have added the content regarding the assessment of the preservation of radiolarian shells in both the water column and sediments. This allows us to examine whether significant dissolution of radiolarian shells occurs during sinking within the water column. We have therefore made following essential revisions to the manuscript:

In the revised manuscript, the title of section "2.2 Radiolarian composition analysis" has been revised to "Radiolarian composition and preservation analysis" (Please see line 85 in the Author's track-changes file), and we have included a second paragraph regarding assessing the preservation of radiolarian shells: "To assess the preservation of radiolarian shells both within the water column and seabed sediments, the proportion of fractured radiolarian shells was quantified in each studied sample under a Nikon optical microscope at x100 magnification. Further scanning electron microscopy (SEM) was employed to examine the potential for dissolution-induced alteration of the radiolarian skeleton.". (Please see lines 103-106 in the Author's track-changes file)

Following the first paragraph of section "3 Results", we have included the result of radiolarian preservation analysis and a new Figure 4 (see below): "The proportion of fractured shells was low in the studied samples (Figure 4), generally ranging from 2 to 5% (mean = 3%) in plankton samples and from 3 to 9% (mean = 6%) in surface sediments. SEM images revealed a typical morphology of the pores on the radiolarian shell (Figure 4E), along with a high degree of integrity in the delicate skeletal structures (Figure 4F). These observations suggest good preservation of radiolarian shells, with no significant dissolution evident in either the water column or sediment samples." (Please see lines 171-175 in the Author's track-changes file)

The first paragraph of Section "4.3 Transfer of radiolarian $\delta^{30}$Si signatures into the sediment record" has been revised to "At each sampling station, $\delta^{30}$Si$_{rad}$ compositions (mean = 1.73‰) in the surface sediment closely resemble those (mean = 1.74‰) in the overlying water column evidenced by the paired t-test (p=0.75), indicating a faithful transfer of the $\delta^{30}$Si signal incorporated into radiolarian skeletons from the water column to sediments. This suggests that dissolution has a minimal impact on the $\delta^{30}$Si$_{rad}$ signatures as radiolarian shells sink through the water column and become incorporated into the sediment record. One of two possibilities may account for this observation: 1) the radiolarian shells may not have undergone substantial dissolution during sinking; 2) the radiolarian shells have experienced substantial dissolution, but this process may not significantly alter their isotope composition. As shell fracture of microfossils preserved within sediments is commonly attributed to partial dissolution (e.g. Murray and Alve, 1999; Ryves et al., 2001), the proportion of fractured radiolarian shells may indicate the potential for radiolarian shell dissolution during sinking. In this study, fractured radiolarian shells comprised only a mean of 3% in plankton samples and 6% in surface sediment (Figure 4). Considering the low proportion of fractured radiolarian shells, the well-preserved state of radiolarian shells (Figure 4), and the minor differences in the relative abundances of prominent radiolarian species between plankton samples and surface sediments at each sampling station (as detailed in section 4.1), we propose that radiolarian shells have not experienced substantial dissolution or remineralisation during their transfer from the water column to the sediments." (Please see lines 268-281 in the Author's track-changes file)

[Figure]

Figure 4 Observations of radiolarian shells preservation at station 12 (water depth: 3497 m) in the plankton sample (100-300 m) (A and C) and surface sediments (B, D, E, and F) using optical and the scanning electron microscope. FRS=Fractured radiolarian shells; DS=Delicate skeletons

The dissolution characteristics of radiolarian shells vary considerably with different taxonomic groups due to notable differences in the chemical constituent of their skeletons. For example, Collodaria radiolarians, a group belonging to polycystine radiolarians, are highly susceptible to dissolution during sinking due to their skeletons composed of both opal and Celestine (Afanasieva et al., 2005; Afanasieva and Amon, 2014). However, as indicated above, our observations provide no evidence that the radiolarian shells have experienced substantial dissolution during sinking within the water column. Therefore, the third part of the discussion need to addresses why radiolarians preserved in surface sediments have not undergone significant dissolution. Only by clarifying the shell chemical composition and dissolution characteristics of different radiolarian groups, as well as the approximate proportion of each radiolarian group in our studied samples, can we better understand why the radiolarian shells are minimally impacted by dissolution during sinking through the water column. This thus allows for a compelling explanation of why the radiolarian silicon isotope signal from the water column is faithfully transferred to the sediments. Therefore, information regarding the radiolarian taxonomy, chemical composition and dissolution characteristics of each taxonomic group, and proportion of each taxonomic group in radiolarian community in our analyzed samples, is essential in the discussion of the manuscript.

In the revised manuscript, we have added "The susceptibility of radiolarian shells to dissolution varies considerably with different taxonomic groups due to the differences in the chemical constituent of their skeletons." to the beginning of the third paragraph of section "4.3 Transfer of radiolarian $\delta^{30}$Si signatures into the sediment record". (Please see lines 282-283 in the Author's track-changes file)

While $\delta^{30}$Si$_{rad}$ values of water column and surface sediments are considered to be the same within analytical precision, we conducted a statistical analysis because it could more objectively describe this similarity in $\delta^{30}$Si$_{rad}$ values in water column and surface sediments.

**References:**

Afanasieva, M. S., Amon, E. O.: Biomineralization of radiolarian skeletons, Paleontological Journal, 48, 1473-1486, https://doi.org/10.1134/S0031030114140020, 2014.

Afanasieva, M. S., Amon, E. O., Agarkov, Y. V., Boltovskoy, D. S.: Radiolarians in the geological Record, Paleontological Journal, 39, 135-392, https://doi.org/10.1666/0094-8373(2005)031, 2005.

Boltovskoy, D.: Vertical distribution patterns of Radiolaria Polycystina (Protista) in the World Ocean: living ranges, isothermal submersion and settling shells, Journal of Plankton Research, 39(2), 330-349, https://doi.org/10.1093/plankt/fbx003, 2017.

Hu, W. F., Zhang, L. L., Chen, M. H., Zeng, L. L., Zhou, W. H., Xiang, R., Zhang, Q., Liu, S. H.: Distribution of living radiolarians in spring in the South China Sea and its responses to environmental factors, Sci. China Earth Sci., 58, 270-85, https://doi.org/10.1007/s11430-014-4950-0, 2015

Kling, S. A.: Vertical distribution of polycystine radiolarians in the central North Pacific, Mar. Micropaleontol., 4, 295-318, https://doi.org/10.1016/0377-8398(79)90022-7, 1979.

Murray, J. W., Alve, E.: Natural dissolution of modern shallow water benthic foraminifera: taphonomic effects on the palaeoecological record, Palaeogeography, Palaeoclimatology, Palaeoecology, 146(1-4), 195-209, https://doi.org/10.1016/S0031-0182(98)00132-1, 1999.

Ryves, D. B., Juggins, S., Fritz, S. C., Battarbee, R. W.: Experimental diatom dissolution and the quantification of microfossil preservation in sediments. Palaeogeography, Palaeoclimatology, Palaeoecology, 172(1-2), 99-113, https://doi.org/10.1016/S0031-0182(01)00273-5, 2001.

**Comments per line:**

**4. L6: You should remove the comma after "water column".**

   **Response:** Yes, the comma after "water column" has been deleted in the revised manuscript. (Please see line 6 in the Author's track-changes file)

**5. L8: Add a comma before "as evidenced".**

   **Response:** Yes, the comma before "as evidenced" has been added. (Please see line 10 in the Author's track-changes file)

**6. L18: Add a comma behind "Based on this".**

**Response:** Yes, the comma before "as evidenced" has been added. (Please see line 20 in the Author's track-changes file)

**7. L25: Remove "actually".**

Response: Yes, the term "actually" has been deleted. (Please see line 27 in the Author's track-changes file)

**8. L26-27: Remove the commas behind "mechanism" and "impact" and "in both glacials".**

Response: Yes, the commas behind "mechanism", "impact", and "in both glacials" have been deleted. (Please see lines 28-29 in the Author's track-changes file)

**9. L29: I suggest removing "wider" or replacing it with "broader".**

Response: According to the reviewer's suggestion, the term "wider" has been deleted. (Please see line 31 in the Author's track-changes file)

**10. L42: Remove "so".**

Response: Yes, the term "so" has been deleted. (Please see line 44 in the Author's track-changes file)

**11. L46: Change to "others".**

Response: We appreciate the reviewer's suggestion. However, we have confirmed that the term "others" is correct in line 46 of the original manuscript.

**12. L49: Remove "also".**

Response: Yes, the word "also" has been deleted. (Please see line 52 in the Author's track-changes file)

**13. L54: The authors claim that purifying radiolarian test from the natural environment have difficulties. Can the authors add a reference here and specify what difficulties they mean?**

Response: We agree with the reviewer that the reference is needed regarding the difficulties in purifying radiolarian test from the natural environment. Prior to the method for extracting and purifying radiolarians from sediments proposed by Zhang and Swann (2023), the approach to obtain purified radiolarians for isotope analysis was manual picking under a light microscope (Hendry et al., 2014; Fontorbe et al., 2016; Fontorbe et al., 2017; Doering et al., 2021). In general, there are three problems with manual picking. Firstly, radiolarian tests from Quaternary sediments in the tropical Pacific Ocean are light with an average weight from 0.063 to 0.136 mg/shell (Moore, 1969; Takahashi, 1982). As such, 7,000 to 15,000 radiolarian tests need to be handpicked per 1 mg of material needed for isotope analysis. Secondly, even if tests were handpicked under a microscope, it is unavoidable to lose some radiolarian tests when transferring samples to a storage vial due to electrostatic adsorption of the light shells to the transfer tool, such as a brush. Finally, the composition of the radiolarian assemblages may be changed substantially because of the randomness of manual picking or priority selection of larger individuals, which may bias the measured isotope compositions. Because these difficulties were stated in Zhang and Swann (2023), we do not reiterate them in this manuscript.

In the revised manuscript, we have included the relevant references regarding the difficulties in successfully culturing radiolarians and purifying radiolarian test from the natural environment

(Suzuki and Aita, 2011; Suzuki and Not, 2015; Zhang and Swann, 2023). (Please see lines 57-58 in the Author's track-changes file)

**References:**

Suzuki, N., Aita, Y.: Radiolaria: achievements and unresolved issues: taxonomy and cytology, Plankton and Benthos Research, 6(2), 69-91, https://doi.org/10.3800/pbr.6.69, 2011.

Suzuki, N., Not, F: Biology and Ecology of Radiolaria, In: Marine Protists, edited by: Ohtsuka, S., Suzaki, T., Horiguchi, T., Suzuki, N., Not, F., Springer, Tokyo, Japan, https://doi.org/10.1007/978-4-431-55130-0_8, 2015.

Zhang, Q., Swann, G. E. A.: An effective method to extract and purify radiolaria from tropical marine sediments, Front. Mar. Sci., 10, 1150518, https://doi.org/10.3389/fmars.2023.1150518, 2023.

**14. L70: Remove the comma behind "box corer" and add an article (either "the" or "a") before "cold storage"**

**Response:** Yes, we have deleted the comma behind "box corer" and added "the" before the term "cold storage". (Please see line 81 in the Author's track-changes file)

**15. L73-78: Under section 2.2 the authors refer to the preparation of radiolarian slides but only refer to a previous paper concerning the actual method. Please add some specifics of which steps this methodology includes concisely.**

**Response:** Based on the reviewer's comment, we have added a brief description regarding the preparation of radiolarian slides to the revised manuscript. Furthermore, we have clarified the methodology used to estimate the total number of radiolarian individuals in each sample in accordance with the comment from the other reviewer as follows: "To determine the species composition of radiolarians, in both plankton samples and surface sediments, all samples were wet-sieved through a 63 μm sieve and prepared into radiolarian slides following the method described by Zhang et al. (2014). Briefly, samples were treated with a sufficient volume of 5% HCl solution for 15 minutes to eliminate calcareous organisms. Subsequently, the residual was processed using a sonic oscillator for one minute, and subjected to differential settling to remove impurities potentially adhering to the radiolarian tests. Following these procedures, all residual material was strewn nearly homogenously onto microscope slides and permanently mounted with Canada Balsam. Radiolarian species were then identified and counted under a Nikon optical microscope at x100 or x200 magnification, with more than 500 specimens identified on slides using the publications of Chen and Tan (1996) and Tan and Chen (1999). Radiolarian diversity was determined by the species richness in each sample. The total number of radiolarian individuals in each sample was estimated from the count data as follows:

$$T = A * (Vt/V)*S*N \qquad (1)$$

Where T is the total number of radiolarian individuals; A is the number of radiolarian shells counted from V fields of view on the slide; Vt is the total number of view fields on the radiolarian slide; V is the number of view fields examined under the microscope for the radiolarian individual count; S is the number of radiolarian slides; and N is the aliquot size of the sample (eight for the plankton sample, and one for the sediment sample in this study). Relative abundances of various species were then calculated based on individual counts of each species and the total number of radiolarian specimens on each slide observed under the microscope." (Please see lines 86-100 in the Author's track-changes file)

**17. L78: Changes to "individual counts".**

**Response:** Thanks for the meticulous review by the reviewer. The phrase that the reviewer mentioned should be "individual count", which appears on line 77 of the current manuscript. We have revised the manuscript accordingly, replacing "individual count" with "individual counts". (Please see line 101 in the Author's track-changes file)

**18. L 80-81: Similarly to my comment above the others only refer to a previous paper concerning the preparation of surface sediment, please add a short description what this includes, freezing, freeze-drying, wet-sieving as stated in the section above? Further, can you specify what "sufficient" radiolarian tests are?**

**Response:** Yes, following the reviewer's comment, we have included a brief overview of Zhang and Swann (2023) in the revised manuscript as follows: Overall, the procedure for extracting and purifying radiolarian tests from marine sediments in Zhang and Swann (2023) comprises four stages: chemical treatment, initial sieving and differential settling, subsequent sieving, and finally density separation (please refer to Figure A in the response to the main question 2). In the first stage, raw samples were treated with ~30% $H_2O_2$ and 15% HCl to remove the organic matters and calcareous components, and to facilitate particle dispersal. Following chemical treatment, the particles were rinsed and filtered using a 53 μm sieve to remove fine detritus, small diatoms, seaweed spines, and some sponge spicules. The filtered particles then underwent differential settling two to three times, followed by sonication, to further isolate radiolarians from large diatoms and other detrital fine-grains. Subsequently, particles were filtered three times using a 53 μm sieve by half immersing the sieve in a container filled with distilled deionized water (DDW) and gently tapping the base of the sieve for 10-15 minutes. This process aimed to remove all monoaxonic spicules and a portion of the small non-monoaxonic spicules. Finally, the retained fraction was further refined by density separation, using specific gravities from 2.1-2.0 g/cm$^3$ at 0.5-unit interval, to remove any remaining coarse detritus and non-monoaxonic sponge spicules. The purity of extracted radiolarian tests in each sample was visually assessed under a Nikon optical microscope at x100 magnification, with selected samples further examined using a scanning electron microscopy (SEM) (JEOL JSM-IT200) equipped with an energy dispersive X-ray spectroscopy (EDS) detector at the Nanoscale and Microscale Research Centre, University of Nottingham. (Please see lines 108-120 in the Author's track-changes file)

We are sorry for the lack of clarity regarding the typical mass of radiolarian tests for isotope analysis. For silicon isotope analysis, approximately 1 to 1.5 mg of purified radiolarian tests is typically required using a Thermo Fisher Scientific Neptune Plus MC-ICP-MS at the Geochronology and Tracers Facility, British Geological Survey. In this study, we typically extracted over 1.5 mg of pure radiolarian tests for isotope analysis from all surface sediment samples. However, due to limited sample volumes, the quantity of pure radiolarian tests obtained from plankton samples was approximately 1 mg.

In the revised manuscript, the statement "…sufficient radiolarian tests…" has been revised to "…a minimum of 1 mg of radiolarian tests…". (Please see line 108 in the Author's track-changes file)

**19. L85: rephrase to "using NaOH fusion".**

**Response:** Yes, the phrase "using the NaOH fusion method" has been revised to "using NaOH

fusion" in the revised manuscript. (Please see line 124 in the Author's track-changes file)

**L86: Remove "between".**

**Response:** Yes, the word "between" has been deleted, and the phrase "a pH of between 2-8" has therefore been revised to "a pH of 2-8". (Please see line 125 in the Author's track-changes file)

20. **L95: You are referring to analytical replicates here. Was the same sample prepared by NaOH fusion measured on a different day, or are you referring to replicates within the standard sample bracketing procedure?**

**Response:** "Analytical replicates" in our study refer to the repeated analysis of the standard-sample-standard bracket, typically performed for three times. As we usually do not have sufficient sample specifically for radiolarian tests from plankton samples, it difficult to conduct a full procedural replicate that encompass both chemical processing and analysis. The samples selected for replication following the standard sample bracketing procedure are the ones that have a higher Si concentration and which can, therefore, be diluted to run multiple times.

   To clarify this point, in the revised manuscript, the statement "Analytical replicates were conducted where sample volume allowed…" has been revised to "Analytical replicates of the standard sample bracketing procedure were conducted where sample volume allowed…". (Please see line 140 in the Author's track-changes file)

21. **L96: "sampling of the standard (diatomite)" What does sampling mean here? Was the diatomite measured on several days? Were the columns prepared again? Was it prepared again?**

**Response:** We are sorry for the confusion caused by the incorrect word. In the revised manuscript, the phrase "…repeated sampling of the standard (diatomite)…" has been revised to "…repeated analysis of the standard (diatomite)…". (Please see line 141 in the Author's track-changes file)

22. **L98: What do you mean by 2σ absolute?**

**Response:** We are sorry for the confusion. The Greek letter σ (sigma) is the symbol for standard deviation. To ensure consistency in the abbreviation of standard deviation throughout the manuscript, the statement "All uncertainties are reported at 2σ absolute" has been revised to "All uncertainties are reported at 2 standard deviations (2 SD)" in the revised manuscript. (Please see lines 144-145 in the Author's track-changes file)

23. **L98-100: I don't see the value of adding the error of the diatomite to your analytical uncertainty. The diatomite and other standards are mainly measured to see if you get the correct values. If at all you can use the diatomite value to normalize your values. But why would you add the uncertainty?**

**Response:** Running Diatomite throughout the analytical session allows us to calculate excess variance i.e. measurements of Diatomite represent a single population, which should yield an MSWD (chi-square) of 1. Values much above 1 indicate the analytical error alone does not adequately capture the session variance and, as such, the addition of an excess is necessary.

**24. L100: Reproducibility and instrument accuracy are generally indicated based on the NBS28, I don't see any values for the NBS28 at all.**

**Response:** Thanks for the reviewer's comment. NBS28 is used as the primary reference against which all other samples are normalised. Therefore, the normalised value derived for the secondary reference material (Diatomite) is viewed as a reflection of the accuracy of the reproducibility and instrument.

**25. L107: Remove "for" after "To assess".**

**Response:** We appreciate reviewer's careful review. The word "for" after "To assess" has been removed in the revised manuscript. (Please see line 154 in the Author's track-changes file)

**26. Figure 3: The numbers in the figure and species names in the legend are hard to read. I suggest increasing the size of the figure for publication.**

**Response:** Thanks for the reviewer's suggestion. In the revised manuscript, we have increased the size of Figure 3. Furthermore, we have also increased the front size of the numbers within the figure from 4 pt to 7 pt, and the front size of species names in the legend from 5 pt to 7 pt. (Please see Figure 3 in the clean version of the revised manuscript)

**27. L125: Change to "constituents".**

**Response:** Yes, the term "constitutes" has been replaced with "constituents". (Please see line 179 in the Author's track-changes file)

**28. L126: Remove "all".**

**Response:** Yes, the word "all" has been removed. Furthermore, according to the comment from the other reviewer, the statement in the manuscript "The concentrations of other elements (Fe, Al, Mg, Sr, Na, K and Cl) were all below the minimum detectable limit for the EDS detector (Figure 4)" has been revised to "other elements (Fe, Al, Mg, Sr, Na, K and Cl) flagged in red within the EDS (A" and B") spectrum images (Figure 5) indicates that either their concentrations were below the minimum detectable limit (~ 0.1 %) for the EDS detector, or the software lacked a 99% statistical confidence in the presence of these elemental signatures within the analysed sample". (Please see lines 179-183 in the Author's track-changes file)

**29. Figure 4: I am wondering if samples have been checked at higher magnification as well. If radiolaria are not completely clean, contaminants are found within the radiolarian structure and are not visible at this resolution with EDS. These would rather be seen by looking closer at individual specimens. Also, what contaminations would the authors have expected to see with the help of the EDS and SEM here that would affect the silicon isotope composition? If the main goal was to check for the effects of dissolution, I would have expected a much higher magnification of the SEM to be able to see if there were traces of dissolution or remineralization. which is hard with these pictures. Contamination by other Si phases, such as diatoms and sponges, is already visible with the light microscope. The SEM and EDS with this kind of use only show that there seems to be no clay contamination, which would also be better seen at higher magnification.**

**Response:** Thanks for the reviewer's constructive comment. Yes, the extracted radiolarian tests

have also been examined at higher magnification. We did not include high-magnification images of the radiolarian shells in Figure 4 in the original manuscript, because images derived from the SEM and EDS analyses were primarily employed to further ascertain the potential presence of any extraneous siliceous phases, such as silicate detritus that potentially affect the silicon isotope composition of the radiolarians. Even if contaminants specifically from siliceous detrital material are present within the internal structures of the radiolarian shells, they can be identified using EDS analysis. The numerous pores on the surface of radiolarian shells allowed the electron beam to penetrate through these pores during EDS mapping analysis, thereby enabling the identification of siliceous detrital material within the shell based on elemental composition.

The reviewer is correct to note that potential contamination from diatoms and sponge spicules could be assessed using light microscopy. However, some other contaminations from silicate detritus, such as clay mineral particles (typically less than 2 μm in size), which could adhere to or fill the radiolarian shells, are challenging to discern clearly even with high-magnification (400x) light microscopy. We, therefore, conducted EDS analysis on extracted radiolarian tests, as silicate detritus typically contains aluminium (Al) and iron (Fe), and can be identified by elemental composition via EDS mapping. Consequently, we used both light microscopy and SEM coupled with an EDS detector to evaluate the purity of radiolarian tests, as stated in lines 81-84 of the original manuscript.

Based on the reviewer's valuable comment, we have included a new Figure (Figure 4, please refer to the figure in the response to main question 3 above) in the revised manuscript to indicate the preservation state of radiolarian shells and the potential for dissolution-induced alteration of the radiolarian skeleton (the original Figures 4 and 5 have been renumbered as Figures 5 and 6, respectively, in the revised manuscript). Consequently, we have made following revisions to the manuscript as detailed in the response to main question 3 above:

In the revised manuscript, the title of section 2.2 "Radiolarian composition analysis" has been revised to "Radiolarian composition and preservation analysis" (Please see line 85 in the Author's track-changes file), and we have included a second paragraph regarding assessing the preservation of radiolarian shells: "To assess the preservation of radiolarian shells both within the water column and seabed sediments, the proportion of fractured radiolarian shells was quantified in each studied sample under a Nikon optical microscope at x100 magnification. Further scanning electron microscopy (SEM) was employed to examine the potential for dissolution-induced alteration of the radiolarian skeleton.". (Please see lines 103-106 in the Author's track-changes file)

Following the first paragraph of section "3 Results", we have included the result of radiolarian preservation analysis and a new Figure 4: "The proportion of fractured shells was low in the studied samples (Figure 4), generally ranging from 2 to 5% (mean = 3%) in plankton samples and from 3 to 9% (mean = 6%) in surface sediments. SEM images revealed a typical morphology of the pores on the radiolarian shell (Figure 4E), along with a high degree of integrity in the delicate skeletal structures (Figure 4F). These observations suggest good preservation of radiolarian shells, with no significant dissolution evident in either the water column or sediment samples." (Please see lines 171-175 in the Author's track-changes file)

The first paragraph of Section "4.3 Transfer of radiolarian $\delta^{30}$Si signatures into the sediment record" has been revised to "At each sampling station, $\delta^{30}$Si$_{rad}$ compositions (mean = 1.73‰) in the surface sediment closely resemble those (mean = 1.74‰) in the overlying water column evidenced by the paired t-test (p=0.75), indicating a faithful transfer of the $\delta^{30}$Si signal

incorporated into radiolarian skeletons from the water column to sediments. This suggests that dissolution has a minimal impact on the $\delta^{30}Si_{rad}$ signatures as radiolarian shells sink through the water column and become incorporated into the sediment record. One of two possibilities may account for this observation: 1) the radiolarian shells may not have undergone substantial dissolution during sinking; 2) the radiolarian shells have experienced substantial dissolution, but this process may not significantly alter their isotope composition. As shell fracture of microfossils preserved within sediments is commonly attributed to partial dissolution (e.g. Murray and Alve, 1999; Ryves et al., 2001), the proportion of fractured radiolarian shells may indicate the potential for radiolarian shell dissolution during sinking. In this study, fractured radiolarian shells comprised only a mean of 3% in plankton samples and 6% in surface sediment (Figure 4). Considering the low proportion of fractured radiolarian shells, the well-preserved state of radiolarian shells (Figure 4), and the minor differences in the relative abundances of prominent radiolarian species between plankton samples and surface sediments at each sampling station (as detailed in section 4.1), we propose that radiolarian shells have not experienced substantial dissolution or remineralisation during their transfer from the water column to the sediments. (Please see lines 268-281 in the Author's track-changes file)

At the beginning of the second paragraph of section "4.3 Transfer of radiolarian $\delta^{30}Si$ signatures into the sediment record", we have added "The susceptibility of radiolarian shells to dissolution varies considerably with different taxonomic groups due to the differences in the chemical constituent of their skeletons." (Please see lines 282-283 in the Author's track-changes file)

**References:**

Murray, J. W., Alve, E.: Natural dissolution of modern shallow water benthic foraminifera: taphonomic effects on the palaeoecological record, Palaeogeography, Palaeoclimatology, Palaeoecology, 146(1-4), 195-209, https://doi.org/10.1016/S0031-0182(98)00132-1, 1999.

Ragueneau, O., Tréguer, P., Leynaert, A., Anderson, R. F., Brzezinski, M. A., DeMaster, D. J., Dugdale, R. C., Dymond, J., Fischer, G., Francois, R., Heinze, C., Maier-Reimer, E., Martin-Jézéquel, V. Nelson D. M., Quéguiner, B.: A review of the Si cycle in the modern ocean: recent progress and missing gaps in the application of biogenic opal as a paleoproductivity proxy, Global. Planet. Change, 26(4), 317-365, https://doi.org/10.1016/S0921-8181(00)00052-7, 2000.

Ryves, D. B., Juggins, S., Fritz, S. C., Battarbee, R. W.: Experimental diatom dissolution and the quantification of microfossil preservation in sediments. Palaeogeography, Palaeoclimatology, Palaeoecology, 172(1-2), 99-113, https://doi.org/10.1016/S0031-0182(01)00273-5, 2001.

Treguer, P., Nelson, D. M., Van Bennekom, A. J., DeMaster, D. J., Leynaert, A., Quéguiner, B. The silica balance in the world ocean: a reestimate, Science, 268(5209), 375-379, https://doi.org/10.1126/science.268.5209.375, 1995.

30. **Figure 5: $\Delta^{30}Si$ is usually referred to as the fractionation factor, so its use here is a bit confusing. Also, (A) already indicates that the data are the same within error, so I see little in plotting the difference here as they are within your analytical uncertainty.**

**Response:** We are sorry for the confusion from the use of the triangle symbol "∆". Yes, the reviewer is correct to point out that $\Delta^{30}Si$ is usually used as the fractionation factor. In this context, the symbol "∆" represents the uppercase of Greek letter Delta when denoting the fractionation factor. To distinguish this from the "∆" used for the fractionation factor, we, in fact, employed a triangle symbol "∆" in Figure 5B to represent the differences in $\delta^{30}Si_{rad}$ values between plankton

samples and surface sediments at each sampling station. While similar in appearance, these two symbols are actually distinct.

To avoid this confusion, we have replaced the triangle symbol "∆" with the capital letter "D" of "Different" in Figure 5B of the manuscript. (Please see Figure 6 in the clean version of the revised manuscript)

Yes, the data presented in Figure 6A (Figure 5A in the original manuscript) are consistent within the reported errors. However, these data points represent the mean $\delta^{30}Si_{rad}$ values measured by the Neptune ICP-MS, and also exhibit discrepancies between paired $\delta^{30}Si_{rad}$ data. These differences, or shifts, in $\delta^{30}Si_{rad}$ values between the water column and sediment samples are presented in Figure 6B (Figure 5B in the original manuscript) to offer a supplementary perspective to the information provided in Figure 6A. Consequently, we respectfully suggest retaining Figure 6B in the revised manuscript.

**31. L134-140: The discussion starts with a comparison of the absolute $\delta^{30}Si_{rad}$ values measured in this study with previous studies. While it is important to note that the range in silicon isotopes measured here agrees with previous studies, I don't see a value in solely comparing absolute values between regions here. While this can serve as an introduction to the discussion, a comparison of the impact of source signatures of $\delta^{30}Si$-DSi and potential fractionation factors of radiolaria is missing from the discussion so far.**

**Response:** Thanks for reviewer's comment. We present a comparison of the absolute radiolarian silicon isotope ($\delta^{30}Si_{rad}$) values obtained in this study with those from previous studies, as this comparison may provide a fundamental dataset for understanding the range of $\delta^{30}Si_{rad}$ composition and its spatial distribution in the oceans. Regarding the discussion on the fractionation factor of the radiolarian, we have added a new Section "4.2 Radiolaria silicon isotope fractionation" in the revised manuscript, as detailed below. (Please see lines 244-266 in the Author's track-changes file)

"4.2 Radiolaria silicon isotope fractionation

The isotope fractionation factor between radiolaria and seawater is a critical parameter for reconstructing past silicon cycle in the mid-upper ocean using $\delta^{30}Si_{rad}$ in downcore records (Hendry et al., 2014; Abelmann et al., 2015; Doering et al., 2021). As discussed previously, the $\delta^{30}Si_{rad}$ signatures in the water column of the SCS are predominantly contributed to by radiolarians from the 0-100 m depth layer. Therefore, $\delta^{30}Si_{sw}$ data pertinent to constraining silicon isotope fractionation by radiolarians in the SCS should derived primarily from this depth interval. Currently, there is no corresponding seawater $\delta^{30}Si$ ($\delta^{30}Si_{sw}$) data from the stations in the SCS where water column radiolaria samples were collected. Previous work, however, has measured seasonal changes in $\delta^{30}Si_{sw}$ in the northern SCS (Cao et al 2012). At the two sites from that study that are off the continental shelf (A10 and SEATS), $\delta^{30}Si_{sw}$ above 100 m varies from 1.4‰ to 2.9‰. Assuming: 1) these values and their depth profiles are broadly comparable to stations where $\delta^{30}Si_{rad}$ was measured; and 2) that radiolarians construct their skeletons in equilibrium with ambient seawater (Abelmann et al., 2015, Fontorbe et al., 2016, Doering et al 2021), the isotope fractionation factor (ε) can be calculated as:

$$\varepsilon \sim \Delta^{30}Si = \delta^{30}Si_{rad} - \delta^{30}Si_{sw} \tag{2}$$

Using Monte Carlo simulations (10,000 replicates) to account for the 2SD uncertainty of both $\delta^{30}Si_{rad}$ and $\delta^{30}Si_{sw}$, ε is estimated to range from –0.33‰ to –0.92‰ (mean = 0.58‰) (Table S1 in

supplementary material 1). Due to the difficulties in culturing radiolarian (e.g. Suzuki and Not, 2015), relatively little is known about the fractionation factor for $\delta^{30}Si_{rad}$. Values for ε obtained here from the SCS are comparable to those measured by Abelmann et al. (2015) in the Southern Ocean (–0.54‰ to –0.91‰, mean = -0.75‰) and by Doering et al. (2021) along the Peruvian Shelf (–0.35‰ to –0.79‰, mean = -0.62‰, mixed radiolaria). The large range of values for ε has been suggested to indicate species/order-specific fractionation during the uptake of silicic acid by radiolaria (Doering et al., 2021). Such a process would necessitate the extraction of species-specific radiolarian samples for isotope analysis in future palaeoceanographic research (e.g., Zhang and Swann, 2023). However in this current study from the SCS, whilst the species composition of the $\delta^{30}Si_{rad}$ samples is known (Figure 3), an absence of matching $\delta^{30}Si_{sw}$ data from exactly the same sampling stations prevents this issue being further investigated at this time."

In addition, the phrase "with the fractionation factor for $\delta^{30}Si_{rad}$ varying from –0.33‰ to –0.92‰ (mean = 0.58‰)" has been added into both the "Abstract" (Please see lines 7-8 in the Author's track-changes file) and Section "5 Conclusions". (Please see lines 319-320 in the Author's track-changes file)

**Reference:**

Abelmann, A., Gersonde, R., Knorr, G., Zhang, X., Chaplgin, B., Maier, E., Esper, O., Friedrichsen, H., Lohmann, G., Meyer, H., Tiedemann, R.: The seasonal sea-ice zone in the glacial Southern Ocean as a carbon sink, Nat. Commun., 6(1), 8136, https://doi.org/10.1038/ncomms9136, 2015.

Cao, Z., Frank, M., Dai, M., Grasse, P., Ehlert, C.: Silicon isotope constraints on sources and utilization of silicic acid in the northern South China Sea, Geochimica et Cosmochimica Acta, 97, 88-104, https://doi.org/10.1016/j.gca.2012.08.039, 2012.

Doering, K., Ehlert, C., Pahnke, K., Frank, M., Schneider, R., Grasse, P.: Silicon isotope signatures of radiolaria reveal taxon-specific differences in isotope fractionation, Front. Mar. Sci., 8, 666896, https://doi.org/10.3389/fmars.2021.666896, 2021.

Fontorbe G, Frings P J, De La Rocha, C., Hendry, K. R., Conley, D. J.: A silicon depleted North Atlantic since the Palaeogene: Evidence from sponge and radiolarian silicon isotopes, Earth Planet. Sci. Lett., 453, 67-77, https://doi.org/10.1016/j.epsl.2016.08.006, 2016.

Suzuki, N., Not, F: Biology and Ecology of Radiolaria, In: Marine Protists, edited by: Ohtsuka, S., Suzaki, T., Horiguchi, T., Suzuki, N., Not, F., Springer, Tokyo, Japan, https://doi.org/10.1007/978-4-431-55130-0_8, 2015.

Zhang, Q., Swann, G. E. A.: An effective method to extract and purify radiolaria from tropical marine sediments, Front. Mar. Sci., 10, 1150518, https://doi.org/10.3389/fmars.2023.1150518, 2023.

**32. L152: Correct to "all the radiolarian shells".**

**Response:** The phrase "the all radiolarian shells" has been revised to "all the radiolarian shells". (Please see line 210 in the Author's track-changes file)

**33. L 153: Why did you expect the $\delta^{30}Si_{rad}$ values to differ between depths? (100-300m versus 0-100m).**

**Response:** A previous study documented variations in the $\delta^{30}Si$ composition among different radiolarian taxa (Doering et al., 2021). Based on findings regarding the vertical distribution of living radiolarians in the water column of the South China Sea using protoplasm staining, Hu et al., (2015) demonstrated the variations in both radiolarian abundance and dominant species composition between the 0-75 m and 75-300 m water layers. Consequently, it is expected that

$\delta^{30}Si_{rad}$ values from plankton samples at 100-300 m water depth would potentially differ from those at 0-100 m.

In the revised manuscript, we have briefly explained why $\delta^{30}Si_{rad}$ values from plankton samples at 100-300 m water depth were expected to potentially differ from those at 0-100 m. The statement in the manuscript "Whilst $\delta^{30}Si_{rad}$ values from plankton samples at 100-300 m water depth were expected to potentially differ from those at 0-100 m, no significant difference was detected (p = 0.52)." has been revised to "Given the differences in both the abundance and dominant species composition of living radiolarians between 0-75 m and the 75-300m water depth in the SCS (Hu et al., 2015), it is expected that $\delta^{30}Si_{rad}$ values from plankton samples at 100-300 m water depth might potentially differ from those at 0-100 m. However, no significant difference was detected (p = 0.52)." (Please see lines 212-215 in the Author's track-changes file)

**References:**

Doering, K., Ehlert, C., Pahnke, K., Frank, M., Schneider, R., Grasse, P.: Silicon isotope signatures of radiolaria reveal taxon-specific differences in isotope fractionation, Front. Mar. Sci., 8, 666896, https://doi.org/10.3389/fmars.2021.666896, 2021.

Hu, W. F., Zhang, L. L., Chen, M. H., Zeng, L. L., Zhou, W. H., Xiang, R., Zhang, Q., Liu, S. H.: Distribution of living radiolarians in spring in the South China Sea and its responses to environmental factors, Sci. China Earth Sci., 58, 270-85, https://doi.org/10.1007/s11430-014-4950-0, 2015

34. L170: Remove "(vital effects)" here.

**Response:** Yes, the phrase "(vital effects)" has been removed. (Please see line 232 in the Author's track-changes file)

**35. L169-172: I don't think this discussion is correct here. As far as I can see, there is no significant difference in species compositions or $\delta^{30}Si_{rad}$ between your water column and surface sediment data (line 153-159). So even if there is a difference in d30Sirad and/or fractionation between radiolarian species/order, there is no way to see any effect of this in your data.**

**Response:** Thanks for reviewer's comment. While statistical analysis suggests no significant difference in either the composition of prominent species or their relative abundances between plankton samples and surface sediments, variations in the relative abundances of some prominent radiolarian species are indeed observed. Due to the high diversity of radiolarians in low-latitude oceans, few species comprise more than 10% of the radiolarian community, and those exceeding 2-3% are generally defined as prominent species (e.g. Zhang et al., 2009; Hu et al., 2015). Therefore, a discernible difference in the relative abundance of a prominent species is considered to exist when the variation in its relative abundance between samples reaches approximately or exceeds a factor of two.

Radiolarian census data in this study showed that the relative abundances of certain prominent radiolarian species, including *Botryocyrtis scutum, Didymocyrtis tetrathalamus t.*, *Tetrapyle octacantha/quadribola*, *Zygocircus capulosus*, varied by near a factor of two or more between the water column (0-100) and surface sediments (please see table A below and Figure 3 in the manuscript; original radiolarian count data has been provided as supplementary material 2 in the revised manuscript) at some stations (e.g. station 13). However, despite variations in the relative abundance of certain prominent radiolarian species, no substantial differences in $\delta^{30}Si_{rad}$ values

were observed between plankton samples and the sediment at each sampling station (Table S1 in the supplementary material 1). We therefore propose two potential reasons for the consistency of $\delta^{30}Si_{rad}$ values between plankton samples and the sediments, as detailed in lines 169-175 of the original manuscript: 1) the minor differences in the relative abundance of most prominent taxa between water column and sediment samples; 2) the high diversity of radiolarians in each sample averaging out the $\delta^{30}Si_{rad}$ signal across different taxa, mitigating potential bias from individual taxa.

Table A Variations in the relative abundance of typical prominent species between the water column (0-100) and surface sediments at station 13

| Prominent species / Sample | Station 13 (0-100 m) | Station 13 (surface sediment) |
|---|---|---|
| *Tetrapyle octacantha/quadribola* | 4.0 % | 8.8 % |
| *Didymocyrtis tetrathalamus t.* | 8.1 % | 2.4 % |
| *Tetrapyle octacantha/quadribola* | 11.7 % | 18.1 % |
| *Zygocircus capulosus* | 2.2 % | 6.2 % |

For clarification, in the revised manuscript, the statement "Although minor differences in the relative abundances of radiolarian species are observed…" has been revised to "Although discernible differences in the relative abundances of some prominent radiolarian species are observed…" (Please see lines 226-227 in the Author's track-changes file). Moreover, the statement "Although variations in the $\delta^{30}Si$ composition have been documented among different radiolarian taxa (Doering et al., 2021), the relative abundance differences between plankton and surface sediment samples do not result in significant $\delta^{30}Si_{rad}$ disparities in our study in the SCS" has been revised to "Although variations in the $\delta^{30}Si$ composition have been documented among different radiolarian taxa (Doering et al., 2021), the relative abundance differences in certain prominent species, such as *Botryocyrtis scutum*, *Didymocyrtis tetrathalamus t.*, *Tetrapyle octacantha/quadribola*, and *Zygocircus capulosus* between plankton and surface sediment samples (Figure 3) do not result in significant $\delta^{30}Si_{rad}$ disparities in our study in the SCS". (Please see lines 233-235 in the Author's track-changes file)

**L180-215: Section 4.2 Transfer of radiolarian d30Si signatures into the sediment record need some re-structuring and rewriting. The first paragraph is fine, picking up on the fact that there is little difference between the water column and the sediment d30Sirad. The following paragraphs, however, are mainly a summary of the literature and should be shortened, and the information from the radiolarian abundances investigated here should be more highlighted. Additionally, references to the SEM and EDS analysis conducted are not referred to at all here.**

**Response:** Due to the inclusion of a new Section "4.2 Radiolaria silicon isotope fractionation" in the revised manuscript, the original Section 4.2 has been renumbere as Section 4.3. As responded to the main question 3, Section "4.3 Transfer of radiolarian $\delta^{30}Si$ signatures into the sediment record" focuses on explaining why the $\delta^{30}Si_{rad}$ signal from the water column is faithfully transferred to the sediments. Our data indicate a good consistency between $\delta^{30}Si_{rad}$ signatures in the water column and the sedimentary record, suggesting that dissolution may have a minimal impact on the $\delta^{30}Si_{rad}$ composition as shells sink through the water column and become incorporated into the sediment record. This may be attributable to one of two possible reasons: 1)

the radiolarian shells may not have undergone substantial dissolution during sinking; 2) the radiolarian shells have experienced substantial dissolution, but this process might not have significantly altered their isotope composition. As shell fracture of microfossils preserved within sediments is commonly attributed to partial dissolution (e.g. Murray and Alve, 1999; Ryves et al., 2001), the proportion of fractured radiolarian shells may indicate the potential for radiolarian shell dissolution during sinking. In this study, the proportion of fractured radiolarian shells was low, and showed only slight difference between plankton samples (mean = 3%) and surface sediments (mean = 6%) (Figure 4). Considering the low proportion of fractured radiolarian shells, the well-preserved state of radiolarian shells (Figure 4), and the minor differences in the relative abundances of prominent radiolarian species between plankton samples and surface sediments at each sampling station (as detailed in section 4.1), we propose that radiolarian shells have not experienced substantial dissolution or remineralisation during their transfer from the water column to the sediments.

The susceptibility of radiolarian shells varies considerably with different taxonomic groups due to notable differences in the chemical constituent of their skeletons. For example, Collodaria radiolarians, a group belonging to polycystine radiolarians, are highly susceptible to dissolution during sinking due to their skeletons composed of both opal and Celestine (Afanasieva et al., 2005; Afanasieva and Amon, 2014). However, as stated above, our observations do not provide evidence that the radiolarian shells have experienced substantial dissolution during sinking within the water column. Only by clarifying the dissolution characteristics of different radiolarian groups which are closely related to the chemical compositions of their shells, alongside the approximate proportion of each group within the radiolarian community of our studied samples, can we better understand why the radiolarian shells are minimally impacted by dissolution during sinking through the water column. This allows for a compelling explanation of why the radiolarian silicon isotope signal from the water column is faithfully transferred to the sediments. Therefore, the information presented in Section 4.3 regarding the radiolarian taxonomy, chemical composition and dissolution characteristics of each taxonomic group, and the proportion of each taxonomic group within the radiolarian community in our analyzed samples, is essential for this study, despite containing a summary of the literature.

The information of the radiolarian abundances in various water depths was primarily used to address the depth range from which the radiolarian community contributes to the $\delta^{30}Si_{rad}$ signature in the water column and sediments, and has been detailed in Section 4.1 as stated in the response to the main question 3.

The SEM and EDS images in the original Figure 4 were primarily employed to examine the purity and cleanliness of the radiolarian tests extracted from the plankton samples and surface sediments, thereby ensuring the reliability of the $\delta^{30}Si_{rad}$ data in this study. As such, the results of the SEM and EDS analysis were not referred in Section 4.2 of the original manuscript. However, we agree with the reviewer that the SEM images with a higher magnification may trace the potential dissolution of radiolarian shells as reviewer suggested above. Therefore, we have included a new Figure (Figure 4, please refer to the figure in the response to main question 3 above) in the revised manuscript to indicate the preservation state of radiolarian shells and the potential for dissolution-induced alteration of the radiolarian skeleton (the original Figures 4 and 5 have been renumbered as Figures 5 and 6, respectively, in the revised manuscript). Accordingly, we have made following revisions to the manuscript as detailed in the response to

main question 3 above:

In section 2.2 of the revised manuscript, we have included a paragraph to assess the preservation of radiolarian shells following the line 78 "To assess the preservation of radiolarian shells both within the water column and seabed sediments, the proportion of fractured radiolarian shells was quantified in each studied sample under a Nikon optical microscope at x100 magnification. Further scanning electron microscopy (SEM) was employed to examine the potential for dissolution-induced alteration of the radiolarian skeleton.". (Please see lines 103-106 in the Author's track-changes file)

Following the first paragraph of section "3 Results", we have included the result of radiolarian preservation analysis "The proportion of fractured shells was low in the studied samples (Figure 4), generally ranging from 2 to 5% (mean = 3%) in plankton samples and from 3 to 9% (mean = 6%) in surface sediments. SEM images revealed a typical morphology of the pores on the radiolarian shell (Figure 4E), along with a high degree of integrity in the delicate skeletal structures (Figure 4F). These observations suggest good preservation of radiolarian shells, with no significant dissolution evident in either the water column or sediment samples." (Please see lines 171-175 in the Author's track-changes file)

The first paragraph of Section "4.3 Transfer of radiolarian $\delta^{30}$Si signatures into the sediment record" has been revised to "At each sampling station, $\delta^{30}$Si$_{rad}$ compositions (mean = 1.73‰) in the surface sediment closely resemble those (mean = 1.74‰) in the overlying water column evidenced by the paired t-test (p=0.75), indicating a faithful transfer of the $\delta^{30}$Si signal incorporated into radiolarian skeletons from the water column to sediments. This suggests that dissolution has a minimal impact on the $\delta^{30}$Si$_{rad}$ signatures as radiolarian shells sink through the water column and become incorporated into the sediment record. One of two possibilities may account for this observation: 1) the radiolarian shells may not have undergone substantial dissolution during sinking; 2) the radiolarian shells have experienced substantial dissolution, but this process may not significantly alter their isotope composition. As shell fracture of microfossils preserved within sediments is commonly attributed to partial dissolution (e.g. Murray and Alve, 1999; Ryves et al., 2001), the proportion of fractured radiolarian shells may indicate the potential for radiolarian shell dissolution during sinking. In this study, fractured radiolarian shells comprised only a mean of 3% in plankton samples and 6% in surface sediment (Figure 4). Considering the low proportion of fractured radiolarian shells, the well-preserved state of radiolarian shells (Figure 4), and the minor differences in the relative abundances of prominent radiolarian species between plankton samples and surface sediments at each sampling station (as detailed in section 4.1), we propose that radiolarian shells have not experienced substantial dissolution or remineralisation during their transfer from the water column to the sediments. (Please see lines 268-281 in the Author's track-changes file)

At the beginning of the second paragraph of section "4.3 Transfer of radiolarian $\delta^{30}$Si signatures into the sediment record", we have added "The susceptibility of radiolarian shells to dissolution varies considerably with different taxonomic groups due to the differences in the chemical constituent of their skeletons." (Please see lines 282-283 in the Author's track-changes file)

**36. L181: Change to "resemble".**
**Response:** Yes, the term "resembles" has been revised to "resemble". (Please see line 268 in the Author's track-changes file)

**37. L201: Add a comma after "in the SCS".**

**Response:** Yes, we have included a comma after "in the SCS" in Line 201 of the current manuscript. (Please see line 298 in the Author's track-changes file)

**38. L206: Change to "in this study,…".**

**Response:** Yes, the phrase "in the current study…" in line 206 of the manuscript has been revised to "in this study,…". (Please see line 303 in the Author's track-changes file)

**39. L209: Correct to "expected to have..".**

**Response:** Yes, the phrase "dissolution is expected have…" in Line 209 has been revised to "dissolution is expected to have…". (Please see line 306 in the Author's track-changes file)